# Graph Posterior Network: Bayesian Predictive Uncertainty for Node Classification

**Maximilian Stadler,**[*] **Bertrand Charpentier,**[*] **Simon Geisler, Daniel Zügner,**
**Stephan Günnemann**
Department of Informatics
Technical University of Munich, Germany
`{stadlmax, charpent, geisler, zuegnerd, guennemann}@in.tum.de`

## Abstract

The interdependence between nodes in graphs is key to improve class predictions on nodes and utilized in approaches like Label Propagation (LP) or in Graph Neural Networks (GNNs). Nonetheless, uncertainty estimation for non-independent node-level predictions is under-explored. In this work, we explore uncertainty quantification for node classification in three ways: **(1)** We derive three axioms explicitly characterizing the expected predictive uncertainty behavior in homophilic attributed graphs. **(2)** We propose a new model Graph Posterior Network (GPN) which explicitly performs Bayesian posterior updates for predictions on *interdependent* nodes. GPN provably obeys the proposed axioms. **(3)** We extensively evaluate GPN and a strong set of baselines on semi-supervised node classification including detection of anomalous features, and detection of left-out classes. GPN outperforms existing approaches for uncertainty estimation in the experiments.

## 1 Introduction

Accurate and rigorous uncertainty estimation is key for reliable machine learning models in safety-critical domains [67]. It quantifies the confidence of machine learning models, thus allowing them to validate knowledgeable predictions or flag predictions on unknown input domains. Uncertainty is commonly divided in *aleatoric* and *epistemic* uncertainty [28]. The aleatoric uncertainty accounts for irreducible uncertainty (e.g., due to inherent sensor noise). The *epistemic* uncertainty accounts for a lack of information for accurate prediction (e.g., test data significantly different from training data).

Traditionally, machine learning models assume i.i.d. inputs, thus performing predictions based on input features only. For uncertainty estimation on i.i.d. inputs, a large class of definitions, models and evaluation methods have been introduced [28, 62, 3, 78, 50]. Further, uncertainty estimation has been successfully applied to different tasks e.g. out-of-distribution (OOD) or shift detection [78], active learning [75, 55], continual learning [4] or reinforcement learning [18].

In contrast, uncertainty estimation on interdependent nodes is more complex than on i.i.d. inputs and under-explored [3]. A node in an attributed graph is characterized by two types of information: its features and its neighborhood. While the feature information indicates the node position in the feature space – similarly to i.i.d. inputs –, the neighborhood information indicates the additional node position in the network space. To leverage the neighborhood information, recent graph neural networks (GNNs) successfully proposed to enrich and correct the possibly noisy information of the features of a single node by aggregating them with the features of its neighborhood [46, 92, 48]. It naturally leads to the distinction between predictions *without network effects* based exclusively on their own node feature representation, and predictions *with network effects* based on neighborhood

---

[*]equal contribution

35th Conference on Neural Information Processing Systems (NeurIPS 2021).

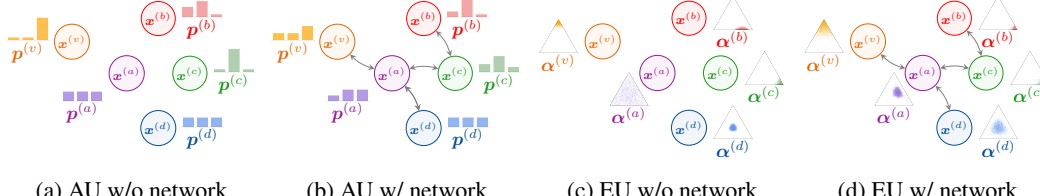

(a) AU w/o network      (b) AU w/ network      (c) EU w/o network      (d) EU w/ network

Figure 1: Illustration of aleatoric uncertainty (AU) and epistemic uncertainty (EU) without and with network effects (i.e. i.i.d. inputs vs interdependent inputs). Nodes have the same features in all cases. Network effects are visualized through edges between nodes which change the predicted distributions. The aleatoric uncertainty is high if the categorical distribution $\hat{y}^{(v)} \sim \text{Cat}(\boldsymbol{p}^{(v)})$ is flat. The epistemic uncertainty is high if the Dirichlet distribution $\boldsymbol{p}^{(v)} \sim \text{Dir}(\boldsymbol{\alpha}^{(v)})$ is spread out. We refer the reader to Section 3.2 for formal definitions of those distributions.

aggregation. The aggregation step commonly assumes *network homophily* which states that nodes with similar properties tend to connect to each other more densely, thus violating the i.i.d. assumption between node features given their neighborhood.

The core motivation of our work is to transfer some of the existing uncertainty estimation definitions, models and evaluations from i.i.d. inputs to interdependent node inputs by leveraging both the feature and the neighborhood information. In particular, we aim at an accurate quantification of the aleatoric and epistemic uncertainty without and with network effect under network homophily (see Fig. 1).

**Our contribution.** In this work, we consider uncertainty estimation on semi-supervised node classification. First, we derive three axioms which materialize reasonable uncertainty for non-independent inputs. These axioms cover the traditional notions of aleatoric and epistemic uncertainty and distinguish between the uncertainty with and without network effects. Second, we propose Graph Posterior Network (GPN)[2] for uncertainty estimation for node classification and prove formally that it follows the axiom requirements contrary to popular GNNs. Third, we build an extensive evaluation setup for uncertainty estimation which relies on the assessment of uncertainty estimation quality of OOD detection and robustness against shifts of the attributed graph properties. Both OOD data and attributed graph shifts distinguish between attribute and structure anomalies. The theoretical properties of GPN manifest in these experiments where it outperforms all other baselines on uncertainty evaluation.

## 2 Related Work

In this section, we cover the related work for predictive uncertainty estimation for i.i.d. inputs and for graphs. To this end, we review the commonly accepted *axioms* defining the desired uncertainty estimation under different circumstances, the *methods* capable of consistent uncertainty quantification and the *evaluation* validating the quality of the uncertainty estimates in practice.

**Uncertainty for i.i.d. inputs –** The related work for uncertainty quantification on i.i.d. inputs is rich as for example shown in a recent survey [3]. Axioms: Far from ID data, the predicted uncertainty is expected to be high [66, 15, 51, 30]. Close to ID data, the desired uncertainty is more complicated. Indeed, while some works expected models to be robust to small dataset shifts [78, 89], other works expected to detect near OOD classes based on uncertainty [98, 50, 13]. Methods: Many methods already exist for uncertainty quantification for i.i.d. inputs like images or tabular data. A first family of models quantifies uncertainty by aggregating statistics (e.g. mean, variance or entropy) from sub-networks with different weights. Important examples are ensemble [52, 96, 97, 38], dropout [88] or Bayesian Neural Networks (BNN) [9, 20, 59, 24, 21]. Most of these approaches require multiple forward-passes for uncertainty quantification. Further, dropout and BNN may have other pitfalls regarding their limited applicability to more complex tasks [77, 41, 34, 27]. A second family quantifies uncertainty by using the logit information. Important examples are temperature scaling which rescale the logits after training [35, 56] and energy-based models which interpret the logits as energy scores [57, 33]. A third family of model quantifies uncertainty based on deep Gaussian Processes (GP). Important examples use GP at activation-level [68] or at (last) layer-level [53, 51, 91, 8]. Finally, a last

---

[2]Project page including code at `https://www.daml.in.tum.de/graph-postnet`

family of models quantifies uncertainty by directly parameterizing a conjugate prior distribution over the target variable. Important examples explicitly parameterize prior distributions [86, 63, 60, 61, 6] or posterior distributions [14, 15]. Methods based on GP and conjugate prior usually have the advantage of deterministic and fast inference. *Evaluation:* Previous works have already proposed empirical evaluation of uncertainty estimation by looking at accuracy, calibration or OOD detection metrics under dataset shifts or adversarial perturbations for i.i.d. inputs [78, 50]. In contrast with all these approaches, this work studies uncertainty quantification for classification of *interdependent nodes*.

**Uncertainty for graphs –** Notably, the recent survey [3] points out that there is only a limited number of studies on uncertainty quantification on GNN and semi-supervised learning. Moreover, they recommend proposing new methods. *Axioms:* To the best of our knowledge, only [23] proposed explicit axioms for node classification for non-attributed graphs. They expect disconnected nodes to recover prior predictions and nodes with higher beliefs to be more convincing. In this work, we clarify the desired uncertainty estimation for node classification on attributed graphs based on *motivated and explicit axioms*. *Methods:* The largest family of models for uncertainty for graphs are dropout- or Bayesian-based methods. Important examples propose to drop or assign probabilities to edges [83, 16, 37, 19, 42]. Further works proposed to combine the uncertainty on the graph structure with uncertainty on the transformation weights similarly to BNN [22, 101, 79, 80]. Importantly, these models do not directly quantify uncertainty on the prediction. Similarly to the i.i.d. case, a second family of models focuses on deterministic uncertainty quantification. Important examples mostly use Graph Gaussian Processes, which do not easily scale to large graphs [74, 103, 58, 12]. Only [102] explicitly parameterized a Dirichlet conjugate prior. They combined it with multiple components (Graph-Based Kernel, dropout, Teacher Network, loss regularizations) which cannot easily distinguish between uncertainty without and with network effects. In contrast, GPN is a simple approach based on conjugate prior parametrization and disentangles uncertainty with and without network effects. *Evaluation:* The evaluation of most of those methods was not focused on the quality of the uncertainty estimates but on the target task metrics (e.g. accuracy for classification, distance to ground truth for regression). Some methods focus on robustness of the target task metrics against adversarial perturbations [36, 107, 106]. Other methods only relied on uncertainty quantification to build more robust models [104, 25]. For node classification, only few works evaluated uncertainty by using Left-Out classes or detection of missclassified samples [102], active learning [74] or visualization [12]. Note that proposed uncertainty evaluations on molecules at graph level [100, 84, 5, 40, 90] is an orthogonal problem. In this work, we propose a *sound and extensive evaluation* for uncertainty in node classification. It distinguishes between OOD nodes w.r.t. features and structure, and graph dataset shifts w.r.t. the percentage of perturbed node features and the percentage of perturbed edges.

## 3 Uncertainty Quantification for Node Classification

We consider the task of (semi-supervised) node classification on an attributed graph $\mathcal{G} = (\boldsymbol{A}, \boldsymbol{X})$ with adjacency matrix $\boldsymbol{A} \in \{0,1\}^{N \times N}$ and node attribute matrix $\boldsymbol{X} \in \mathbb{R}^{N \times D}$. We aim at inferring the labels $y^{(v)} \in \{1, ..., C\}$ plus the the aleatoric uncertainty $u_{\text{alea}}^{(v)}$ and the epistemic uncertainty $u_{\text{epist}}^{(v)}$ of unlabeled nodes $v \in \mathcal{T}$ given a set of labelled nodes $u \in \mathcal{U}$ in the graph where $\mathcal{V} = \mathcal{T} \cup \mathcal{U}$ denotes the set of vertices.

### 3.1 Axioms

Uncertainty estimation in the setting of interdependent inputs is not well-studied. It often leaves the expected behavior and interpretations for uncertainty estimation unclear. Thus, we need well-grounded axioms to derive meaningful models. In this section, we aim at specifying the desired uncertainty predictions under various circumstances in homophilic attributed graphs. To this end, we propose three axioms which are based on the two following distinctions. The first distinction differentiates between aleatoric and epistemic uncertainty which are commonly used concepts under the i.i.d. assumptions [28, 62]. The second distinction differentiates between uncertainty without and with network effects which are motivated by the concepts of attribute and structure anomalies used in the attributed graph setting [11]. These new axioms cover all possible combinations encountered by these distinctions and extend the axioms proposed by [23] for non-attributed graphs. We designed the axioms to be informal and generic so that they are application independent, model-agnostic and do not require complex mathematical notations similarly to [23, 76]. In practice, formal definitions need to instantiate general concepts like aleatoric/epistemic uncertainty and with/without network effects

noting that some definitions might be more convenient depending on the task. The first axiom deals with (epistemic and aleatoric) uncertainty estimation without network effects (see Fig. 1a, 1c). :

**Axiom 3.1.** *A node's prediction in the absence of network effects should only depend on its own features. A node with features more different from training features should be assigned higher uncertainty.*

Axiom 3.1 states that if a node $v$ has no neighbors, then the final prediction $\boldsymbol{p}^{(v)}$ should only depend on its own node features $\boldsymbol{x}^{(v)}$. Further, for anomalous features the model should fall back to safe prior predictions, indicating high aleatoric and epistemic uncertainty. This aligns with [23] which expects to recover prior predictions for non-attributed nodes without network effect, and [66, 15] which expect to recover prior predictions far from training data for i.i.d. inputs. The second axiom deals with epistemic uncertainty estimation with network effects (see Fig. 1c, 1d):

**Axiom 3.2.** *All else being equal, if a node's prediction in the absence of network effects is more epistemically certain, then its neighbors' predictions in the presence of network effects should become more epistemically certain.*

Axiom 3.2 states that a node $v$ with confident feature predictions $\boldsymbol{x}^{(v)}$ is expected to be convincing and make its neighbors $u \in \mathcal{N}(v)$ more confident. Conversely, a node with anomalous features is expected to make its neighborhood less confident. This axiom materializes the network homophily assumption at the epistemic level i.e. connected nodes have similar epistemic uncertainty estimates. For non-attributed graphs, [23] similarly expects a more confident node to have more influence on a direct neighbor. The third axiom deals with aleatoric uncertainty estimation with network effects (see Fig. 1a, 1b):

**Axiom 3.3.** *All else being equal, a node's prediction in the presence of network effects should have higher aleatoric uncertainty if its neighbors' predictions in the absence of network effects have high aleatoric uncertainty. Further, a node prediction in the presence network effects should have higher aleatoric uncertainty if its neighbors' predictions in the absence network effects are more conflicting.*

Axiom 3.3 states that no clear classification decision should be made for a node $v$ if no clear classification decisions can be made for its neighbors. Further, the classification decision becomes less certain if a neighbor has a conflicting classification decision. Note that this axiom is more subtle than the direct application of network homophily at the aleatoric level. Indeed a node can have a high aleatoric uncertainty contrary to its neighbors which predict different classes with low aleatoric uncertainty. This aligns with the intuition that conflicting information from the neighborhood provides an irreducible uncertainty to the considered node.

## 3.2 Graph Posterior Network

The Bayesian update rule is a key component of GPN to model uncertainty on the predicted categorical distribution. For a single categorical distribution $y \sim \mathrm{Cat}(\boldsymbol{p})$, the *standard* Bayesian update is straightforward. A natural choice for a prior distribution over the parameters $\boldsymbol{p}$ is its conjugate prior i.e. the Dirichlet distribution $\mathbb{P}(\boldsymbol{p}) = \mathrm{Dir}(\boldsymbol{\alpha}^{\mathrm{prior}})$ with $\alpha_c^{\mathrm{prior}} \in \mathbb{R}_+^C$. Given the observations $y^{(1)}, ..., y^{(N)}$, the Bayesian update then consists in applying the Bayes' theorem

$$\mathbb{P}\left(\boldsymbol{p} \,|\, \{y^{(j)}\}_{j=1}^N\right) \propto \mathbb{P}\left(\{y^{(j)}\}_{j=1}^N \,|\, \boldsymbol{p}\right) \times \mathbb{P}(\boldsymbol{p}) \tag{1}$$

producing the posterior distribution $\mathbb{P}(\boldsymbol{p} \,|\, \{y^{(j)}\}_{j=1}^N) = \mathrm{Dir}(\boldsymbol{\alpha}^{\mathrm{post}})$ where $\boldsymbol{\alpha}^{\mathrm{post}} = \boldsymbol{\alpha}^{\mathrm{prior}} + \boldsymbol{\beta}$ are the parameters of the posterior and $\beta_c = \sum_j = \mathbb{1}_{y^{(j)}=c}$ are the class counts. This framework naturally disentangles the aleatoric and epistemic uncertainty by defining the Dirichlet mean $\bar{\boldsymbol{p}} = \frac{\boldsymbol{\alpha}}{\alpha_0}$ and the total evidence count $\alpha_0 = \sum_c \alpha_c$. Indeed, the aleatoric uncertainty is commonly measured by the entropy of the categorical distribution i.e. $u_{\mathrm{alea}} = \mathbb{H}\left[\mathrm{Cat}(\bar{\boldsymbol{p}})\right]$ [62, 14, 15] and the epistemic uncertainty can be measured by the total evidence count $\alpha_0$ of observations i.e. $u_{\mathrm{epist}} = -\alpha_0$ [14, 15]. Alternatively, the epistemic uncertainty can also be measured with the Dirichlet differential entropy [62]. Note that the reparameterization using $\bar{\boldsymbol{p}}$ and $\alpha_0$ can apply to any class counts including the prior counts $\boldsymbol{\alpha}^{\mathrm{prior}}$, the class counts $\boldsymbol{\beta}$ and the posterior counts $\boldsymbol{\alpha}^{\mathrm{post}}$.

For classification, the predicted categorical distribution $\hat{y}^{(v)} \sim \mathrm{Cat}(\boldsymbol{p}^{(v)})$ additionally depends on the specific input $v$. Hence, the *input-dependent* Bayesian rule [14, 15] extends the Bayesian treatment of a single categorical distribution to classification by predicting an individual posterior update for any possible input. Specifically, it first introduces a fixed Dirichlet prior over the categorical

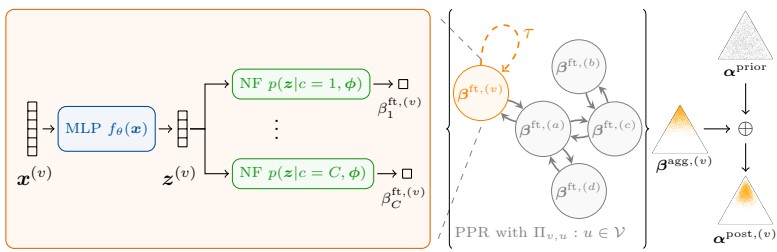

Figure 2: Overview of Graph Posterior Network: (1) node-level pseudo-counts computed by the feature encoder in the orange box, (2) PPR-based message passing visualized between the curly braces, and (3) input-dependent Bayesian update illustrated with the Dirichlet triangles on the right.

distribution $\boldsymbol{p}^{(v)} \sim \text{Dir}(\boldsymbol{\alpha}^{\text{prior}})$ where $\boldsymbol{\alpha}^{\text{prior}} \in \mathbb{R}_+^C$ is usually set to 1, and second predicts the input-dependent update $\boldsymbol{\beta}^{(v)}$ which forms the posterior distribution $\boldsymbol{p}^{(v)} \sim \text{Dir}(\boldsymbol{\alpha}^{\text{post},(v)})$ where the posterior parameters are equal to

$$\boldsymbol{\alpha}^{\text{post},(v)} = \boldsymbol{\alpha}^{\text{prior}} + \boldsymbol{\beta}^{(v)}. \tag{2}$$

The variable $\boldsymbol{\beta}^{(v)}$ can be interpreted as learned class pseudo-counts and its parametrization is crucial. For i.i.d. inputs, PostNet [14] models the pseudo-counts $\boldsymbol{\beta}^{(v)}$ in two main steps. **(1)** it maps the inputs features $\boldsymbol{x}^{(v)}$ onto a low-dimensional latent vector $\boldsymbol{z}^{(v)} = f_\theta(\boldsymbol{x}^{(v)}) \in \mathbb{R}^H$. **(2)**, it fits one conditional probability density $\mathbb{P}(\boldsymbol{z}^{(v)}|c; \boldsymbol{\phi})$ per class on this latent space with normalizing flows. The final pseudo count for class $c$ is set proportional to its respective conditional density i.e. $\beta_c^{(v)} = N \, \mathbb{P}(\boldsymbol{z}^{(v)}|c; \boldsymbol{\phi}) \, \mathbb{P}(c)$ where $N$ is a total certainty budget and $\mathbb{P}(c) = \frac{1}{C}$ for balanced classes. Note that this implies $\alpha_0^{(v)} = N \, \mathbb{P}(\boldsymbol{z}^{(v)}|\boldsymbol{\phi})$. This architecture has the advantage of decreasing the evidence outside the known distribution when increasing the evidence inside the known distribution, thus leading to consistent uncertainty estimation far from training data.

**Bayesian Update for Interdependent Inputs.** We propose a simple yet efficient modification for parameterizing $\beta_c^{(v)}$ to extend the input-dependent Bayesian update for interdependent attributed nodes. The core idea is to first predict the feature class pseudo-counts $\boldsymbol{\beta}^{\text{ft},(v)}$ based on independent node features only, and then diffuse them to form the aggregated class pseudo-counts $\boldsymbol{\beta}^{\text{agg},(v)}$ based on neighborhood features. Hence, the feature class pseudo-counts $\boldsymbol{\beta}^{\text{ft},(v)}$ intuitively act as uncertainty estimates without network effects while the aggregated class pseudo-counts $\boldsymbol{\beta}^{\text{agg},(v)}$ intuitively act as uncertainty estimates with network effects.

To this end, GPN performs three main steps (see Fig. 2). **(1)** A (feature) encoder maps the features of $v$ onto a low-dimensional latent representation $\boldsymbol{z}$ i.e. $\boldsymbol{z}^{(v)} = f_\theta(\boldsymbol{x}^{(v)}) \in \mathbb{R}^H$. In practice, we use a simple MLP encoder in our experiments similarly to APPNP [48], **(2)** One conditional probability density per class $\mathbb{P}(\boldsymbol{z}^{(v)} \,|\, c; \boldsymbol{\phi})$ is used to compute $\beta_c^{\text{ft},(v)}$ i.e $\beta_c^{\text{ft},(v)} \propto \mathbb{P}(\boldsymbol{z}^{(v)} \,|\, c; \boldsymbol{\phi})$. Note that the the total feature evidence $\alpha_0^{\text{ft},(v)} = \sum_c \beta_c^{\text{ft},(v)}$ and the parameter $\bar{\boldsymbol{p}}^{\text{ft},(v)} = \boldsymbol{\beta}^{\text{ft},(v)}/\alpha_0^{\text{ft},(v)}$ are only based on node features and can be seen as epistemic and aleatoric uncertainty measures *without network effects*. In practice, we used radial normalizing flows for density estimation similarly to [14] and scaled the certainty $N$ budget w.r.t. the latent dimension $H$ similarly to [15]. **(3)** A Personalized Page Rank (PPR) message passing scheme is used to diffuse the feature class pseudo-counts $\beta_c^{\text{ft},(v)}$ and form the aggregated class pseudo-counts $\beta_c^{\text{agg},(v)}$ i.e.

$$\beta_c^{\text{agg},(v)} = \sum_{u \in \mathcal{V}} \Pi_{v,u}^{ppr} \beta_c^{\text{ft},(u)} \tag{3}$$

where $\Pi_{v,u}^{ppr}$ are the dense PPR scores implicitly reflecting the importance of node $u$ on $v$. We approximate the dense PPR scores using power iteration similarly to [48]. The aggregated pseudo-count $\beta_c^{\text{agg},(v)}$ is then used in the input-dependent Bayesian update (see Eq. 2). Remark that the scores $\Pi_{v,u}^{ppr}$ define a valid conditional distribution over all nodes associated to the PPR random walk (i.e. $\sum_u \Pi_{v,u}^{ppr} = 1$). It can be viewed as a soft neighborhood for $v$ accounting for all neighborhood hops through infinitely many message passing steps [48]. Hence, on one hand, the PPR scores define a probability distribution over nodes using the node edges only. On the other hand, the quantity $\mathbb{P}(\boldsymbol{z}^{(u)} \,|\, c; \boldsymbol{\phi})$ defines a probability distribution over nodes using the node features only. Therefore, we can equiv-

alently rewrite this step using probabilistic notations $\mathbb{P}(v \mid u) = \Pi^{ppr}_{v,u}$ and $\mathbb{P}(u \mid c) = \mathbb{P}(z^{(u)} \mid c; \phi)$:

$$\beta^{\text{agg},(v)}_c \propto \bar{\mathbb{P}}(v \mid c) = \sum_{u \in \mathcal{V}} \mathbb{P}(v \mid u)\,\mathbb{P}(u \mid c) \tag{4}$$

Interestingly, the quantity $\bar{\mathbb{P}}(v \mid c)$ defines a valid distribution which normalizes over all node features and accounts for the soft neighborhood (i.e. $\int ... \int \bar{\mathbb{P}}(v \mid c) dz^{(u_1)} ... dz^{(u_{|\mathcal{V}|})} = 1$). Hence, the message passing step is a simple but efficient method to transform the feature distributions of a single node into a joint distributions over the soft neighborhood features. Finally, the evidence $\alpha^{\text{agg},(v)}_0 = \sum_c \beta^{\text{agg},(v)}_c$ and the parameter $\boldsymbol{p}^{\text{agg},(v)} = \boldsymbol{\beta}^{\text{agg},(v)} / \alpha^{\text{agg},(v)}_0$ are based on neighborhood features and can be seen as epistemic and aleatoric uncertainty measures *with network effects*. Remark that, the sequential processing of the features (i.e. steps (1)+(2)) and network information (i.e. step (3)) in GPN is a key element to differentiate between the uncertainty without and with network effects and is a building block to provably obey the axioms.

GPN extends both APPNP [48] and PostNet [14] approaches. The key difference to APPNP is the density estimation modeling the epistemic uncertainty (i.e. steps (1)+(2)) and the input-dependent Bayesian update allowing to recover the prior prediction (i.e. Eq. 2). The key difference to PostNet is the PPR diffusion which accounts for dependence between nodes (step (3)).

**Optimization.** We follow [14] and train GPN by minimizing the following Bayesian loss with two terms i.e.:

$$\mathcal{L}^{(v)} = -\mathbb{E}_{\boldsymbol{p}^{(v)} \sim \mathbb{Q}^{post,(v)}} \left[ \log \mathbb{P}(y^{(v)} \mid \boldsymbol{p}^{(v)}) \right] - \lambda\,\mathbb{H}\left[ \mathbb{Q}^{post,(v)} \right] \tag{5}$$

where $\lambda$ is a regularization factor. It can be computed quickly in closed-form and provides theoretical guarantees for optimal solutions [14]. All parameters of GPN are trained jointly. Similarly to [15], we also observed that "warm-up" training for the normalizing flows is helpful.

### 3.3 Uncertainty Estimation Guarantees

In this section, we provide theoretical guarantees showing that GPN fulfills the three axioms under mild assumptions given the specific definitions of concepts of aleatoric/epistemic uncertainty and with/without network effects presented in Sec. 3.2. Throughout this section, we consider a GPN model parameterized with a (feature) encoder $f_\phi$ with piecewise ReLU activations, a PPR diffusion, and a density estimator $\mathbb{P}(z^{\text{ft},(v)} \mid \omega)$ with bounded derivatives. We present detailed proofs in appendix.

The first theorem shows that GPN follows Ax. 3.1 and guarantees that GPN achieves reasonable uncertainty estimation on extreme node features without network effects:

**Theorem 1.** *Lets consider a GPN model. Let $f_\phi(\boldsymbol{x}^{(v)}) = V^{(l)}\boldsymbol{x}^{(v)} + a^{(l)}$ be the piecewise affine representation of the ReLU network $f_\phi$ on the finite number of affine regions $Q^{(l)}$ [7]. Suppose that $V^{(l)}$ have independent rows, then for any node $v$ and almost any $\boldsymbol{x}^{(v)}$ we have $\mathbb{P}(f_\phi(\delta \cdot \boldsymbol{x}^{(v)}) \mid c; \phi) \underset{\delta \to \infty}{\to} 0$. Without network effects, it implies that $\beta^{ft,(v)}_c = \beta^{agg,(v)}_c \underset{\delta \to \infty}{\to} 0$.*

The proof relies on two main points: the equivalence of the GPN and PostNet architectures without network effects, and the uncertainty guarantees of PostNet far from training data similarly to [15]. It intuitively states that, without network effects, GPN predict small evidence (i.e. $\boldsymbol{\beta}^{\text{agg},(v)} \approx \boldsymbol{0}$) far from training features (i.e. $||\delta \cdot \boldsymbol{x}^{(v)}|| \to \infty$) and thus recover the prior prediction (i.e. $\boldsymbol{\alpha}^{\text{post},(v)} \approx \boldsymbol{\alpha}^{\text{prior}}$). Note that contrary to GPN, methods which do not account for node features (e.g. Label Propagation) or methods which only use ReLU activations [39] cannot validate Ax. 3.1. Further, methods which perform aggregation steps in early layers (e.g. GCN [46]) do not separate the processing of the feature and network information making unclear if they fulfill the Ax. 3.1 requirements.

The second theorem shows that GPN follows Ax. 3.2 and guarantees that a node $v$ becomes more epistemically certain if its neighbors are more epistemically certain:

**Theorem 2.** *Lets consider a GPN model. Then, given a node $v$, the aggregated feature evidence $\alpha^{agg,(v)}_0$ is increasing if the feature evidence $\alpha^{ft,(u)}_0$ of one of its neighbors $u \in \mathcal{N}(v)$ is increasing.*

The proof directly relies on Eq. 3. Intuitively, this theorem states that the epistemic uncertainty $u^{(v)}_{\text{epist}} = -\alpha^{\text{agg},(v)}_0$ of a node $v$ with network effects decreases if the epistemic uncertainty of the neighboring nodes without network effects decreases. Note that contrary to GPN, methods which do not model the epistemic uncertainty explicitly (e.g. GCN [46], GAT [92] or APPNP [48]) are not guaranteed to fulfil Ax. 3.2.

The third theorem shows that GPN follows Ax. 3.3. It guarantees that a node $v$ becomes more aleatorically uncertain if its neighbors are more aleatorically uncertain, or if a neighbor prediction disagrees more with the current node prediction:

**Theorem 3.** *Lets consider a GPN model. Lets denote $\bar{\boldsymbol{p}}^{agg,\,(v)} = \boldsymbol{\beta}^{agg,(v)}/\alpha_0^{agg,(v)}$ the diffused categorical prediction for node $v$ where $c^*$ is its winning class. Further, lets denote $\bar{\boldsymbol{p}}^{ft,\,(u)} = \boldsymbol{\beta}^{ft,(v)}/\alpha_0^{ft,(v)}$ the non-diffused categorical prediction for a node $u \in \mathcal{V}$. First, there exists normalized weights $\Pi_{v,u}'$ such that $\sum_{u \in \mathcal{V}} \Pi_{v,u} \, \mathbb{H}\left[\mathrm{Cat}(\bar{\boldsymbol{p}}^{ft,\,(u)})\right] \leq \mathbb{H}\left[\mathrm{Cat}(\bar{\boldsymbol{p}}^{agg,\,(v)})\right]$. Second, if for any node $u \in \mathcal{V}$ the probability of $\bar{\boldsymbol{p}}_{c^*}^{ft,\,(u)}$ decreases, then $\mathbb{H}\left[\mathrm{Cat}(\bar{\boldsymbol{p}}^{agg,\,(v)})\right]$ increases.*

The proof of the first part of the theorem is based on the entropy convexity. Intuitively, it states that the aleatoric uncertainty $u_{\mathrm{alea}}^{(v)} = \mathbb{H}\left[\mathrm{Cat}(\bar{\boldsymbol{p}}^{agg,\,(v)})\right]$ of a node $v$ with network effects is lower bounded by a weighted average of the aleatoric uncertainty without network effects of its soft neighborhood. The second part of the theorem intuitively states that if the prediction of a neighboring node $u$ without neighbor effects disagrees more with the current class prediction $c^*$ of the node $v$, then the aleatoric uncertainty $u_{\mathrm{alea}}^{(v)} = \mathbb{H}\left[\mathrm{Cat}(\bar{\boldsymbol{p}}^{agg,\,(v)})\right]$ with network effects becomes higher. Note that contrary to GPN, methods which do not use edges (e.g. PostNet [14]) cannot validate Ax. 3.3 and Ax. 3.2.

### 3.4 Limitations & Impact

**OOD data close to ID data.** While GPN is guaranteed to provide consistent uncertainty estimates for nodes with extreme OOD features, it does not guarantee any specific uncertainty estimation behavior for OOD data close to ID data. Note that there exist two possible desired behaviors for OOD close to ID data: being robust to small dataset shifts [78, 89] or detect near OOD data [98, 50, 13]. The duality of these two views makes unclear what would be the desired behavior even for i.i.d. data.

**Non-homophilic uncertainty.** Our approach assumes that connected nodes are likely to have similar uncertainty estimates as defined in Ax. 3.2 and Ax. 3.3. Contrary to [105], we do not tackle the problem of heterophilic graphs where two neighboring nodes might reasonably have different uncertainty estimates.

**Task-specific OOD.** Density estimation is shown to be inappropriate for OOD detection when acting directly on raw images [72, 17, 71] or on arbitrarily transformed space [54]. One of the reasons is that normalizing flows learn pixel correlations in images. This phenomena does not happen for tabular data with more semantic features [47]. First note that, similarly to tabular data, semantic node features are less likely to suffer from the same flaws. Second, following previous works [14, 15, 47, 69, 98], GPN mitigates this issue by using density estimation on a latent space which is low-dimensional and task-specific. Nonetheless, we emphasize that GPN provides predictive uncertainty estimates which depends on the considered task i.e. OOD data w.r.t. features which are not useful for the specific task are likely not to be encoded in the latent space, and thus not to be detected.

**Broader Impact.** The Assessment List for Trustworthy AI (ALTAI) [1] includes robustness, safety, and accountability. Uncertainty estimation is a key element to make AI systems follow these values. For example. an automated decision maker should know when it does not know. In this regard, GPN significantly improves the reliability of predictions on interdependent data under perturbations even though a user should not blindly rely on it. Further, ALTAI also mentions privacy and fairness. Therein, we raise awareness on the risk of using interconnected information which can amplify privacy or fairness violation in the presence of personal data.

## 4 Experiments

In this section, we provide an extensive evaluation set-up for uncertainty quantification for node classification. It compares **GPN** to 13 baselines on 8 datasets and consists in two task types. First, we evaluate the detection of OOD nodes with features perturbations and Left-Out classes. Second, we evaluate the robustness of accuracy, calibration and uncertainty metrics w.r.t. feature and edge shifts.

### 4.1 Set-up

**Ablation.** In the experiments, GPN uses a MLP as feature encoder, radial normalizing flows [82] for the density estimation and a certainty budget $N$ which scales with respect to the latent dimension [15]. We provide an ablation study covering aleatoric uncertainty through APPNP, feature-level

estimates through PostNet, diffusing resulting pseudo-counts after training, and GPN with diffusion of $\log(\beta_c^{\text{ft},(v)})$ instead of $\beta_c^{\text{ft},(v)}$ (see App. E.1). The complete GPN model outperforms the ablated models for uncertainty estimation. Further, we provide a hyper-parameter study covering for example different number of flow layers, latent dimensions, PPR teleport probabilities (see App. E.2)).

**Baselines.** We used 13 baselines covering a wide variety of models for semi-supervised node classification and uncertainty estimation. We show the results of 5 baselines in the main paper and the full results in appendix. It contains two standard GNNs (i.e. Vanilla GCN **VGCN** [46, 87] and **APPNP** [48]), one robust GNN (i.e. **RGCN** [104]), one dropout-based method for GNN (i.e. **DropEdge** [83]), two Graph Gaussian Processes methods (i.e. **GGP** [74] and **Matern-GGP** [12]), the Graph-based Kernel Dirichlet GCN method (i.e. **GKDE-GCN** [102]) and two parameter-less methods (i.e. **GKDE** [102] and Label Propagation **LP** see App.). Further, we also compared to direct adaptation of dropout (i.e. **VGCN-Dropout**[29]), ensemble (i.e. **VGCN-Ensemble** [52]), BNN (i.e. **VGCN-BNN** [9]) and energy-based models (i.e. **VGCN-Energy** [57]) to vanilla GCNs. All models are trained using the same number of layers and similar number of hidden dimensions. We used early stopping and report the used hyperparameters in appendix. The results are averaged over 10 initialization seeds per split. Further model details are given in appendix.

**Datasets.** We used 8 datasets with different properties summarized in appendix. We show the results of 3 datasets in the main paper and the full results in appendix. It contains common citation network datasets (i.e. **CoraML** [65, 32, 31, 85], **CiteSeer** [32, 31, 85], **PubMed** [73], **CoauthorPhysics** [87] **CoauthorCS** [87]) and co-purchase datasets (i.e. **AmazonPhotos** [64, 87], **AmazonComputers** [64, 87]). The results are averaged over 10 initialization splits with a train/val/test split of $5\%/15\%/80\%$ using stratified sampling. Further, we evaluate on the large **OGBN Arxiv** dataset with $169, 343$ nodes and $2, 315, 598$ edges [43, 94]. Further dataset details are given in the appendix.

## 4.2 Results

**OOD Detection.** In this section, we evaluate uncertainty estimation for OOD detection. To this end, we use the Area Under Receiving Operator Characteristics Curve (AUC-ROC) with aleatoric scores $u_{\text{alea}}^{(v)}$ (**Alea**) and epistemic scores $u_{\text{epist}}^{(v)}$ (**Epist**) similarly to [14, 102, 60, 63, 61, 57]. For GPN, we differentiate between epistemic uncertainty scores without network effects (**w/o Net.**) and with network effects (**w/ Net.**). Further, we report results with the Area Under the Precision-Recall Curve (AUC-PR) in appendix. The definition of OOD for nodes in the presence of feature and network information is more complex than for i.i.d. input features. Hence, we propose two types of OOD nodes: nodes with OOD feature perturbations and nodes from Left-Out classes. For feature perturbations, we compute the accuracy on the perturbed nodes (**OOD-Acc**) to evaluate if the model can correct anomalous features. For Left-Out classes, we compute the accuracy on the observed classes (**ID-Acc**). We report the short results in Tab. 1. We set a threshold of 64 GiB and 12 hours per training run. We also exclude methods which do not use attributes for detection of OOD feature perturbations.

*Feature perturbations:* These perturbations aim at isolating the contribution of the node feature information on the model predictions. To this end, we randomly select a subset of the nodes. For each single node $v$, we perturb individually its features using a Bernoulli or a Normal distribution (i.e. $\boldsymbol{x}^{(v)} \sim \text{Ber}(0.5)$ and $\boldsymbol{x}^{(v)} \sim \mathcal{N}(\boldsymbol{0}, \boldsymbol{1})$) keeping all other node features fixed. We then compare the uncertainty prediction on the perturbed and unperturbed node. On one hand, Bernoulli noise corresponds to small perturbations in the domain of discrete bag-of-words features. On the other hand, Normal noise corresponds to extreme perturbations out of the domain of discrete bag-of-words features. In practice, we expect out-of-domain perturbations to be easily detected [14]. First, we remark that uncertainty estimates of GPN based on features achieves an absolute improvement of at least $+15\%$ and $+29\%$ for Bernoulli and Normal perturbations over all baselines using network effects. This shows that GPN disentangles well the uncertainty without and with network effects. Second, we remark that all uncertainty estimates with network effects achieve poor results. This is expected if models can recover the correct prediction after aggregation steps. Specifically, we observe that GPN achieves an accuracy improvement between $+16\%$ and $+64\%$ for Normal perturbations on perturbed nodes compared to baselines. It stresses that GPN performs a consistent evidence aggregation from neighborhood to recover from anomalous features. Further, note that GPN is still capable to detect those perturbed nodes almost perfectly using feature uncertainty. These remarks aligns with Ax. 3.1.

*Left-Out classes:* Detection of Left-Out classes involves both feature and neighborhood information. In this case, we remove the Left-Out classes from the training set but keep them in the graph similarly to [102]. We observe that the uncertainty estimates with network effects of GPN achieves an absolute improvement between $+12\%$ and $+16\%$ compared to its uncertainty estimates without network effects. It highlights the benefit of incorporating network information for uncertainty predictions when OOD samples (i.e. samples from the Left-Out classes) are likely to be connected to each other. This remark aligns with Ax. 3.2. Further, GPN outperforms other baselines by $+2\%$ to $+22\%$ for LOC detection while maintaining a competitive accuracy on other classes.

*Misclassified samples:* In addition to the OOD scores, we also report the results for the detection of misclassified samples with aleatoric and epistemic uncertainty on several datasets and models in App. E.3 for the sake of completeness. GPN performs competitively with the baselines. Moreover, we observe that epistemic uncertainty is better for OOD detection and aleatoric uncertainty is better for misclassification detection as already observed e.g. in [102].

| | Model | ID-ACC | OOD-AUC-ROC
Leave-Out Classes | OOD-ACC | OOD-AUC-ROC
$x^{(v)} \sim \mathrm{Ber}(0.5)$ | OOD-ACC | OOD-AUC-ROC
$x^{(v)} \sim \mathcal{N}(0,1)$ |
|---|---|---|---|---|---|---|---|
| CoraML | Matern-GGP | 87.03 | 83.13 / 82.98 / *n.a.* | *n.a.* | *n.a.* | *n.a.* | *n.a.* |
| | VGCN-Dropout | 89.08 | 81.27 / 71.65 / *n.a.* | 77.76 | 62.06 / 50.38 / *n.a.* | 18.28 | 40.53 / 71.06 / *n.a.* |
| | VGCN-Energy | 89.66 | 81.70 / 83.15 / *n.a.* | 78.90 | 63.68 / 66.26 / *n.a.* | 18.37 | 9.34 / 0.32 / *n.a.* |
| | VGCN-Ensemble | **89.87** | 81.85 / 74.24 / *n.a.* | 78.00 | 63.58 / 56.81 / *n.a.* | 21.00 | 33.72 / 64.92 / *n.a.* |
| | GKDE-GCN | 89.33 | 82.23 / 82.09 / *n.a.* | 76.40 | 61.74 / 63.15 / *n.a.* | 16.86 | 40.03 / 1.42 / *n.a.* |
| | GPN | 88.51 | 83.25 / **86.28** / 80.95 | **80.98** | 57.99 / 55.23 / **89.47** | **81.53** | 55.96 / 56.51 / **100.00** |
| Amazon Photos | Matern-GGP | 88.65 | 87.26 / 86.75 / *n.a.* | *n.a.* | *n.a.* | *n.a.* | *n.a.* |
| | VGCN-Dropout | 94.04 | 80.90 / 70.11 / *n.a.* | 83.86 | 56.85 / 55.04 / *n.a.* | 22.29 | 49.11 / 66.74 / *n.a.* |
| | VGCN-Energy | 94.24 | 82.44 / 79.64 / *n.a.* | 83.91 | 57.91 / 59.07 / *n.a.* | 21.40 | 31.07 / 6.42 / *n.a.* |
| | VGCN-Ensemble | **94.28** | 82.72 / 88.53 / *n.a.* | 84.40 | 57.86 / 56.01 / *n.a.* | 20.30 | 44.14 / 69.01 / *n.a.* |
| | GKDE-GCN | 89.84 | 73.65 / 69.09 / *n.a.* | 73.17 | 57.01 / 58.00 / *n.a.* | 24.04 | 24.45 / 9.82 / *n.a.* |
| | GPN | 94.01 | 82.72 / **91.98** / 76.57 | **87.47** | 56.25 / 60.52 / **75.24** | **88.29** | 51.89 / 61.89 / **100.00** |
| OGBN Arxiv | Matern-GGP | *n.f.* | *n.f.* | *n.f.* | *n.f.* | *n.f.* | *n.f.* |
| | VGCN-Dropout | 75.47 | 65.35 / 64.24 / *n.a.* | 65.30 | 48.11 / 50.64 / *n.a.* | 49.90 | 60.10 / 62.87 / *n.a.* |
| | VGCN-Energy | 75.61 | 64.91 / 64.50 / *n.a.* | 65.70 | 46.16 / 48.54 / *n.a.* | 51.30 | 53.83 / 48.53 / *n.a.* |
| | VGCN-Ensemble | **76.12** | 65.93 / 70.77 / *n.a.* | **67.00** | 45.99 / 47.41 / *n.a.* | 49.00 | 59.94 / 66.44 / *n.a.* |
| | GKDE-GCN | 73.89 | 68.84 / 72.44 / *n.a.* | 65.20 | 50.98 / 51.31 / *n.a.* | 45.40 | 53.94 / 55.28 / *n.a.* |
| | GPN | 73.84 | 66.33 / **74.82** / 62.17 | 65.50 | 51.49 / 55.82 / **93.05** | **65.50** | 51.43 / 55.85 / **95.54** |

Table 1: LOC and Feature Perturbations: Accuracy is reported on ID nodes for LOC experiments and on OOD nodes for feature perturbation experiments. OOD-AUC-ROC scores are given as *[Alea w/ Net] / [Epist w/ Net] / [Epist w/o Net]*. $n.a.$ means either model or metric not applicable and $n.f.$ means not finished within our constraints.

**Attributed Graph Shifts.** In this section, we focus on evaluating the robustness of the accuracy, calibration and the evolution of the uncertainty estimation under node feature shifts and edges shifts. This aligns with [78] which aims at evaluating the reliability of uncertainty estimates under dataset shifts for i.i.d. inputs. Specifically, we evaluates the evolution of the accuracy, the ECE [70] calibration score, the epistemic and the aleatoric uncertainty measures.

*Feature shifts:* We perturbed the features of a fraction of the nodes using unit Gaussian perturbations. We report the short results in Fig. 3 and the full results in appendix. On one hand, we observe that GPN is significantly more robust to feature perturbations than all baselines. Indeed, the accuracy of GPN decreases by less than $5\%$ even when $80\%$ of the nodes are perturbed while the accuracy of other baselines decreases by more than $50\%$ when only $20\%$ of the nodes are perturbed. Similarly, we observed that GPN remains calibrated even when a high fraction of nodes are perturbed contrary to baselines. Hence, GPN intuitively discards uncertain features from perturbed nodes and only accounts for certain features from other nodes for more accurate predictions. On the other hand, we observe that, as desired, the average epistemic uncertainty of GPN consistently decreases when more nodes are perturbed. This remark aligns with Ax. 3.2. In contrast, baselines dangerously become more certain while achieving a poorer accuracy similarly to ReLU networks [39]. Hence GPN predictions are significantly more reliable than baselines under feature shifts.

*Edge shifts:* For edge shifts, we perturbed a fraction of edges at random. We report the results in appendix. As desired, we observe that the aleatoric uncertainty increases for all models including GPN. This aligns with Ax. 3.3 and the expectations that conflicting neighborhood should lead to more aleatorically uncertain predictions. Furthermore, the average epistemic uncertainty of GPN remains constant which is reasonable since the average evidence of a node's neighborhood remains constant.

**Qualitative Evaluation.** We show the abstracts of the CoraML papers achieving the highest and the lowest epistemic uncertainty without network effects in Tab. 2 and in the appendix. Interestingly, we

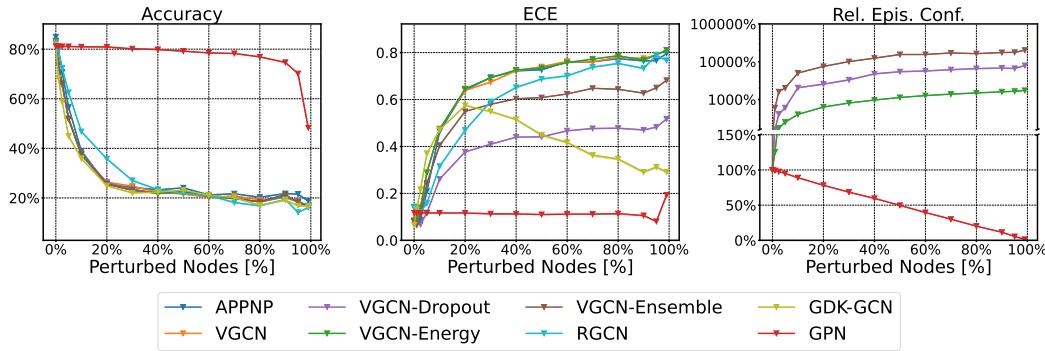

Figure 3: Accuracy, ECE, and average epistemic confidence under feature shifts for CoraML. We perturb features of different percentage of nodes using a Unit Gaussian noise.

observed that most uncertain papers corresponds to short and unconventional abstracts which can be seen as anomalous features. Furthermore, we also ranked the nodes w.r.t. to their epistemic uncertainty with network effects. In this case, we observed that $78/100$ nodes with the highest uncertainty do not belong to the largest connected component of the CoraML dataset. We propose additional uncertainty visualizations for GPN in App. E.6.

**Inference & training time.** We provide a comparison of inference and training times for most of the datasets and models under consideration in in App. E.7. GPN needs a single pass for uncertainty estimation but requires the additional evaluation of one normalizing flow per class compared to APPNP. Hence, GPN brings a small computational overhead for uncertainty estimation at inference time. Furthermore, GPN is usually converging relatively fast during training and does not require pre-computing kernel values. In contrast, GKDE-GCN [102] requires the computation of the underlying Graph Kernel with a complexity of $\mathcal{O}\left(N^2\right)$ where $N$ is the number of nodes in the graph. Finally, GPN is significantly more efficient than dropout or ensemble approaches as it does not require training or evaluating multiple models.

| | |
|---|---|
| IlliGAL Report No. 95006 July 1995 | Report of the 1996 Workshop on Reinforcement |
| Reihe FABEL-Report Status: extern Dokumentbezeichner: Org/Reports/nr-35 Erstellt am: 21.06.94 Korrigiert am: 28.05.95 ISSN 0942-413X | We tend to think of what we really know as what we can talk about, and disparage knowledge that we can't verbalize. [Dowling 1989, p. 252] |
| Keith Mathias and Darrell Whitley Technical Report CS-94-101 January 7, 1994 | Multigrid Q-Learning Charles W. Anderson and Stewart G. Crawford-Hines Technical Report CS-94-121 October 11, 1994 |
| Internal Report 97-01 | A Learning Result for Abstract |

Table 2: A selection of abstracts from CoraML which are assigned low feature evidences by GPN.

## 5 Conclusion

We introduce a well-grounded framework for uncertainty estimation on interdependent nodes. First, we propose explicit and motivated axioms describing desired properties for aleatoric and epistemic uncertainty in the absence or in the presence of network effects. Second, we propose GPN, a GNN for uncertainty estimation which provably follows our axioms. GPN performs a Bayesian update over the class predictions based on density estimation and diffusion. Third, we conduct extensive experiments to evaluate the uncertainty performances of a broad range of baselines for OOD detection and robustness against node feature or edge shifts. GPN outperforms all baselines in these experiments.

## Acknowledgments and Disclosure of Funding

This research was supported by the BMW AG, by the Helmholtz Association under the joint research school "Munich School for Data Science - MUDS", and by a grant from Software Campus through the German Federal Ministry of Education and Research.

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
