# A Proofs

**Lemma 1.** *[15] Let a model be parameterized with an encoder $f_\phi$ with piecewise ReLU activations, a decoder $g_\psi$ and the density estimator $\mathbb{P}(z \mid \omega)$. Let $f_\phi(x) = V^{(l)}x + a^{(l)}$ be the piecewise affine representation of the ReLU network $f_\phi$ on the finite number of affine regions $Q^{(l)}$ [7]. Suppose that $V^{(l)}$ have independent rows and the density function $\mathbb{P}(z \mid \omega)$ has bounded derivatives, then for almost any $x$ we have $\mathbb{P}(f_\phi(\delta \cdot x) \mid \omega) \underset{\delta \to \infty}{\to} 0$. i.e the evidence becomes small far from training data.*

**Theorem 1.** *Lets consider a GPN model. Let $f_\phi(x^{(v)}) = V^{(l)}x^{(v)} + a^{(l)}$ be the piecewise affine representation of the ReLU network $f_\phi$ on the finite number of affine regions $Q^{(l)}$ [7]. Suppose that $V^{(l)}$ have independent rows, then for any node $v$ and almost any $x^{(v)}$ we have $\mathbb{P}(f_\phi(\delta \cdot x^{(v)}) \mid c; \phi) \underset{\delta \to \infty}{\to} 0$. Without network effects, it implies that $\beta_c^{ft,(v)} = \beta_c^{agg,(v)} \underset{\delta \to \infty}{\to} 0$.*

*Proof.* First, remark that each normalizing flow density in GPN fulfills the conditions of Lem. 1. This means that $\mathbb{P}(f_\phi(\delta \cdot x^{(v)}) \mid c; \phi) \underset{\delta \to \infty}{\to} 0$ which implies $\beta_c^{ft,(v)} \underset{\delta \to \infty}{\to} 0$. Further note that in the absence of network effects, the PPR diffusion has no effect on the pseudo-counts i.e. $\beta_c^{ft,(v)} = \beta_c^{agg,(v)}$. $\qquad \square$

**Theorem 2.** *Lets consider a GPN model. Then, given a node $v$, the aggregated feature evidence $\alpha_0^{agg,(v)}$ is increasing if the feature evidence $\alpha_0^{ft,(u)}$ of one of its neighbor $u \in \mathcal{N}(v)$ is increasing.*

*Proof.* We recall first the definition of the aggregated and feature total evidence pseudo-count and the PPR diffusion step of GPN:

$$\alpha_0^{ft,(v)} = \sum_c \beta_c^{ft,(v)}$$

$$\alpha_0^{agg,(v)} = \sum_c \beta_c^{agg,(v)}$$

$$\beta_c^{agg,(v)} = \sum_{u \in \mathcal{V}} \Pi_{v,u}^{ppr} \beta_c^{ft,(u)}$$

We combine these three equations to show a closed-form relation between the aggregated and the feature evidence pseudo-count:

$$\begin{aligned}
\alpha_0^{agg,(v)} &= \sum_c \beta_c^{agg,(v)} \\
&= \sum_c \sum_{u \in \mathcal{V}} \Pi_{v,u}^{ppr} \beta_c^{ft,(u)} \\
&= \sum_{u \in \mathcal{V}} \Pi_{v,u}^{ppr} \sum_c \beta_c^{ft,(u)} \\
&= \sum_{u \in \mathcal{V}} \Pi_{v,u}^{ppr} \alpha_0^{ft,(u)}
\end{aligned}$$

This shows that the aggregated evidence is the diffused feature evidence with PPR. Hence, the aggregated evidence $\alpha_0^{agg,(v)}$ is a strictly increasing function w.r.t. to the feature evidence of each individual neighbor $\alpha_0^{ft,(u)}$ when $u \in \mathcal{N}(v)$. $\qquad \square$

**Theorem 3.** *Lets consider a GPN model. Lets denote $\bar{p}^{agg,(v)} = \beta^{agg,(v)}/\alpha_0^{agg,(v)}$ the diffused categorical prediction for node $v$ where $c^*$ is its winning class. Further, lets denote $\bar{p}^{ft,(u)} = \beta^{ft,(v)}/\alpha_0^{ft,(v)}$ the non-diffused categorical prediction for a node $u \in \mathcal{V}$. First, there exists normalized weights $\Pi'_{v,u}$ such that $\sum_{u \in \mathcal{V}} \Pi'_{v,u} \mathbb{H}\left[\mathrm{Cat}(\bar{p}^{ft,(u)})\right] \leq \mathbb{H}\left[\mathrm{Cat}(\bar{p}^{agg,(v)})\right]$. Second, if for any node $u \in \mathcal{V}$ the probability of $\bar{p}_{c^*}^{ft,(u)}$ decreases, then $\mathbb{H}\left[\mathrm{Cat}(\bar{p}^{agg,(v)})\right]$ increases.*

*Proof.* We first show the first part of the theorem. To this end, we use the relation between the aggregated and the feature evidence to derive a closed form relation between $\bar{p}^{\mathrm{agg},\,(v)}$ and $\bar{p}^{\mathrm{ft},\,(u)}$.

$$
\begin{aligned}
\bar{p}^{\mathrm{agg},\,(v)} &= \frac{\beta^{\mathrm{agg},(v)}}{\alpha_0^{\mathrm{agg},(v)}} \\
&= \frac{\sum_{u\in\mathcal{V}} \Pi_{v,u}^{ppr} \beta^{\mathrm{ft},(u)}}{\sum_{u'\in\mathcal{V}} \Pi_{v,u'}^{ppr} \alpha_0^{\mathrm{ft},(u')}} \\
&= \frac{\sum_{u\in\mathcal{V}} \Pi_{v,u}^{ppr} \alpha_0^{\mathrm{ft},(u)} \bar{p}^{\mathrm{ft},\,(u)}}{\sum_{u'\in\mathcal{V}} \Pi_{v,u'}^{ppr} \alpha_0^{\mathrm{ft},(u')}} \\
&= \sum_{u\in\mathcal{V}} \Pi'_{v,u} \bar{p}^{\mathrm{ft},\,(u)}
\end{aligned}
$$

where $\Pi'_{v,u} = \frac{\Pi_{v,u}^{ppr} \alpha_0^{\mathrm{ft},(u)}}{\sum_{u'\in\mathcal{V}} \Pi_{v,u'}^{ppr} \alpha_0^{\mathrm{ft},(u')}}$. Hence, the probability vector $\bar{p}^{\mathrm{agg},\,(v)}$ is a convex combination of the probability vectors $\bar{p}^{\mathrm{agg},\,(u)}$ of other nodes. Further, using the concavity of the entropy function, we obtain the results:

$$
\sum_{u\in\mathcal{V}} \Pi'_{v,u} \mathbb{H}\left[\mathrm{Cat}(\bar{p}^{\mathrm{ft},\,(u)})\right] \leq \mathbb{H}\left[\mathrm{Cat}(\bar{p}^{\mathrm{agg},\,(v)})\right]
$$

Second, we show the second part of the theorem. To this end, we suppose that, for a neighboring node $u \in \mathcal{N}(v)$, the probability of the winning class $c^*$ decreases and the probability of another class $c'$ increases i.e. $\bar{p}_{c^*}^{\mathrm{ft},\,(u)} - \epsilon$ and $\bar{p}_{c'}^{\mathrm{ft},\,(u)} + \epsilon$. We define the following univariate function:

$$
f(\epsilon) = \mathbb{H}\left[\mathrm{Cat}(\bar{p}_\epsilon^{\mathrm{agg},\,(v)})\right]
$$

where $\bar{p}_\epsilon^{\mathrm{agg},\,(v)}$ is the new aggregated probability vector of the node $v$ after the epsilon change of the probability vector for node $u$. Note that $\bar{p}_{\epsilon,c^*}^{\mathrm{agg},\,(v)} = \sum_{w\in\mathcal{V}} \Pi_{v,w}^{ppr} \bar{p}_{c^*}^{\mathrm{ft},\,(w)} - \Pi_{v,u}^{ppr}\epsilon$ and $\bar{p}_{\epsilon,c'}^{\mathrm{agg},\,(v)} = \sum_{w\in\mathcal{V}} \Pi_{v,w}^{ppr} \bar{p}_{c'}^{\mathrm{ft},\,(w)} + \Pi_{v,u}^{ppr}\epsilon$. We compute the derivative of $f(\epsilon)$:

$$
\begin{aligned}
\frac{\partial f(\epsilon)}{\partial \epsilon} &= \frac{\partial(\bar{p}_{\epsilon,c^*}^{\mathrm{agg},\,(v)} \log \bar{p}_{\epsilon,c^*}^{\mathrm{agg},\,(v)} + \bar{p}_{\epsilon,c'}^{\mathrm{agg},\,(v)} \log \bar{p}_{\epsilon,c'}^{\mathrm{agg},\,(v)})}{\partial \epsilon} \\
&= \log \frac{\sum_{w\in\mathcal{V}} \Pi_{v,w}^{ppr} \bar{p}_{c^*}^{\mathrm{ft},\,(w)} - \Pi_{v,u}^{ppr}\epsilon}{\sum_{w\in\mathcal{V}} \Pi_{v,w}^{ppr} \bar{p}_{c'}^{\mathrm{ft},\,(w)} + \Pi_{v,u}^{ppr}\epsilon}
\end{aligned}
$$

Hence, we note that as long as the class $c^*$ is winning (i.e $\bar{p}_{\epsilon,c^*}^{\mathrm{agg},\,(v)} = \sum_{w\in\mathcal{V}} \Pi_{v,w}^{ppr} \bar{p}_{c^*}^{\mathrm{ft},\,(w)} - \Pi_{v,u}^{ppr}\epsilon \geq \sum_{w\in\mathcal{V}} \Pi_{v,w}^{ppr} \bar{p}_{c'}^{\mathrm{ft},\,(w)} + \Pi_{v,u}^{ppr}\epsilon = \bar{p}_{\epsilon,c'}^{\mathrm{agg},\,(v)}$), the function $f(\epsilon)$ is increasing. It means that the entropy $\mathbb{H}\left[\mathrm{Cat}(\bar{p}^{\mathrm{agg},\,(v)})\right]$ increases when a neighboring node disagree more with the winning class $c^*$. In particular, note that the conclusion holds if the epsilon decrease of the winning class is compensated by the probability increase of $K$ different classes. It would correspond to composing $K$ decreasing functions $f_k(\epsilon_k)$ where $k \in \{0, ..., K\}$ and $\sum_k \epsilon_k = \epsilon$. $\qquad\square$

## B  Dataset Details

We consider the citation network datasets *CoraML* [65, 32, 31, 85, 10], *CiteSeer* [32, 31, 85], *PubMed* [73], *CoauthorPhysics* and *CoauthorCS* (based on the *Microsoft Academic Graph* from the *KDD Cup 2016* challenge) [87] as well as two co-purchase datasets, *AmazonPhotos* and *AmazonComputers* [64, 87]. Details are presented in the Table 3. For all those datasets, we consider a train/val/test split of $5/15/80$ using stratified sampling. For *CoraML*, this corresponds to the default split of 20 training samples per class with the difference of representing larger classes with more and smaller ones with less. We also note that this is significantly closer to the default split than approaches like [93, 44] introducing a $60/20/20$ split. For all those datasets, we average the results over individual predictions from 10 random splits together with 10 random model initializations per split, i.e. 100 runs for each dataset and model. Those dataset are part of *PyTorch-Geometric* and under a MIT license. Besides those datasets, we also report results for the large dataset *OGBN Arxiv* dataset [43] which is based on the *Microsoft Academic Graph* [94] with the public split based on publication years. Since this makes random splits unnecessary, we report results as averages over 10 runs. This dataset is also available under a MIT license.

|  | CoraML | CiteSeer | PubMed | Amazon Computers | Amazon Photos | Coauthor CS | Coauthor Physics | OGBN - Arxiv |
|---|---|---|---|---|---|---|---|---|
| vertices | 2,995 | 3,327 | 19,717 | 13,752 | 7,650 | 18,333 | 34,493 | 169,343 |
| edges | 16,316 | 9,104 | 88,648 | 491,722 | 238,162 | 163,788 | 495,924 | 2,315,598 |
| homophily | 78.86% | 73.55% | 80.24% | 77.22% | 82.72% | 80.81% | 93.15% | 65.42% |
| feature dimension | 2,879 | 3,703 | 500 | 767 | 745 | 6,805 | 8,415 | 128 |
| max words | 176 | 54 | 122 | 767 | 745 | 666 | 335 | *N/A* |
| mean words | 50.47 | 31.61 | 50.11 | 267.24 | 258.81 | 59.57 | 32.97 | *N/A* |
| median words | 49 | 32 | 50 | 204 | 193 | 45 | 27 | *N/A* |
| classes | 7 | 6 | 3 | 10 | 8 | 15 | 5 | 40 |
| left-out classes | 3 | 2 | 1 | 5 | 3 | 4 | 2 | 15 |
| fraction left-out | 44.91% | 33.18% | 39.94% | 29.52% | 40.46% | 40.72% | 18.18% | 39.11% |

Table 3: Dataset details summarizing the graph size, the homophily ratio (fraction of intra-class edges), the number of classes, and statistics on the Bag-Of-Word features when available. In particular, *OGBN-Arxiv* uses averaged skip-gram embeddings for the nodes' features and thus does not require bag-of-word-features. Further, we provide the number of left-out classes and the fraction of left-out nodes for the LOC experiments.

## C  Metrics

**ACC -** Accuracy is simply the fraction of predictions $\hat{y}^{(u)}$ that correspond to the ground-truth targets $y^{(u)}$ out of a set of all predictions of size $N$, i.e. $acc = \frac{1}{N} \sum_{u \in \mathcal{V}} 1_{y^{(u)} = \hat{y}^{(u)}}$.

**Brier -** The Brier Score is computed as $1/C \sum_v^N ||\boldsymbol{p}^{(v)} - \boldsymbol{y}^{(v)}||_2$ where $\boldsymbol{y}^{(v)}$ represent the one-hot encoded ground-truth label for a node $v$ while $\boldsymbol{p}^{(v)}$ is the predicted probability score.

**ECE -** The expected calibration error on the other hand requires binning the predicted probability scores into $M$ equally spaced bins with $conf(B_m)$ being the average probability score in the bin $m$. With $acc(B_m)$ being the average accuracy of predictions in bin $m$, we obtain the final metric as

$$ECE = \sum_m^M \frac{|B_m|}{n} |acc(B_m) - conf(B_m)| \tag{6}$$

**OOD -** Furthermore, we use **AUC-ROC** and **AUC-PR** scores for OOD-detection experiments. This problem is considered as a binary classification problem with the positive targets being the OOD-nodes and the negative targets being ID-data points. We use the same aleatoric and epistemic uncertainty measures used in [14]. For aleatoric uncertainty measures, we use $u_{\text{alea}}^{(v)} = -\max_c \bar{\boldsymbol{p}}_c^{(u)}$. For epistemic uncertainty, we use $u_{\text{epist}}^{(v)} = -\alpha_0^{(v)}$ for Dirichlet-based methods and $u_{\text{epist}}^{(v)} = \frac{1}{\sum_c \text{Var } \bar{\boldsymbol{p}}_c^{(u)}}$ for other methods. For Dirichlet-based methods and GPs, the corresponding quantities are predicted directly. For ensemble and dropout baselines, these quantities are computed based on the empirical mean and empirical variance.

Note that the *vacuity* uncertainty measure proposed in [102] and motivated from work on *subjective logic* is just the inverse transformation of $\alpha_0$ given by $u_{\text{vacuity}} = \frac{C}{\alpha_0}$. Hence, the AUC-ROC and

AUC-PR scores which evaluate the ranking of the examples lead to the *exact* same final scores using $u_{\text{vacuity}}$ or $u_{\text{epist}}$.

# D   Model Details

We follow [87, 49, 102] and baselines from the OGBN leaderboard[3] for the choice of the architectures. By default, we use a hidden dimension of $h = 64$ and $l = 2$ layers for parametric models on all datasets except for *OGBN Arxiv*. In this case, we use early stopping with a patience of $50$ and a maximum of $100,000$ epochs. For *OGBN Arxiv*, we use a hidden dimension of $h = 256$ with $l = 3$ layers and use batch-norm. In this case, we use early stopping with a patience of $200$, a maximum number of $1,000$ epochs and no weight-decay for all models. For Gaussian Processes, we implement our experiment pipelines and models in *PyTorch* [81] and rely on *PyTorch-Geometric* [26]. For all models, we use the Adam optimizer [45] with its default parameters and a learning rate of $0.01$. For further details, we provide the code in the supplementary material.

**GPN -** We use a similar backbone architecture as for APPNP [48].[4] We report all the used hyper-parameters in Tab. 4. Similarly to [15], we use a certainty budget $N$ which scales exponentially w.r.t. the latent dimension (i.e. $N_H = \sqrt{4\pi}^H$) and 5 warm-up epochs maximizing the log likelihood of the normalizing flows. Furthermore, we use $K = 10$ power-iterations steps to approximate PPR scores. We do not use weight decay for the Normalizing Flows. Those parameters have been obtained after conducting an ablation and a hyper-parameter study on the *CoraML* and *OGBN Arxiv* datasets (see Sec.E.1 and Sec.E.2). Finally, we recall for completeness the closed-form of the Bayesian loss introduced in [14] when $\mathbb{Q}^{post,(v)} = \text{Dir}(\boldsymbol{\alpha}^{(v)})$ and $\mathbb{P}(y^{(v)} \mid \boldsymbol{p}^{(v)}) = \text{Cat}(y^{(v)} \mid \boldsymbol{p}^{(v)})$:

$$\mathcal{L}^{(v)} = -\mathbb{E}_{\boldsymbol{p}^{(v)} \sim \mathbb{Q}^{post,(v)}} \left[ \log \mathbb{P}(y^{(v)} \mid \boldsymbol{p}^{(v)}) \right] - \lambda \, \mathbb{H}\left[ \mathbb{Q}^{post,(v)} \right] \tag{7}$$

The expected likelihood term is equal to:

$$\mathbb{E}_{\boldsymbol{p} \sim \text{Dir}(\boldsymbol{\alpha})}[\log \text{Cat}(y \mid \boldsymbol{p})] = \psi(\alpha_y) - \psi(\alpha_0) \tag{8}$$

The entropy term is equal to:

$$\mathbb{H}[\text{Dir}(\boldsymbol{\alpha})] = \log B(\boldsymbol{\alpha}) + (\alpha_0 - C)\psi(\alpha_0) - \sum_c (\alpha_c - 1)\psi(\alpha_c) \tag{9}$$

| | $H$ | $L$ | $n_{layers}$ | $n_{radial}$ | $ACT$ | $p_{drop}$ | $\tau$ | budget | $\lambda$ | weight decay |
|---|---|---|---|---|---|---|---|---|---|---|
| **(1)** | 64 | 10 | 2 | 10 | ReLU | 0.5 | 0.2 | $N_H \cdot C$ | $1.0e - 05$ | 0.0005 |
| **(2)** | 64 | 16 | 2 | 10 | ReLU | 0.5 | 0.1 | $N_H$ | $1.0e - 3$ | 0.001 |
| **(3)** | 256 | 16 | 3 | 10 | ReLU + BN | 0.25 | 0.2 | $N_H$ | $1.0e - 3$ | 0.0 |

Table 4: GPN hyperparameters used in our experiments. (1) is used for *Amazon Photos* and *Amazon Computers* datasets, (3) is used for *OGBN-Arxiv* dataset and (2) is used for all other datasets. For all those settings, we furthermore use $K = 10$ power-iterations steps, 5 warmup epochs for the Normalizing Flows, and no weight decay for the normalizing flows.

**GKDE -** We adopt the Graph Kernel Dirichlet Estimate from [102] as a standalone and parameterless baseline. With $d_{v,u}$ being the shortest path between nodes $v$ and $u$ and the Gaussian transformation $g(d_{v,u}) = 1/\sigma\sqrt{2\pi} \exp\left(-d_{v,u}^2/2\sigma^2\right)$, a Dirichlet estimate is obtained in the following way

$$\boldsymbol{\alpha}^{(v)} = 1 + \boldsymbol{e}^{(v)} \quad \text{with} \quad \boldsymbol{e}^{(v)} = \sum_{u \in \mathcal{T}} \boldsymbol{h}(y^{(u)}, d_{v,u}) \quad \text{and} \quad h_c(y^{(u)}, d_{v,u}) = \begin{cases} 0 & y^{(u)} \neq c \\ g(d_{v,u}) & y^{(u)} = c \end{cases} \tag{10}$$

Similar to [99], we use $\sigma = 1$. We would also like to point out that the computation of this kernel requires extracting the shorted distance of each node to each labeled node $u \in \mathcal{T}$. Larger datasets like *OGBN-Arxiv* come with larger sets of labeled data with the size $|\mathcal{T}|$ having a same magnitude

---

[3]https://ogb.stanford.edu/docs/leader_nodeprop/#ogbn-arxiv
[4]Code available at https://www.daml.in.tum.de/graph-postnet

as the number of nodes in the graph, i.e. $|\mathcal{V}|$. This approach therefore scales quadratically with the number of nodes in the graph and therefore does not generalize well to larger datasets.

**LP -** Following the idea of the GKDE baseline, we propose similar Dirichlet estimates by relying on Label Propagation which achieve strong results in Left-Out classes experiments. GKDE extracts Dirichlet evidence scores by relying on node distances. We propose taking the density of labeled nodes in neighborhoods instead. To this end, we define one initial conditional density per class $\rho_0(u \mid c)$ and diffuse them with Personalized Page Rank i.e.

$$\rho_0(u \mid c) = \begin{cases} 0 & u \in \mathcal{U} \\ \frac{1}{|\mathcal{L}_c|} \cdot \delta_{y^{(u)},c} & u \in \mathcal{L} \end{cases} \to \rho(v \mid c) = \sum_u \Pi_{v,u}^{ppr} \cdot \rho_0(u \mid c) \tag{11}$$

where $\mathcal{L}_c$ is the set of labeled nodes for class $c$. The diffused density $\rho(v \mid c)$ is still a valid density i.e. $\sum_c \sum_u \rho(u \mid c) = \sum_c \sum_u \rho_0(u \mid c) = 1$. Finally, we use this diffused conditional densities to obtain Dirichlet evidence scores in a similar fashion to the GKDE kernel [102], i.e. $\alpha_c^{(v)} = 1.0 + \rho(v \mid c)$. The diffusion is performed with power-iteration similar to APPNP [48]. We use a teleport probability of $\tau = 0.1$ and $K = 10$ power iteration steps.

**Gaussian Processes -** We use the official implementations for **MaternGGP** [12] and the re-implementation[5] for **GGP** [74]. The re-implementation transfers the official implementation to Tensorflow 2.0 [2] which we wrapped in our Pytorch pipeline. Since those approaches do not scale well to large real-world datasets, we restrict to a single random initialization. GGP only finished the experiments on CoraML and CiteSeer. Similarly, MaternGGP did not finished the experiments on OGBN Arxiv. For recall, we set the memory an time limits to 64 GiB and 12 hours per run. For comparison, note that all GNN-based models require significantly less memory and finished *all* runs in a couple of hours.

**GKDE-SGCN -** We use the hyper-parameters suggested in the original paper [102]. We set the regularization factor $\lambda$ to $0.001$. This factor determines the weight of the Graph-Kernel-Dirichlet-Estimate which is key for OOD detection in graphs. Note that we did not use different factors for OOD experiments and classification experiments contrary to [102] since it leads to the leakage of task information. Indeed, the clean accuracy is significantly higher for $\lambda = 0.001$ compared to $\lambda = 0.1$. For *OGBN Arxiv*, we did not use teacher training as it harmed the performance. In this case, we used a dropout probability of $p = 0.5$ and $\lambda = 0.0001$ after a small grid-search with the overall architecture following the initial remarks above.

**APPNP -** We follow [49] and use an architecture that is comparable to other GNN approaches. We use ReLU activations, dropout with $p = 0.5$, no dropout on the adjacency matrix, a teleport probability of $\tau = 0.1$ and $K = 10$ power iteration steps. We also use a weight decay of $\lambda = 0.0001$.

**VGCN -** We use ReLU activations, dropout with $p = 0.8$, and a weight decay of $\lambda = 0.0001$. For the larger dataset *OGBN Arxiv*, we use a dropout of $p = 0.5$. **DropEdge** is similar to the Vanilla GCN model with an additional dropout on the edges with a dropout probability of $p = 0.5$ on both features and edges. For evaluation of dropout models, i.e. **DropEdge** and **VGCN-Dropout**, we use $S = 10$ Monte-Carlo samples having shown a reasonable estimate with more samples not leading to a visible improvement. For ensembles, i.e. **VGCN-Ensemble**, we use an ensemble of models of 10 different random initializations. For **VGCN-Energy**, we follow [57] and use a temperature of $T = 1.0$.

**VGCN-BNN -** We follow *Bayes by Backprop* [9] and adopt a Bayesian GNN with uncertain weights. We use a hidden dimension of $h = 32$. This is equivalent to $h = 64$ for other models as each weight is represented by one mean parameter $\mu$ and one variance parameter $\Sigma$. We use 10 bayesian samples in our experiments. We follow the grid search suggested in the original paper [9]. We finally adopt $\pi = 0.75$, $\sigma_1 = 1.0$, and $\sigma_2 = 1.0e - 6$. Since this models assumes uncertain weights, we do not apply any weight decay during training. Note that we do not report results for the larger dataset *OGBN Arxiv* for this baseline.

**RGCN -** We follow [104] for the hyper-parameter selection. We use a hidden size of $h = 32$. Again this is equivalent to $h = 64$ since RGCN models a mean parameter $\mu$ and a variance parameter $\Sigma$ per layer. We further use dropout $p = 0.5$ on the features, $\gamma = 1$, $\beta_{KL} = 5.04 - 4$ and $\beta_{reg} = 5.0e - 4$. As the latter is already a weight regularization in the loss, we do not apply weight decay.

---

[5]https://github.com/FelixOpolka/GGP-TF2

# E Additional Experiments

## E.1 Additional Experiments - Ablation Study

In this section, we evaluate the contribution of each component of GPN. To this end, we use **PostNet** which first trains the feature encoder and the normalizing flows without diffusion and **PostNet-Diff** which diffuses the ablation-counts only at test time. Further, we also compare to **APPNP** [48] which does not model the epistemic uncertainty with density estimation and **GPN-LOG** which diffuse the parameters $\log(\beta_c^{\mathrm{ft},(v)})$ instead of $\beta_c^{\mathrm{ft},(v)}$. We observed that training with diffusion is beneficial for all metrics. Further, we noted that diffusing $\log(\beta_c^{\mathrm{ft},(v)})$ improves accuracy and calibration to the cost of a lower Left-Out classes detection scores. Similarly, APPNP also showed better accuracy when diffusing logits instead of softmax outputs in the original paper [48]. Finally, APPNP achieves significantly worse results for all OOD detection tasks showing the benefit of modelling the epistemic uncertainty.

| Model | ACC | ECE | Ber-**FT-AUC-ROC** | $\mathcal{N}$-**FT-AUC-ROC** | LOC-ID-ACC | LOC-EPIS-AUC-ROC |
|---|---|---|---|---|---|---|
| APPNP | $82.18 \pm 0.08$ | $8.46 \pm 0.09$ | $60.51 \pm 0.06$ | $16.32 \pm 0.31$ | $88.71 \pm 0.03$ | $84.75 \pm 0.06$ |
| PostNet | $52.24 \pm 0.90$ | $8.22 \pm 1.39$ | $83.09 \pm 4.41$ | $100.00 \pm 0.00$ | $69.37 \pm 0.50$ | $70.14 \pm 0.93$ |
| PostNet-Diff | $77.10 \pm 0.58$ | $30.35 \pm 0.86$ | $83.09 \pm 4.41$ | $100.00 \pm 0.00$ | $86.18 \pm 0.65$ | $78.45 \pm 0.80$ |
| GPN | $81.57 \pm 0.18$ | $12.77 \pm 1.11$ | $84.69 \pm 4.60$ | $100.00 \pm 0.00$ | $89.17 \pm 0.39$ | $87.34 \pm 0.76$ |
| GPN-LOG-$\beta$ | $82.90 \pm 0.26$ | $8.77 \pm 0.33$ | $91.24 \pm 2.67$ | $100.00 \pm 0.00$ | $91.17 \pm 0.59$ | $70.88 \pm 1.74$ |

Table 5: Accuracy, Calibration, and OOD-detection results for the ablation study on CoraML for the validation split. All models use epistemic uncertainty measures except APPNP which uses an aleatoric measure.

## E.2 Additional Experiments - Hyper-parameter study

Besides the previously mentioned ablation study, we also performed a study on the influences of hyperparameters. We show findings for the *CoraML* dataset averaging runs with 3 different random dataset splits and 2 different random initializations. We analyzed the influence of the latent dimension, the number of radial layers, the teleport probability, the certainty budget, weight decay, and entropy regularization.

## E.3 Additional Experiments - Misclassification

Work like [102] found that measures of aleatoric uncertainty are well suited for detecting inputs which are not classified correctly. Epistemic measures of uncertainty are found to be better suited for detecting OOD samples. Orthogonal work like [35, 78] uses calibration to determine how reliable predictions are. We adopt this point of view while reporting results for the task of OOD detection using epistemic measures. We also present aleatoric measures as a reference because simple baselines without any intrinsic uncertainty estimates solely can quantify uncertainty through simple aleatoric uncertainty measures. To facilitate an easy comparison to work like [102], we also present results for misclassification experiments in Tab. 10 and in Tab. 11. As in [99], we can observe that aleatoric uncertainty mostly is better for detecting misclassified samples for all models than epistemic uncertainty. GPN achieves a similar performance in this task while also showing that for some datasets measures not accounting for network effects can detect misclassified samples better than measures which account for network effects.

| Latent Dim | ACC | ECE | Ber-**FT-AUC-ROC** | $\mathcal{N}$-**FT-AUC-ROC** | LOC-ID-ACC | LOC-Epis-AUC-ROC |
|---|---|---|---|---|---|---|
| 6 | $79.18 \pm 0.16$ | $10.50 \pm 0.56$ | $81.75 \pm 4.74$ | $100.00 \pm 0.00$ | $90.17 \pm 0.41$ | $88.65 \pm 0.58$ |
| 10 | $79.14 \pm 0.44$ | $10.96 \pm 0.23$ | $77.92 \pm 4.44$ | $100.00 \pm 0.00$ | $89.39 \pm 0.58$ | $87.39 \pm 0.43$ |
| 16 | $81.57 \pm 0.18$ | $12.77 \pm 1.11$ | $84.69 \pm 4.60$ | $100.00 \pm 0.00$ | $89.17 \pm 0.39$ | $87.34 \pm 0.76$ |

Table 6: Accuracy, Calibration, and OOD-detection results of GPN on the CoraML dataset with latent dimensions in $[6, 10, 16]$.

| Radial Layers | ACC | ECE | Ber-**FT-AUC-ROC** | $\mathcal{N}$-**FT-AUC-ROC** | LOC-ID-ACC | LOC-Epis-AUC-ROC |
|---|---|---|---|---|---|---|
| 6 | $80.04 \pm 0.85$ | $11.26 \pm 0.48$ | $84.26 \pm 2.38$ | $100.00 \pm 0.00$ | $90.74 \pm 0.32$ | $86.62 \pm 0.25$ |
| 10 | $81.57 \pm 0.18$ | $12.77 \pm 1.11$ | $84.69 \pm 4.60$ | $100.00 \pm 0.00$ | $89.17 \pm 0.39$ | $87.34 \pm 0.76$ |
| 16 | $75.88 \pm 0.95$ | $8.77 \pm 0.32$ | $94.06 \pm 1.37$ | $100.00 \pm 0.00$ | $89.39 \pm 0.20$ | $84.14 \pm 0.12$ |

Table 7: Accuracy, Calibration, and OOD-detection results of GPN on the CoraML dataset with the number of radial layers in $[6, 10, 16]$.

| $\tau_{\text{teleport}}$ | ACC | ECE | Ber-**FT-AUC-ROC** | $\mathcal{N}$-**FT-AUC-ROC** | LOC-ID-ACC | LOC-Epis-AUC-ROC |
|---|---|---|---|---|---|---|
| 0.05 | $80.39 \pm 0.33$ | $11.98 \pm 0.53$ | $85.31 \pm 2.79$ | $100.00 \pm 0.00$ | $89.32 \pm 0.41$ | $86.63 \pm 0.65$ |
| 0.1 | $81.57 \pm 0.18$ | $12.77 \pm 1.11$ | $84.69 \pm 4.60$ | $100.00 \pm 0.00$ | $89.17 \pm 0.39$ | $87.34 \pm 0.76$ |
| 0.2 | $80.00 \pm 0.30$ | $8.27 \pm 0.59$ | $86.65 \pm 1.46$ | $100.00 \pm 0.00$ | $88.89 \pm 0.31$ | $87.30 \pm 0.30$ |

Table 8: Accuracy, Calibration, and OOD-detection results of GPN on the CoraML dataset with a teleport probability $\tau$ in $[0.05, 0.1, 0.2]$.

## E.4 Additional Experiments - OOD Detection

In this section, we provide additional results for the OOD detection experiments for the 8 datasets and the 13 baselines. We show the results for feature perturbations experiments using AUC-ROC and AUC-APR scores in Tab. 12 and Tab. 13. We show results for clean accuracy and calibration on the unperturbed graphs, and Left-Out classes using AUC-ROC and AUC-APR scores in Tab. 14 and Tab. 15. Similarly to Tab. 1, we observe that GPN achieves the best results for the detection of feature perturbations with uncertainty without network effects by a significant margin while still maintaining a high accuracy. Further, GPN also performs favourably on Left-Out classes experiments using the uncertainty measures with network effects. All these observations show that GPN disentangles well between uncertainty without and with network effects while being robust against feature perturbations.

## E.5 Additional Experiments - Attributed Graph Shifts

In this section, we provide additional results for the attributed graph shifts experiments. We show the results of the feature shifts and the results of the edge shifts in Figures 5 to 12. The feature shifts include Bernoulli and Normal shifts (i.e. $\boldsymbol{x}^{(v)} \sim \text{Ber}(0.5)$ and $\boldsymbol{x}^{(v)} \sim \mathcal{N}(\boldsymbol{0}, \boldsymbol{1})$) with up to 99% of the nodes being perturbed. The edge shifts include randomly moved edges and DICE attacks [95] where we perturb up to 99% of the edges in the graph. The results are consistent with the observations made in Sec. 4. For feature shifts, we observe that GPN is more robust to feature perturbations than competitors for accuracy and calibration similarly to results Tab. 3. GPN is particularly robust against unit Gaussians perturbations. Further, as desired, GPN is more epistemically uncertain when features of a larger fraction of nodes are perturbed. For edge shifts, all models including GPN become more aleatorically uncertain. This aligns with Ax. 3.3. Furthermore, the average epistemic uncertainty of GPN remains constant which is reasonable since the average evidence of a node's neighborhood remains constant. We observed that the exponential activation in the last layer of GKDE-GCN leads to numerical instabilities under perturbations.

| Budget | ACC | ECE | Ber-**FT-AUC-ROC** | $\mathcal{N}$-**FT-AUC-ROC** | LOC-ID-ACC | LOC-Epis-AUC-ROC |
|---|---|---|---|---|---|---|
| $N_H$ | $81.57 \pm 0.18$ | $12.77 \pm 1.11$ | $84.69 \pm 4.60$ | $100.00 \pm 0.00$ | $89.17 \pm 0.39$ | $87.34 \pm 0.76$ |
| $N_H \cdot C$ | $79.76 \pm 0.93$ | $11.24 \pm 0.89$ | $82.82 \pm 5.68$ | $100.00 \pm 0.00$ | $88.68 \pm 0.75$ | $86.39 \pm 1.27$ |

Table 9: Accuracy, Calibration, and OOD-detection results of GPN on the CoraML dataset with a budget that scales exponentially w.r.t. the latent dimension and a budget that scales exponentially w.r.t. the latent dimension and the number of classes.

| Dataset | Model | Alea w/ Net | Epis w/ Net | Epis w/o Net | Alea w/ Net | Epis w/ Net | Epis w/o Net |
|---|---|---|---|---|---|---|---|
| | | misclassification AUC-ROC | | | misclassification AUC-PR | | |
| CoraML | APPNP | $\mathbf{83.64 \pm 0.08}$ | *n.a.* | *n.a.* | $48.39 \pm 0.19$ | *n.a.* | *n.a.* |
| | VGCN | $81.02 \pm 0.07$ | *n.a.* | *n.a.* | $48.30 \pm 0.23$ | *n.a.* | *n.a.* |
| | VGCN-Dropout | $81.42 \pm 0.06$ | $65.52 \pm 0.28$ | *n.a.* | $49.34 \pm 0.17$ | $26.11 \pm 0.21$ | *n.a.* |
| | VGCN-Energy | $81.02 \pm 0.07$ | *n.a.* | *n.a.* | $48.30 \pm 0.23$ | *n.a.* | *n.a.* |
| | VGCN-Ensemble | $81.12 \pm 0.17$ | $72.62 \pm 0.19$ | *n.a.* | $49.16 \pm 0.59$ | $31.88 \pm 0.29$ | *n.a.* |
| | VGCN-BNN | $80.64 \pm 0.10$ | $65.40 \pm 0.48$ | *n.a.* | $47.49 \pm 0.26$ | $26.70 \pm 0.33$ | *n.a.* |
| | GKDE-GCN | $80.80 \pm 0.14$ | $76.83 \pm 0.17$ | *n.a.* | $\mathbf{49.61 \pm 0.47}$ | $45.87 \pm 0.48$ | *n.a.* |
| | GPN | $81.19 \pm 0.13$ | $78.10 \pm 0.26$ | $77.46 \pm 0.24$ | $49.51 \pm 0.26$ | $44.42 \pm 0.32$ | $43.31 \pm 0.41$ |
| CiteSeer | APPNP | $73.55 \pm 0.08$ | *n.a.* | *n.a.* | $51.70 \pm 0.15$ | *n.a.* | *n.a.* |
| | VGCN | $74.64 \pm 0.09$ | *n.a.* | *n.a.* | $54.32 \pm 0.18$ | *n.a.* | *n.a.* |
| | VGCN-Dropout | $74.81 \pm 0.11$ | $64.09 \pm 0.16$ | *n.a.* | $55.12 \pm 0.23$ | $39.41 \pm 0.19$ | *n.a.* |
| | VGCN-Energy | $74.64 \pm 0.09$ | *n.a.* | *n.a.* | $54.32 \pm 0.18$ | *n.a.* | *n.a.* |
| | VGCN-Ensemble | $74.42 \pm 0.26$ | $68.15 \pm 0.23$ | *n.a.* | $53.28 \pm 0.33$ | $43.30 \pm 0.30$ | *n.a.* |
| | VGCN-BNN | $73.28 \pm 0.11$ | $61.68 \pm 0.29$ | *n.a.* | $54.62 \pm 0.21$ | $37.99 \pm 0.24$ | *n.a.* |
| | GKDE-GCN | $75.45 \pm 0.11$ | $73.83 \pm 0.12$ | *n.a.* | $54.78 \pm 0.19$ | $53.57 \pm 0.20$ | *n.a.* |
| | GPN | $\mathbf{75.89 \pm 0.15}$ | $74.16 \pm 0.17$ | $72.50 \pm 0.12$ | $\mathbf{60.78 \pm 0.32}$ | $59.32 \pm 0.40$ | $52.10 \pm 0.18$ |
| PubMed | APPNP | $80.98 \pm 0.02$ | *n.a.* | *n.a.* | $37.79 \pm 0.08$ | *n.a.* | *n.a.* |
| | VGCN | $81.16 \pm 0.02$ | *n.a.* | *n.a.* | $38.24 \pm 0.08$ | *n.a.* | *n.a.* |
| | VGCN-Dropout | $80.46 \pm 0.04$ | $72.69 \pm 0.09$ | *n.a.* | $38.63 \pm 0.11$ | $25.90 \pm 0.09$ | *n.a.* |
| | VGCN-Energy | $81.16 \pm 0.02$ | *n.a.* | *n.a.* | $38.24 \pm 0.08$ | *n.a.* | *n.a.* |
| | VGCN-Ensemble | $\mathbf{81.31 \pm 0.06}$ | $79.30 \pm 0.08$ | *n.a.* | $38.06 \pm 0.33$ | $31.73 \pm 0.19$ | *n.a.* |
| | VGCN-BNN | $79.96 \pm 0.07$ | $72.63 \pm 0.45$ | *n.a.* | $39.31 \pm 0.08$ | $27.59 \pm 0.39$ | *n.a.* |
| | GKDE-GCN | $80.95 \pm 0.09$ | $73.99 \pm 0.27$ | *n.a.* | $39.64 \pm 0.10$ | $33.19 \pm 0.14$ | *n.a.* |
| | GPN | $80.46 \pm 0.13$ | $75.38 \pm 0.25$ | $80.48 \pm 0.13$ | $40.74 \pm 0.19$ | $35.11 \pm 0.11$ | $\mathbf{51.12 \pm 0.52}$ |

Table 10: Misclassification Scores on the clean graphs given as AUC-ROC and AUC-PR scores. AUC-ROC and AUC-APR scores are given as *[Alea w/ Net] / [Epist w/ Net] / [Epist w/o Net]*. *n.a.* means either model or metric not applicable. Bold numbers indicate the best model across all the uncertainty metrics for each dataset.

## E.6 Additional Experiments - Qualitative Evaluation

In this section, we provide additional qualitative evaluations. Therefore, we present the abstracts of the most epistemically uncertain papers and the most epistemically certain papers in CoraML in Tab. 16 and Tab. 17. Most epistemically certain papers shows significantly more reasonable abstracts compared to most epistemically uncertain papers.

Additionally, we provide visualizations of the latent space of GPN on the clean CoraML graph in Fig. 13, and on the CoraML graph where $10\%$ of the nodes are perturbed with the unit Gaussian perturbation in Fig. 13, and on Left-Out classes experiments in Fig. 15. We used T-SNE projections for 2D visualizations. We observed that the latent representations correlate with the class assignment. Further, GPN is capable to separate nodes with perturbed features in the latent space. The nodes with perturbed features are assigned high uncertainty without network effects but low uncertainty with network effects. This stresses the capacity of GPN to recover from feature perturbations.

## E.7 Additional Experiments - Inference & Training Time

We provide a comparison of inference times for most of the datasets and models under consideration in Tab. 18 and a comparison of training times in Tab. 19. GPN needs a single pass for uncertainty estimation but requires the additional evaluation of one normalizing flow per class compared to APPNP. Hence, GPN brings a small computational overhead for uncertainty estimation during inference but is significantly faster than ensemble or dropout approaches. Furthermore, GPN is usually converging relatively fast during training and does not require a pre-computing kernel values. In contrast, GKDE-GCN requires the computation of the underlying Graph Kernel with a complexity of $\mathcal{O}(N^2)$ where $N$ is the number of nodes in the graph (see Sec. D). Finally, GPN is significantly more efficient than dropout or ensemble approaches as it does not require training or evaluation of multiple models.

| Dataset | Model | Alea w/ Net | Epis w/ Net | Epis w/o Net | Alea w/ Net | Epis w/ Net | Epis w/o Net |
|---|---|---|---|---|---|---|---|
| | | | misclassification AUC-ROC | | | misclassification AUC-PR | |
| Amazon Computers | APPNP | $79.75 \pm 0.03$ | n.a. | n.a. | $45.10 \pm 0.11$ | n.a. | n.a. |
| | VGCN | $82.08 \pm 0.03$ | n.a. | n.a. | $45.52 \pm 0.12$ | n.a. | n.a. |
| | VGCN-Dropout | $\mathbf{82.70 \pm 0.04}$ | $72.02 \pm 0.11$ | n.a. | $47.19 \pm 0.17$ | $31.45 \pm 0.14$ | n.a. |
| | VGCN-Energy | $82.08 \pm 0.03$ | n.a. | n.a. | $45.52 \pm 0.12$ | n.a. | n.a. |
| | VGCN-Ensemble | $82.05 \pm 0.06$ | $67.62 \pm 0.44$ | n.a. | $45.40 \pm 0.24$ | $26.33 \pm 0.37$ | n.a. |
| | VGCN-BNN | $82.15 \pm 0.17$ | $48.65 \pm 0.82$ | n.a. | $\mathbf{69.61 \pm 0.42}$ | $30.87 \pm 0.43$ | n.a. |
| | GKDE-GCN | $79.66 \pm 0.19$ | $73.66 \pm 0.15$ | n.a. | $63.26 \pm 0.57$ | $56.93 \pm 0.54$ | n.a. |
| | GPN | $82.20 \pm 0.10$ | $77.58 \pm 0.16$ | $70.06 \pm 0.19$ | $47.93 \pm 0.42$ | $41.80 \pm 0.44$ | $28.70 \pm 0.51$ |
| Amazon Photos | APPNP | $85.74 \pm 0.06$ | n.a. | n.a. | $37.00 \pm 0.20$ | n.a. | n.a. |
| | VGCN | $87.94 \pm 0.05$ | n.a. | n.a. | $48.35 \pm 0.19$ | n.a. | n.a. |
| | VGCN-Dropout | $\mathbf{89.52 \pm 0.05}$ | $78.46 \pm 0.10$ | n.a. | $50.27 \pm 0.19$ | $23.08 \pm 0.12$ | n.a. |
| | VGCN-Energy | $87.94 \pm 0.05$ | n.a. | n.a. | $48.35 \pm 0.19$ | n.a. | n.a. |
| | VGCN-Ensemble | $88.08 \pm 0.16$ | $76.05 \pm 0.54$ | n.a. | $49.40 \pm 0.67$ | $24.13 \pm 0.67$ | n.a. |
| | VGCN-BNN | $84.17 \pm 0.19$ | $51.84 \pm 0.66$ | n.a. | $51.05 \pm 0.64$ | $18.69 \pm 0.55$ | n.a. |
| | GKDE-GCN | $84.11 \pm 0.29$ | $75.07 \pm 0.50$ | n.a. | $\mathbf{54.35 \pm 0.58}$ | $45.43 \pm 0.68$ | n.a. |
| | GPN | $87.21 \pm 0.10$ | $83.38 \pm 0.30$ | $79.93 \pm 0.23$ | $46.32 \pm 0.39$ | $37.07 \pm 0.60$ | $29.90 \pm 0.71$ |
| Coauthor CS | APPNP | $89.92 \pm 0.03$ | n.a. | n.a. | $37.98 \pm 0.12$ | n.a. | n.a. |
| | VGCN | $89.46 \pm 0.03$ | n.a. | n.a. | $38.86 \pm 0.10$ | n.a. | n.a. |
| | VGCN-Dropout | $88.46 \pm 0.04$ | $79.03 \pm 0.08$ | n.a. | $38.06 \pm 0.12$ | $17.98 \pm 0.09$ | n.a. |
| | VGCN-Energy | $89.46 \pm 0.03$ | n.a. | n.a. | $38.86 \pm 0.10$ | n.a. | n.a. |
| | VGCN-Ensemble | $89.51 \pm 0.08$ | $86.61 \pm 0.09$ | n.a. | $38.74 \pm 0.23$ | $30.60 \pm 0.38$ | n.a. |
| | VGCN-BNN | $89.01 \pm 0.06$ | $78.40 \pm 0.23$ | n.a. | $38.17 \pm 0.17$ | $19.06 \pm 0.20$ | n.a. |
| | GKDE-GCN | $89.24 \pm 0.05$ | $80.98 \pm 0.13$ | n.a. | $39.30 \pm 0.27$ | $30.52 \pm 0.25$ | n.a. |
| | GPN | $85.72 \pm 0.15$ | $81.56 \pm 0.29$ | $\mathbf{94.41 \pm 0.11}$ | $46.12 \pm 0.32$ | $38.98 \pm 0.28$ | $\mathbf{77.26 \pm 0.45}$ |
| Coauthor Physics | APPNP | $93.27 \pm 0.02$ | n.a. | n.a. | $38.14 \pm 0.09$ | n.a. | n.a. |
| | VGCN | $92.86 \pm 0.02$ | n.a. | n.a. | $37.19 \pm 0.10$ | n.a. | n.a. |
| | VGCN-Dropout | $92.28 \pm 0.03$ | $89.85 \pm 0.04$ | n.a. | $35.47 \pm 0.11$ | $23.70 \pm 0.08$ | n.a. |
| | VGCN-Energy | $92.86 \pm 0.02$ | n.a. | n.a. | $37.19 \pm 0.10$ | n.a. | n.a. |
| | VGCN-Ensemble | $92.95 \pm 0.07$ | $91.92 \pm 0.07$ | n.a. | $37.96 \pm 0.28$ | $28.44 \pm 0.21$ | n.a. |
| | VGCN-BNN | $92.44 \pm 0.09$ | $89.03 \pm 0.27$ | n.a. | $36.79 \pm 0.21$ | $25.11 \pm 0.49$ | n.a. |
| | GKDE-GCN | $92.77 \pm 0.02$ | $86.12 \pm 0.06$ | n.a. | $37.08 \pm 0.11$ | $25.13 \pm 0.10$ | n.a. |
| | GPN | $91.14 \pm 0.04$ | $89.63 \pm 0.07$ | $\mathbf{93.89 \pm 0.05}$ | $41.43 \pm 0.13$ | $35.64 \pm 0.16$ | $\mathbf{59.03 \pm 0.28}$ |
| OGBN Arxiv | APPNP | $77.55 \pm 0.05$ | n.a. | n.a. | $54.57 \pm 0.14$ | n.a. | n.a. |
| | VGCN | $77.89 \pm 0.05$ | n.a. | n.a. | $54.87 \pm 0.12$ | n.a. | n.a. |
| | VGCN-Dropout | $78.11 \pm 0.05$ | $71.74 \pm 0.14$ | n.a. | $55.40 \pm 0.13$ | $43.43 \pm 0.18$ | n.a. |
| | VGCN-Energy | $77.89 \pm 0.05$ | n.a. | n.a. | $54.87 \pm 0.12$ | n.a. | n.a. |
| | VGCN-Ensemble | $\mathbf{78.14}$ | $71.48$ | n.a. | $53.95$ | $42.87$ | n.a. |
| | GKDE-GCN | $77.47 \pm 0.33$ | $77.55 \pm 0.33$ | n.a. | $\mathbf{61.62 \pm 1.00}$ | $62.33 \pm 1.00$ | n.a. |
| | GPN | $75.44 \pm 0.19$ | $72.71 \pm 0.28$ | $61.45 \pm 0.49$ | $55.64 \pm 0.37$ | $52.99 \pm 0.49$ | $39.37 \pm 0.42$ |

Table 11: Misclassification Scores on the clean graphs given as AUC-ROC and AUC-PR scores. AUC-ROC and AUC-APR scores are given as *[Alea w/ Net] / [Epist w/ Net] / [Epist w/o Net]*. *n.a.* means either model or metric not applicable. Bold numbers indicate the best model across all the uncertainty metrics for each dataset.

# F   Axioms Diagram

We provide a larger version of Figure 1 to visualize the distinction between aleatoric and epistemic uncertainty and the distinction between uncertainty without and with network effects in Fig. 16. These two distinctions are used in the axioms in Sec. 3.1.

| | Model | $x^{(v)} \sim \text{Ber}(0.5)$ | | | | $x^{(v)} \sim \mathcal{N}(0,1)$ | | | |
| | | OOD-ACC | OOD-AUC-ROC | | | OOD-ACC | OOD-AUC-ROC | | |
| | | | Alea w/ Net | Epist w/ Net | Epist w/o Net | | Alea w/ Net | Epist w/ Net | Epist w/o Net |
|---|---|---|---|---|---|---|---|---|---|
| CoraML | APPNP | $80.85 \pm 0.09$ | $64.41 \pm 0.02$ | n.a. | n.a. | $17.99 \pm 0.36$ | $7.98 \pm 0.13$ | n.a. | n.a. |
| | VGCN | $78.90 \pm 0.09$ | $63.68 \pm 0.03$ | n.a. | n.a. | $18.37 \pm 0.31$ | $9.34 \pm 0.13$ | n.a. | n.a. |
| | RGCN | $79.78 \pm 0.16$ | $70.30 \pm 0.05$ | n.a. | n.a. | $33.37 \pm 0.35$ | $32.13 \pm 0.28$ | n.a. | n.a. |
| | VGCN-Dropout | $77.76 \pm 0.15$ | $62.06 \pm 0.06$ | $50.38 \pm 0.12$ | n.a. | $18.28 \pm 0.35$ | $40.53 \pm 0.25$ | $71.06 \pm 0.29$ | n.a. |
| | DropEdge | $77.40 \pm 0.14$ | $63.10 \pm 0.04$ | $52.84 \pm 0.10$ | n.a. | $16.60 \pm 0.26$ | $23.10 \pm 0.29$ | $46.82 \pm 0.41$ | n.a. |
| | VGCN-Energy | $78.90 \pm 0.09$ | $63.68 \pm 0.04$ | $66.26 \pm 0.04$ | n.a. | $18.37 \pm 0.31$ | $9.34 \pm 0.13$ | $0.32 \pm 0.03$ | n.a. |
| | VGCN-Ensemble | $78.00 \pm 0.00$ | $63.58 \pm 0.00$ | $56.81 \pm 0.03$ | n.a. | $21.00 \pm 0.00$ | $33.72 \pm 0.02$ | $64.92 \pm 0.08$ | n.a. |
| | VGCN-BNN | $77.01 \pm 0.16$ | $64.74 \pm 0.07$ | $62.45 \pm 0.37$ | n.a. | $18.79 \pm 0.31$ | $34.85 \pm 0.50$ | $67.43 \pm 0.71$ | n.a. |
| | GKDE-GCN | $76.40 \pm 0.33$ | $61.74 \pm 0.05$ | $63.15 \pm 0.10$ | n.a. | $16.86 \pm 0.35$ | $40.03 \pm 0.46$ | $1.42 \pm 0.15$ | n.a. |
| | GPN | $\mathbf{80.98} \pm 0.22$ | $57.99 \pm 0.12$ | $55.23 \pm 0.16$ | $\mathbf{89.47} \pm 0.86$ | $\mathbf{81.53} \pm 0.23$ | $55.96 \pm 0.08$ | $56.51 \pm 0.21$ | $\mathbf{100.00} \pm 0.00$ |
| CiteSeer | APPNP | $\mathbf{73.14} \pm 0.12$ | $65.43 \pm 0.02$ | n.a. | n.a. | $20.13 \pm 0.22$ | $4.78 \pm 0.11$ | n.a. | n.a. |
| | VGCN | $71.30 \pm 0.13$ | $65.27 \pm 0.03$ | n.a. | n.a. | $17.55 \pm 0.36$ | $5.48 \pm 0.11$ | n.a. | n.a. |
| | RGCN | $72.29 \pm 0.09$ | $71.99 \pm 0.04$ | n.a. | n.a. | $28.15 \pm 0.40$ | $23.28 \pm 0.41$ | n.a. | n.a. |
| | VGCN-Dropout | $69.80 \pm 0.19$ | $63.47 \pm 0.09$ | $51.82 \pm 0.12$ | n.a. | $19.60 \pm 0.28$ | $31.79 \pm 0.27$ | $72.62 \pm 0.34$ | n.a. |
| | DropEdge | $72.00 \pm 0.23$ | $65.00 \pm 0.05$ | $54.71 \pm 0.16$ | n.a. | $18.00 \pm 0.47$ | $17.80 \pm 0.25$ | $44.78 \pm 0.52$ | n.a. |
| | VGCN-Energy | $71.30 \pm 0.13$ | $65.27 \pm 0.03$ | $68.16 \pm 0.06$ | n.a. | $17.55 \pm 0.36$ | $5.48 \pm 0.11$ | $0.03 \pm 0.01$ | n.a. |
| | VGCN-Ensemble | $72.00 \pm 0.00$ | $65.20 \pm 0.00$ | $51.81 \pm 0.01$ | n.a. | $18.00 \pm 0.00$ | $21.22 \pm 0.01$ | $52.80 \pm 0.02$ | n.a. |
| | VGCN-BNN | $70.38 \pm 0.15$ | $65.52 \pm 0.09$ | $49.33 \pm 0.70$ | n.a. | $16.27 \pm 0.33$ | $23.24 \pm 0.86$ | $60.07 \pm 1.69$ | n.a. |
| | GKDE-GCN | $72.75 \pm 0.18$ | $66.70 \pm 0.09$ | $67.29 \pm 0.06$ | n.a. | $18.79 \pm 0.33$ | $35.46 \pm 0.71$ | $0.21 \pm 0.04$ | n.a. |
| | GPN | $65.00 \pm 0.43$ | $59.47 \pm 0.16$ | $55.95 \pm 0.18$ | $\mathbf{80.06} \pm 1.18$ | $\mathbf{66.70} \pm 0.23$ | $51.65 \pm 0.16$ | $65.58 \pm 0.26$ | $\mathbf{100.00} \pm 0.00$ |
| PubMed | APPNP | $82.80 \pm 0.10$ | $62.22 \pm 0.02$ | n.a. | n.a. | $40.38 \pm 0.22$ | $5.41 \pm 0.22$ | n.a. | n.a. |
| | VGCN | $82.49 \pm 0.10$ | $62.16 \pm 0.06$ | n.a. | n.a. | $37.80 \pm 0.40$ | $6.54 \pm 0.23$ | n.a. | n.a. |
| | RGCN | $\mathbf{83.75} \pm 0.12$ | $64.87 \pm 0.05$ | n.a. | n.a. | $47.82 \pm 0.36$ | $29.60 \pm 0.34$ | n.a. | n.a. |
| | VGCN-Dropout | $82.26 \pm 0.06$ | $60.39 \pm 0.10$ | $51.80 \pm 0.14$ | n.a. | $37.79 \pm 0.45$ | $23.86 \pm 0.35$ | $38.16 \pm 0.53$ | n.a. |
| | DropEdge | $82.70 \pm 0.12$ | $62.21 \pm 0.10$ | $55.48 \pm 0.18$ | n.a. | $36.36 \pm 0.47$ | $13.32 \pm 0.35$ | $21.68 \pm 0.53$ | n.a. |
| | VGCN-Energy | $82.49 \pm 0.10$ | $62.16 \pm 0.06$ | $65.38 \pm 0.07$ | n.a. | $37.80 \pm 0.40$ | $6.54 \pm 0.23$ | $2.97 \pm 0.10$ | n.a. |
| | VGCN-Ensemble | $82.00 \pm 0.00$ | $62.42 \pm 0.00$ | $60.34 \pm 0.10$ | n.a. | $39.10 \pm 0.10$ | $11.74 \pm 0.03$ | $18.79 \pm 0.04$ | n.a. |
| | VGCN-BNN | $82.30 \pm 0.14$ | $62.36 \pm 0.15$ | $59.35 \pm 0.99$ | n.a. | $37.56 \pm 0.54$ | $12.74 \pm 0.34$ | $27.56 \pm 0.64$ | n.a. |
| | GKDE-GCN | $82.54 \pm 0.11$ | $60.62 \pm 0.11$ | $63.00 \pm 0.17$ | n.a. | $37.77 \pm 0.48$ | $24.07 \pm 0.43$ | $3.43 \pm 0.13$ | n.a. |
| | GPN | $81.54 \pm 0.39$ | $57.05 \pm 0.08$ | $58.87 \pm 0.14$ | $\mathbf{84.07} \pm 0.55$ | $\mathbf{81.73} \pm 0.34$ | $53.43 \pm 0.04$ | $60.94 \pm 0.13$ | $\mathbf{100.00} \pm 0.00$ |
| Amazon Computers | APPNP | $75.00 \pm 0.09$ | $67.83 \pm 0.02$ | n.a. | n.a. | $18.25 \pm 0.46$ | $5.94 \pm 0.11$ | n.a. | n.a. |
| | VGCN | $81.54 \pm 0.08$ | $58.03 \pm 0.02$ | n.a. | n.a. | $20.38 \pm 0.29$ | $24.56 \pm 0.33$ | n.a. | n.a. |
| | RGCN | $61.39 \pm 0.39$ | $57.92 \pm 0.04$ | n.a. | n.a. | $39.60 \pm 0.45$ | $33.60 \pm 0.35$ | n.a. | n.a. |
| | VGCN-Dropout | $\mathbf{81.79} \pm 0.10$ | $57.15 \pm 0.04$ | $55.52 \pm 0.08$ | n.a. | $21.52 \pm 0.36$ | $40.32 \pm 0.29$ | $66.21 \pm 0.33$ | n.a. |
| | DropEdge | $81.20 \pm 0.08$ | $57.88 \pm 0.02$ | $55.51 \pm 0.07$ | n.a. | $21.75 \pm 0.44$ | $33.10 \pm 0.26$ | $57.47 \pm 0.37$ | n.a. |
| | VGCN-Energy | $81.54 \pm 0.08$ | $58.03 \pm 0.02$ | $58.66 \pm 0.03$ | n.a. | $20.38 \pm 0.29$ | $24.56 \pm 0.33$ | $4.82 \pm 0.09$ | n.a. |
| | VGCN-Ensemble | $81.00 \pm 0.00$ | $58.22 \pm 0.01$ | $53.24 \pm 0.12$ | n.a. | $23.60 \pm 0.16$ | $28.97 \pm 0.05$ | $66.42 \pm 0.19$ | n.a. |
| | VGCN-BNN | $61.25 \pm 0.17$ | $56.01 \pm 0.11$ | $51.16 \pm 0.83$ | n.a. | $28.54 \pm 0.47$ | $21.72 \pm 0.19$ | $56.41 \pm 0.60$ | n.a. |
| | GKDE-GCN | $59.83 \pm 0.73$ | $56.38 \pm 0.12$ | $55.91 \pm 0.05$ | n.a. | $28.46 \pm 0.36$ | $26.48 \pm 0.45$ | $16.10 \pm 0.53$ | n.a. |
| | GPN | $79.70 \pm 0.46$ | $61.21 \pm 0.11$ | $61.07 \pm 0.11$ | $\mathbf{86.15} \pm 0.28$ | $\mathbf{79.87} \pm 0.46$ | $60.42 \pm 0.12$ | $61.56 \pm 0.12$ | $\mathbf{100.00} \pm 0.00$ |
| Amazon Photos | APPNP | $\mathbf{88.12} \pm 0.10$ | $65.02 \pm 0.03$ | n.a. | n.a. | $19.37 \pm 0.45$ | $8.42 \pm 0.29$ | n.a. | n.a. |
| | VGCN | $83.91 \pm 0.08$ | $57.91 \pm 0.02$ | n.a. | n.a. | $21.40 \pm 0.49$ | $31.07 \pm 0.34$ | n.a. | n.a. |
| | RGCN | $79.50 \pm 0.72$ | $57.22 \pm 0.04$ | n.a. | n.a. | $42.38 \pm 0.40$ | $32.02 \pm 0.31$ | n.a. | n.a. |
| | VGCN-Dropout | $83.86 \pm 0.18$ | $56.85 \pm 0.04$ | $55.04 \pm 0.08$ | n.a. | $22.29 \pm 0.55$ | $49.11 \pm 0.31$ | $66.74 \pm 0.35$ | n.a. |
| | DropEdge | $85.69 \pm 0.15$ | $57.32 \pm 0.04$ | $55.31 \pm 0.07$ | n.a. | $22.90 \pm 0.43$ | $39.14 \pm 0.20$ | $56.18 \pm 0.21$ | n.a. |
| | VGCN-Energy | $83.91 \pm 0.08$ | $57.91 \pm 0.02$ | $59.07 \pm 0.02$ | n.a. | $21.40 \pm 0.49$ | $31.07 \pm 0.34$ | $6.42 \pm 0.07$ | n.a. |
| | VGCN-Ensemble | $84.40 \pm 0.16$ | $57.86 \pm 0.01$ | $56.01 \pm 0.19$ | n.a. | $20.30 \pm 0.21$ | $44.14 \pm 0.05$ | $69.01 \pm 0.14$ | n.a. |
| | VGCN-BNN | $82.00 \pm 0.19$ | $56.78 \pm 0.09$ | $49.21 \pm 0.58$ | n.a. | $25.84 \pm 0.46$ | $23.16 \pm 0.37$ | $59.31 \pm 0.73$ | n.a. |
| | GKDE-GCN | $73.17 \pm 0.94$ | $57.01 \pm 0.10$ | $58.00 \pm 0.05$ | n.a. | $24.04 \pm 0.42$ | $24.45 \pm 0.62$ | $9.82 \pm 0.36$ | n.a. |
| | GPN | $87.47 \pm 0.20$ | $56.25 \pm 0.16$ | $60.52 \pm 0.18$ | $\mathbf{75.24} \pm 0.63$ | $\mathbf{88.29} \pm 0.20$ | $51.89 \pm 0.09$ | $61.89 \pm 0.18$ | $\mathbf{100.00} \pm 0.00$ |
| Coauthor CS | APPNP | $89.28 \pm 0.07$ | $72.01 \pm 0.02$ | n.a. | n.a. | $12.58 \pm 0.33$ | $23.09 \pm 0.31$ | n.a. | n.a. |
| | VGCN | $89.33 \pm 0.05$ | $67.65 \pm 0.02$ | n.a. | n.a. | $13.29 \pm 0.22$ | $30.13 \pm 0.32$ | n.a. | n.a. |
| | RGCN | $\mathbf{90.50} \pm 0.06$ | $71.13 \pm 0.02$ | n.a. | n.a. | $41.67 \pm 0.35$ | $52.81 \pm 0.21$ | n.a. | n.a. |
| | VGCN-Dropout | $88.96 \pm 0.09$ | $65.91 \pm 0.05$ | $60.56 \pm 0.07$ | n.a. | $13.31 \pm 0.32$ | $67.56 \pm 0.26$ | $85.81 \pm 0.22$ | n.a. |
| | DropEdge | $89.44 \pm 0.07$ | $67.94 \pm 0.01$ | $63.68 \pm 0.06$ | n.a. | $11.65 \pm 0.28$ | $49.77 \pm 0.33$ | $70.31 \pm 0.49$ | n.a. |
| | VGCN-Energy | $89.33 \pm 0.05$ | $67.65 \pm 0.02$ | $70.14 \pm 0.02$ | n.a. | $13.29 \pm 0.22$ | $30.13 \pm 0.32$ | $0.89 \pm 0.06$ | n.a. |
| | VGCN-Ensemble | $89.00 \pm 0.00$ | $67.64 \pm 0.01$ | $64.41 \pm 0.08$ | n.a. | $11.00 \pm 0.00$ | $60.89 \pm 0.11$ | $85.09 \pm 0.32$ | n.a. |
| | VGCN-BNN | $88.16 \pm 0.10$ | $67.09 \pm 0.12$ | $59.69 \pm 0.35$ | n.a. | $12.48 \pm 0.25$ | $67.00 \pm 0.54$ | $87.27 \pm 0.47$ | n.a. |
| | GKDE-GCN | $88.14 \pm 0.14$ | $67.69 \pm 0.07$ | $70.08 \pm 0.14$ | n.a. | $9.71 \pm 0.29$ | $32.67 \pm 0.51$ | $0.23 \pm 0.03$ | n.a. |
| | GPN | $83.99 \pm 0.31$ | $57.66 \pm 0.10$ | $62.08 \pm 0.18$ | $\mathbf{97.84} \pm 0.23$ | $\mathbf{83.96} \pm 0.31$ | $57.04 \pm 0.09$ | $62.39 \pm 0.18$ | $\mathbf{100.00} \pm 0.00$ |
| Coauthor Physics | APPNP | $96.16 \pm 0.08$ | $67.63 \pm 0.02$ | n.a. | n.a. | $28.71 \pm 0.40$ | $24.97 \pm 0.20$ | n.a. | n.a. |
| | VGCN | $96.00 \pm 0.00$ | $60.30 \pm 0.02$ | n.a. | n.a. | $33.26 \pm 0.67$ | $40.19 \pm 0.42$ | n.a. | n.a. |
| | RGCN | $94.69 \pm 0.07$ | $65.84 \pm 0.03$ | n.a. | n.a. | $58.56 \pm 0.36$ | $52.91 \pm 0.32$ | n.a. | n.a. |
| | VGCN-Dropout | $95.90 \pm 0.05$ | $58.97 \pm 0.03$ | $57.64 \pm 0.05$ | n.a. | $32.52 \pm 0.60$ | $55.07 \pm 0.48$ | $61.85 \pm 0.50$ | n.a. |
| | DropEdge | $95.90 \pm 0.03$ | $60.40 \pm 0.04$ | $59.09 \pm 0.05$ | n.a. | $30.53 \pm 0.58$ | $43.30 \pm 0.43$ | $51.07 \pm 0.44$ | n.a. |
| | VGCN-Energy | $96.00 \pm 0.00$ | $60.30 \pm 0.02$ | $61.59 \pm 0.02$ | n.a. | $33.26 \pm 0.67$ | $40.19 \pm 0.42$ | $11.45 \pm 0.22$ | n.a. |
| | VGCN-Ensemble | $96.00 \pm 0.00$ | $60.29 \pm 0.00$ | $59.05 \pm 0.01$ | n.a. | $31.70 \pm 0.21$ | $52.08 \pm 0.10$ | $68.48 \pm 0.10$ | n.a. |
| | VGCN-BNN | $95.65 \pm 0.07$ | $60.99 \pm 0.19$ | $56.95 \pm 0.35$ | n.a. | $32.95 \pm 0.60$ | $62.53 \pm 0.75$ | $71.96 \pm 0.73$ | n.a. |
| | GKDE-GCN | $\mathbf{96.61} \pm 0.05$ | $60.46 \pm 0.01$ | $60.99 \pm 0.07$ | n.a. | $28.84 \pm 0.31$ | $29.12 \pm 0.33$ | $2.46 \pm 0.10$ | n.a. |
| | GPN | $92.70 \pm 0.11$ | $59.92 \pm 0.06$ | $58.62 \pm 0.18$ | $\mathbf{99.15} \pm 0.05$ | $\mathbf{92.70} \pm 0.11$ | $58.66 \pm 0.08$ | $59.00 \pm 0.18$ | $\mathbf{100.00} \pm 0.00$ |
| OGBN Arxiv | APPNP | $63.50 \pm 0.95$ | $62.51 \pm 0.51$ | n.a. | n.a. | $51.10 \pm 1.12$ | $59.92 \pm 0.64$ | n.a. | n.a. |
| | VGCN | $65.70 \pm 0.47$ | $46.16 \pm 0.13$ | n.a. | n.a. | $51.30 \pm 0.75$ | $53.83 \pm 0.59$ | n.a. | n.a. |
| | VGCN-Dropout | $65.30 \pm 0.70$ | $48.11 \pm 0.23$ | $50.64 \pm 0.20$ | n.a. | $49.90 \pm 0.77$ | $60.10 \pm 0.68$ | $62.87 \pm 0.29$ | n.a. |
| | VGCN-Energy | $65.70 \pm 0.47$ | $46.16 \pm 0.13$ | $48.54 \pm 0.20$ | n.a. | $51.30 \pm 0.75$ | $53.83 \pm 0.59$ | $48.53 \pm 0.48$ | n.a. |
| | VGCN-Ensemble | $\mathbf{67.00}$ | $45.99$ | $47.41$ | n.a. | $49.00$ | $59.94$ | $66.44$ | n.a. |
| | GKDE-GCN | $65.20 \pm 0.49$ | $50.98 \pm 0.23$ | $51.31 \pm 0.22$ | n.a. | $45.40 \pm 0.62$ | $53.94 \pm 1.41$ | $55.28 \pm 1.69$ | n.a. |
| | GPN | $65.50 \pm 0.70$ | $51.49 \pm 0.37$ | $55.82 \pm 0.30$ | $\mathbf{93.05} \pm 3.44$ | $\mathbf{65.50} \pm 0.70$ | $51.43 \pm 0.32$ | $55.85 \pm 0.30$ | $\mathbf{95.54} \pm 0.89$ |

Table 12: Accuracy and OOD detection scores on Bernoulli and unit Gaussian feature perturbations using AUC-ROC. OOD-AUC-ROC scores are given as *[Alea w/ Net] / [Epist w/ Net] / [Epist w/o Net]*. *n.a.* means either model or metric not applicable. Bold numbers indicate best results for Accuracy and OOD detection.

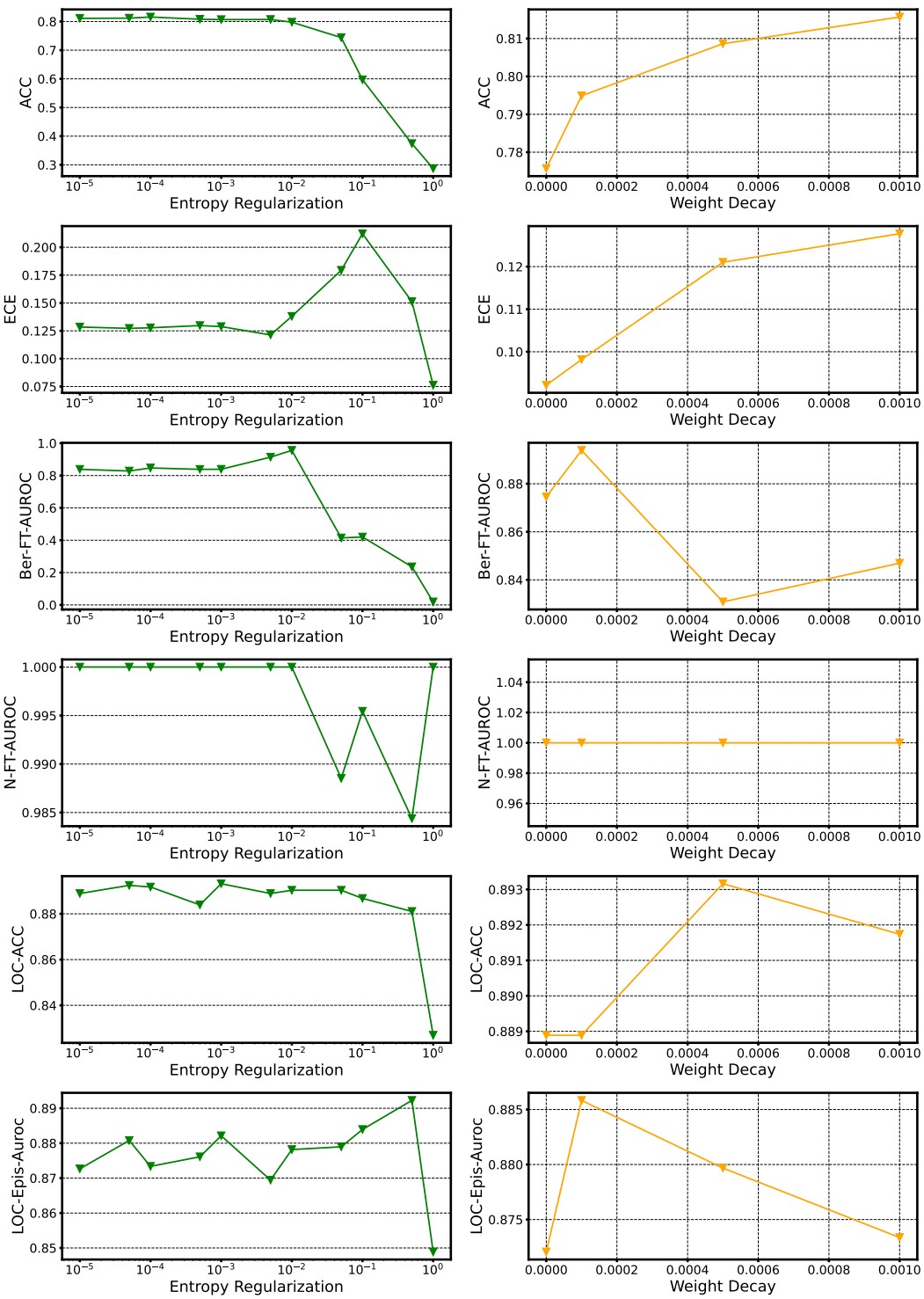

Figure 4: Accuracy, Calibration, and OOD-detection results of GPN on CoraML for the entropy regularization factor and the weight decay.

| | Model | $x^{(v)} \sim \text{Ber}(0.5)$ | | | | $x^{(v)} \sim \mathcal{N}(0,1)$ | | | |
| | | **OOD-ACC** | **OOD-AUC-PR** | | | **OOD-ACC** | **OOD-AUC-PR** | | |
| | | | *Alea w/ Net* | *Epist w/ Net* | *Epist w/o Net* | | *Alea w/ Net* | *Epist w/ Net* | *Epist w/o Net* |
|---|---|---|---|---|---|---|---|---|---|
| **CoraML** | APPNP | $80.85 \pm 0.09$ | $11.86 \pm 0.06$ | *n.a.* | *n.a.* | $17.99 \pm 0.36$ | $2.55 \pm 0.00$ | *n.a.* | *n.a.* |
| | VGCN | $78.90 \pm 0.09$ | $11.92 \pm 0.07$ | *n.a.* | *n.a.* | $18.37 \pm 0.31$ | $2.57 \pm 0.00$ | *n.a.* | *n.a.* |
| | VGCN-Dropout | $77.76 \pm 0.15$ | $12.51 \pm 0.21$ | $4.58 \pm 0.01$ | *n.a.* | $18.28 \pm 0.35$ | $3.91 \pm 0.02$ | $62.19 \pm 0.39$ | *n.a.* |
| | DropEdge | $77.40 \pm 0.14$ | $10.67 \pm 0.09$ | $4.90 \pm 0.01$ | *n.a.* | $16.60 \pm 0.26$ | $2.93 \pm 0.01$ | $42.96 \pm 0.39$ | *n.a.* |
| | VGCN-Energy | $78.90 \pm 0.09$ | $11.92 \pm 0.07$ | $10.43 \pm 0.08$ | *n.a.* | $18.37 \pm 0.31$ | $2.57 \pm 0.00$ | $2.50 \pm 0.00$ | *n.a.* |
| | VGCN-Ensemble | $78.00 \pm 0.00$ | $11.78 \pm 0.00$ | $5.38 \pm 0.00$ | *n.a.* | $21.00 \pm 0.00$ | $3.43 \pm 0.00$ | $60.47 \pm 0.09$ | *n.a.* |
| | VGCN-BNN | $77.01 \pm 0.16$ | $10.94 \pm 0.09$ | $7.75 \pm 0.15$ | *n.a.* | $18.79 \pm 0.31$ | $3.49 \pm 0.03$ | $62.75 \pm 0.65$ | *n.a.* |
| | RGCN | $79.78 \pm 0.16$ | $16.43 \pm 0.14$ | *n.a.* | *n.a.* | $33.37 \pm 0.35$ | $4.82 \pm 0.14$ | *n.a.* | *n.a.* |
| | GKDE-GCN | $76.40 \pm 0.33$ | $9.55 \pm 0.10$ | $9.79 \pm 0.09$ | *n.a.* | $16.86 \pm 0.35$ | $34.20 \pm 0.53$ | $2.64 \pm 0.06$ | *n.a.* |
| | GPN | $\mathbf{80.98 \pm 0.22}$ | $6.96 \pm 0.07$ | $5.63 \pm 0.03$ | $\mathbf{25.80 \pm 1.43}$ | $\mathbf{81.53 \pm 0.23}$ | $6.63 \pm 0.07$ | $6.21 \pm 0.08$ | $\mathbf{100.00 \pm 0.00}$ |
| **CiteSeer** | APPNP | $\mathbf{73.14 \pm 0.12}$ | $7.49 \pm 0.01$ | *n.a.* | *n.a.* | $20.13 \pm 0.22$ | $2.27 \pm 0.00$ | *n.a.* | *n.a.* |
| | VGCN | $71.30 \pm 0.13$ | $8.92 \pm 0.03$ | *n.a.* | *n.a.* | $17.55 \pm 0.36$ | $2.28 \pm 0.00$ | *n.a.* | *n.a.* |
| | RGCN | $72.29 \pm 0.09$ | $14.49 \pm 0.14$ | *n.a.* | *n.a.* | $28.15 \pm 0.40$ | $3.26 \pm 0.07$ | *n.a.* | *n.a.* |
| | VGCN-Dropout | $69.80 \pm 0.19$ | $8.07 \pm 0.04$ | $4.26 \pm 0.01$ | *n.a.* | $19.60 \pm 0.28$ | $2.97 \pm 0.01$ | $64.29 \pm 0.37$ | *n.a.* |
| | DropEdge | $72.00 \pm 0.23$ | $9.34 \pm 0.10$ | $4.60 \pm 0.02$ | *n.a.* | $18.00 \pm 0.47$ | $2.48 \pm 0.01$ | $42.23 \pm 0.50$ | *n.a.* |
| | VGCN-Energy | $71.30 \pm 0.13$ | $8.92 \pm 0.03$ | $9.09 \pm 0.05$ | *n.a.* | $17.55 \pm 0.36$ | $2.28 \pm 0.00$ | $2.26 \pm 0.00$ | *n.a.* |
| | VGCN-Ensemble | $72.00 \pm 0.00$ | $8.76 \pm 0.00$ | $4.34 \pm 0.00$ | *n.a.* | $18.00 \pm 0.00$ | $2.58 \pm 0.00$ | $49.06 \pm 0.01$ | *n.a.* |
| | VGCN-BNN | $70.38 \pm 0.15$ | $8.86 \pm 0.07$ | $5.14 \pm 0.16$ | *n.a.* | $16.27 \pm 0.33$ | $2.68 \pm 0.02$ | $57.70 \pm 1.51$ | *n.a.* |
| | GKDE-GCN | $72.75 \pm 0.18$ | $8.57 \pm 0.06$ | $9.82 \pm 0.11$ | *n.a.* | $18.79 \pm 0.33$ | $33.75 \pm 0.72$ | $2.27 \pm 0.01$ | *n.a.* |
| | GPN | $65.00 \pm 0.43$ | $6.27 \pm 0.09$ | $5.37 \pm 0.06$ | $\mathbf{14.25 \pm 1.03}$ | $\mathbf{66.70 \pm 0.23}$ | $4.86 \pm 0.03$ | $29.98 \pm 0.62$ | $\mathbf{100.00 \pm 0.00}$ |
| **PubMed** | APPNP | $82.80 \pm 0.10$ | $1.08 \pm 0.00$ | *n.a.* | *n.a.* | $40.38 \pm 0.22$ | $0.40 \pm 0.00$ | *n.a.* | *n.a.* |
| | VGCN | $82.49 \pm 0.10$ | $1.29 \pm 0.01$ | *n.a.* | *n.a.* | $37.80 \pm 0.40$ | $0.40 \pm 0.00$ | *n.a.* | *n.a.* |
| | RGCN | $\mathbf{83.75 \pm 0.12}$ | $1.42 \pm 0.03$ | *n.a.* | *n.a.* | $47.82 \pm 0.36$ | $0.64 \pm 0.02$ | *n.a.* | *n.a.* |
| | VGCN-Dropout | $82.26 \pm 0.06$ | $1.41 \pm 0.03$ | $0.77 \pm 0.00$ | *n.a.* | $37.79 \pm 0.45$ | $0.48 \pm 0.00$ | $25.20 \pm 0.50$ | *n.a.* |
| | DropEdge | $82.70 \pm 0.12$ | $1.25 \pm 0.01$ | $0.93 \pm 0.01$ | *n.a.* | $36.36 \pm 0.47$ | $0.42 \pm 0.00$ | $15.48 \pm 0.51$ | *n.a.* |
| | VGCN-Energy | $82.49 \pm 0.10$ | $1.29 \pm 0.01$ | $1.50 \pm 0.01$ | *n.a.* | $37.80 \pm 0.40$ | $0.40 \pm 0.00$ | $0.58 \pm 0.04$ | *n.a.* |
| | VGCN-Ensemble | $82.00 \pm 0.00$ | $1.46 \pm 0.01$ | $1.05 \pm 0.00$ | *n.a.* | $39.10 \pm 0.10$ | $0.42 \pm 0.00$ | $13.71 \pm 0.03$ | *n.a.* |
| | VGCN-BNN | $82.30 \pm 0.14$ | $1.28 \pm 0.01$ | $1.77 \pm 0.16$ | *n.a.* | $37.56 \pm 0.54$ | $0.42 \pm 0.00$ | $15.60 \pm 0.60$ | *n.a.* |
| | GKDE-GCN | $82.54 \pm 0.11$ | $1.31 \pm 0.03$ | $1.31 \pm 0.01$ | *n.a.* | $37.77 \pm 0.48$ | $16.95 \pm 0.49$ | $0.92 \pm 0.08$ | *n.a.* |
| | GPN | $81.54 \pm 0.39$ | $0.85 \pm 0.00$ | $1.01 \pm 0.01$ | $\mathbf{3.31 \pm 0.24}$ | $\mathbf{81.73 \pm 0.34}$ | $0.78 \pm 0.00$ | $1.27 \pm 0.01$ | $\mathbf{99.98 \pm 0.00}$ |
| **Amazon Computers** | APPNP | $75.00 \pm 0.09$ | $2.18 \pm 0.01$ | *n.a.* | *n.a.* | $18.25 \pm 0.46$ | $0.56 \pm 0.00$ | *n.a.* | *n.a.* |
| | VGCN | $81.54 \pm 0.08$ | $1.43 \pm 0.00$ | *n.a.* | *n.a.* | $20.38 \pm 0.29$ | $0.67 \pm 0.00$ | *n.a.* | *n.a.* |
| | RGCN | $61.39 \pm 0.39$ | $1.40 \pm 0.00$ | *n.a.* | *n.a.* | $39.60 \pm 0.45$ | $0.86 \pm 0.02$ | *n.a.* | *n.a.* |
| | VGCN-Dropout | $\mathbf{81.79 \pm 0.10}$ | $1.38 \pm 0.00$ | $1.24 \pm 0.00$ | *n.a.* | $21.52 \pm 0.36$ | $0.88 \pm 0.01$ | $20.49 \pm 0.27$ | *n.a.* |
| | DropEdge | $81.20 \pm 0.08$ | $1.46 \pm 0.00$ | $1.23 \pm 0.00$ | *n.a.* | $21.75 \pm 0.36$ | $0.77 \pm 0.00$ | $16.09 \pm 0.22$ | *n.a.* |
| | VGCN-Energy | $81.54 \pm 0.08$ | $1.43 \pm 0.00$ | $1.44 \pm 0.00$ | *n.a.* | $20.38 \pm 0.29$ | $0.67 \pm 0.00$ | $0.56 \pm 0.00$ | *n.a.* |
| | VGCN-Ensemble | $81.00 \pm 0.00$ | $1.44 \pm 0.00$ | $1.13 \pm 0.00$ | *n.a.* | $23.60 \pm 0.16$ | $0.71 \pm 0.00$ | $51.48 \pm 0.16$ | *n.a.* |
| | VGCN-BNN | $61.25 \pm 0.17$ | $1.35 \pm 0.03$ | $1.29 \pm 0.05$ | *n.a.* | $28.54 \pm 0.47$ | $0.64 \pm 0.00$ | $21.82 \pm 0.68$ | *n.a.* |
| | GKDE-GCN | $59.83 \pm 0.73$ | $1.35 \pm 0.01$ | $1.32 \pm 0.00$ | *n.a.* | $28.46 \pm 0.36$ | $2.96 \pm 0.14$ | $1.86 \pm 0.16$ | *n.a.* |
| | GPN | $79.70 \pm 0.46$ | $1.55 \pm 0.01$ | $1.55 \pm 0.01$ | $\mathbf{4.26 \pm 0.08}$ | $\mathbf{79.87 \pm 0.46}$ | $2.56 \pm 0.01$ | $2.77 \pm 0.03$ | $\mathbf{100.00 \pm 0.00}$ |
| **Amazon Photos** | APPNP | $\mathbf{88.12 \pm 0.10}$ | $\mathbf{5.44 \pm 0.05}$ | *n.a.* | *n.a.* | $19.37 \pm 0.45$ | $1.02 \pm 0.00$ | *n.a.* | *n.a.* |
| | VGCN | $83.91 \pm 0.08$ | $3.61 \pm 0.07$ | *n.a.* | *n.a.* | $21.40 \pm 0.49$ | $1.39 \pm 0.01$ | *n.a.* | *n.a.* |
| | RGCN | $79.50 \pm 0.72$ | $3.33 \pm 0.04$ | *n.a.* | *n.a.* | $42.38 \pm 0.40$ | $1.85 \pm 0.05$ | *n.a.* | *n.a.* |
| | VGCN-Dropout | $83.86 \pm 0.18$ | $3.12 \pm 0.04$ | $2.19 \pm 0.01$ | *n.a.* | $22.29 \pm 0.55$ | $2.09 \pm 0.02$ | $21.31 \pm 0.24$ | *n.a.* |
| | DropEdge | $85.69 \pm 0.15$ | $3.56 \pm 0.03$ | $2.25 \pm 0.01$ | *n.a.* | $22.90 \pm 0.43$ | $1.63 \pm 0.01$ | $16.54 \pm 0.20$ | *n.a.* |
| | VGCN-Energy | $83.91 \pm 0.08$ | $3.61 \pm 0.07$ | $5.32 \pm 0.07$ | *n.a.* | $21.40 \pm 0.49$ | $1.39 \pm 0.01$ | $1.01 \pm 0.00$ | *n.a.* |
| | VGCN-Ensemble | $84.40 \pm 0.16$ | $3.27 \pm 0.02$ | $2.27 \pm 0.02$ | *n.a.* | $20.30 \pm 0.21$ | $1.87 \pm 0.00$ | $59.99 \pm 0.19$ | *n.a.* |
| | VGCN-BNN | $82.00 \pm 0.19$ | $3.70 \pm 0.09$ | $2.07 \pm 0.04$ | *n.a.* | $25.84 \pm 0.46$ | $1.18 \pm 0.01$ | $28.71 \pm 0.92$ | *n.a.* |
| | GKDE-GCN | $73.17 \pm 0.94$ | $2.72 \pm 0.04$ | $3.13 \pm 0.04$ | *n.a.* | $24.04 \pm 0.42$ | $4.88 \pm 0.19$ | $1.17 \pm 0.03$ | *n.a.* |
| | GPN | $87.47 \pm 0.20$ | $2.38 \pm 0.01$ | $2.81 \pm 0.02$ | $4.66 \pm 0.18$ | $\mathbf{88.29 \pm 0.20}$ | $2.10 \pm 0.01$ | $3.32 \pm 0.04$ | $\mathbf{100.00 \pm 0.00}$ |
| **Coauthor CS** | APPNP | $89.28 \pm 0.07$ | $2.32 \pm 0.01$ | *n.a.* | *n.a.* | $12.58 \pm 0.33$ | $0.50 \pm 0.00$ | *n.a.* | *n.a.* |
| | VGCN | $89.33 \pm 0.05$ | $1.75 \pm 0.01$ | *n.a.* | *n.a.* | $13.29 \pm 0.22$ | $0.57 \pm 0.00$ | *n.a.* | *n.a.* |
| | RGCN | $\mathbf{90.50 \pm 0.06}$ | $1.89 \pm 0.00$ | *n.a.* | *n.a.* | $41.67 \pm 0.35$ | $1.16 \pm 0.02$ | *n.a.* | *n.a.* |
| | VGCN-Dropout | $88.96 \pm 0.09$ | $1.62 \pm 0.02$ | $1.05 \pm 0.00$ | *n.a.* | $13.31 \pm 0.32$ | $1.68 \pm 0.02$ | $71.46 \pm 0.30$ | *n.a.* |
| | DropEdge | $89.44 \pm 0.07$ | $1.97 \pm 0.03$ | $1.18 \pm 0.00$ | *n.a.* | $11.65 \pm 0.28$ | $0.90 \pm 0.01$ | $55.73 \pm 0.45$ | *n.a.* |
| | VGCN-Energy | $89.33 \pm 0.05$ | $1.75 \pm 0.01$ | $2.38 \pm 0.01$ | *n.a.* | $13.29 \pm 0.22$ | $0.57 \pm 0.00$ | $0.42 \pm 0.00$ | *n.a.* |
| | VGCN-Ensemble | $89.00 \pm 0.00$ | $1.76 \pm 0.00$ | $1.19 \pm 0.00$ | *n.a.* | $11.00 \pm 0.00$ | $1.39 \pm 0.01$ | $73.57 \pm 0.28$ | *n.a.* |
| | VGCN-BNN | $88.16 \pm 0.10$ | $1.91 \pm 0.02$ | $1.01 \pm 0.01$ | *n.a.* | $12.48 \pm 0.25$ | $1.53 \pm 0.03$ | $73.43 \pm 0.70$ | *n.a.* |
| | GKDE-GCN | $88.14 \pm 0.14$ | $2.00 \pm 0.02$ | $2.70 \pm 0.02$ | *n.a.* | $9.71 \pm 0.29$ | $15.14 \pm 0.55$ | $0.42 \pm 0.00$ | *n.a.* |
| | GPN | $83.99 \pm 0.31$ | $1.04 \pm 0.01$ | $1.22 \pm 0.01$ | $\mathbf{29.25 \pm 1.92}$ | $\mathbf{83.96 \pm 0.31}$ | $1.02 \pm 0.01$ | $1.26 \pm 0.01$ | $\mathbf{100.00 \pm 0.00}$ |
| **Coauthor Physics** | APPNP | $96.16 \pm 0.08$ | $0.83 \pm 0.00$ | *n.a.* | *n.a.* | $28.71 \pm 0.40$ | $0.30 \pm 0.00$ | *n.a.* | *n.a.* |
| | VGCN | $96.00 \pm 0.00$ | $0.62 \pm 0.00$ | *n.a.* | *n.a.* | $33.26 \pm 0.67$ | $0.44 \pm 0.01$ | *n.a.* | *n.a.* |
| | RGCN | $94.69 \pm 0.07$ | $0.80 \pm 0.00$ | *n.a.* | *n.a.* | $58.56 \pm 0.36$ | $0.75 \pm 0.02$ | *n.a.* | *n.a.* |
| | VGCN-Dropout | $95.90 \pm 0.05$ | $0.61 \pm 0.00$ | $0.57 \pm 0.00$ | *n.a.* | $32.52 \pm 0.60$ | $0.75 \pm 0.01$ | $19.76 \pm 0.41$ | *n.a.* |
| | DropEdge | $95.90 \pm 0.03$ | $0.62 \pm 0.00$ | $0.63 \pm 0.01$ | *n.a.* | $30.53 \pm 0.58$ | $0.53 \pm 0.01$ | $12.71 \pm 0.29$ | *n.a.* |
| | VGCN-Energy | $96.00 \pm 0.00$ | $0.62 \pm 0.00$ | $0.72 \pm 0.00$ | *n.a.* | $33.26 \pm 0.67$ | $0.44 \pm 0.01$ | $0.23 \pm 0.00$ | *n.a.* |
| | VGCN-Ensemble | $96.00 \pm 0.00$ | $0.62 \pm 0.00$ | $0.57 \pm 0.00$ | *n.a.* | $31.70 \pm 0.21$ | $0.71 \pm 0.01$ | $41.30 \pm 0.30$ | *n.a.* |
| | VGCN-BNN | $95.65 \pm 0.07$ | $0.69 \pm 0.01$ | $0.57 \pm 0.01$ | *n.a.* | $32.95 \pm 0.60$ | $1.17 \pm 0.04$ | $44.88 \pm 0.80$ | *n.a.* |
| | GKDE-GCN | $\mathbf{96.61 \pm 0.05}$ | $0.62 \pm 0.00$ | $0.74 \pm 0.00$ | *n.a.* | $28.84 \pm 0.31$ | $6.53 \pm 0.18$ | $0.22 \pm 0.00$ | *n.a.* |
| | GPN | $92.70 \pm 0.11$ | $0.63 \pm 0.00$ | $0.60 \pm 0.00$ | $\mathbf{23.91 \pm 1.06}$ | $\mathbf{92.70 \pm 0.11}$ | $0.62 \pm 0.00$ | $0.62 \pm 0.00$ | $\mathbf{100.00 \pm 0.00}$ |
| **OGBN Arxiv** | APPNP | $63.50 \pm 0.95$ | $0.79 \pm 0.16$ | *n.a.* | *n.a.* | $51.10 \pm 1.12$ | $0.42 \pm 0.10$ | *n.a.* | *n.a.* |
| | VGCN | $65.70 \pm 0.47$ | $0.20 \pm 0.00$ | *n.a.* | *n.a.* | $51.30 \pm 0.75$ | $0.30 \pm 0.02$ | *n.a.* | *n.a.* |
| | VGCN-Dropout | $65.30 \pm 0.70$ | $0.21 \pm 0.00$ | $0.37 \pm 0.03$ | *n.a.* | $49.90 \pm 0.77$ | $0.47 \pm 0.05$ | $17.68 \pm 0.48$ | *n.a.* |
| | VGCN-Energy | $65.70 \pm 0.47$ | $0.20 \pm 0.00$ | $2.32 \pm 0.38$ | *n.a.* | $51.30 \pm 0.75$ | $0.30 \pm 0.02$ | $0.24 \pm 0.01$ | *n.a.* |
| | VGCN-Ensemble | $\mathbf{67.00}$ | $0.20$ | $0.19$ | *n.a.* | $49.00$ | $0.46$ | $18.22$ | *n.a.* |
| | GKDE-GCN | $65.20 \pm 0.49$ | $0.76 \pm 0.05$ | $0.76 \pm 0.05$ | *n.a.* | $45.40 \pm 0.62$ | $4.99 \pm 0.97$ | $5.04 \pm 0.97$ | *n.a.* |
| | GPN | $65.50 \pm 0.70$ | $0.23 \pm 0.01$ | $0.26 \pm 0.01$ | $\mathbf{46.84 \pm 5.20}$ | $\mathbf{65.50 \pm 0.70}$ | $0.23 \pm 0.01$ | $0.26 \pm 0.01$ | $\mathbf{48.97 \pm 1.51}$ |

Table 13: Accuracy and OOD detection scores on Bernoulli and unit Gaussian feature perturbations using AUC-APR. OOD-AUC-APR scores are given as *[Alea w/ Net] / [Epist w/ Net] / [Epist w/o Net]*. *n.a.* means either model or metric not applicable. Bold numbers indicate best results for Accuracy and OOD detection.

| | | Clean Graph | | Leave-Out Classes | | | | | | |
|---|---|---|---|---|---|---|---|---|---|---|
| | **Model** | **ID-ACC** | **ID-ECE** | **ID-ACC** | **OOD-AUC-ROC** | | | **OOD-AUC-PR** | | |
| | | | | | *Alea w/ Net* | *Epist w/ Net* | *Epist w/o Net* | *Alea w/ Net* | *Epist w/ Net* | *Epist w/o Net* |
| CoraML | LP | $78.41 \pm 0.00$ | $64.12 \pm 0.00$ | $86.40 \pm 0.00$ | $83.78 \pm 0.00$ | $80.86 \pm 0.00$ | *n.a.* | $74.80 \pm 0.00$ | $71.15 \pm 0.00$ | *n.a.* |
| | GKDE | $72.88 \pm 0.00$ | $56.46 \pm 0.00$ | $83.02 \pm 0.00$ | $74.46 \pm 0.00$ | $71.86 \pm 0.00$ | *n.a.* | $66.19 \pm 0.00$ | $64.05 \pm 0.00$ | *n.a.* |
| | Matern-GGP | $79.70 \pm 0.02$ | $9.88 \pm 0.02$ | $87.03 \pm 0.01$ | $83.13 \pm 0.00$ | $82.98 \pm 0.00$ | *n.a.* | $71.42 \pm 0.00$ | $71.04 \pm 0.14$ | *n.a.* |
| | GGP | $79.04 \pm 0.01$ | $21.67 \pm 0.01$ | $88.65 \pm 0.00$ | $81.49 \pm 0.00$ | $82.03 \pm 0.00$ | *n.a.* | $74.13 \pm 0.00$ | $74.77 \pm 0.00$ | *n.a.* |
| | APPNP | $\mathbf{84.94} \pm 0.02$ | $8.27 \pm 0.10$ | $\mathbf{90.20} \pm 0.02$ | $83.71 \pm 0.06$ | *n.a.* | *n.a.* | $78.77 \pm 0.07$ | *n.a.* | *n.a.* |
| | VGCN | $83.14 \pm 0.03$ | $7.96 \pm 0.16$ | $89.66 \pm 0.05$ | $81.70 \pm 0.07$ | *n.a.* | *n.a.* | $75.67 \pm 0.10$ | *n.a.* | *n.a.* |
| | RGCN | $82.79 \pm 0.06$ | $14.39 \pm 0.17$ | $88.66 \pm 0.03$ | $80.37 \pm 0.11$ | *n.a.* | *n.a.* | $76.97 \pm 0.11$ | *n.a.* | *n.a.* |
| | VGCN-Dropout | $82.30 \pm 0.06$ | $13.88 \pm 0.13$ | $89.08 \pm 0.04$ | $81.27 \pm 0.07$ | $71.65 \pm 0.10$ | *n.a.* | $75.55 \pm 0.12$ | $60.65 \pm 0.11$ | *n.a.* |
| | DropEdge | $83.07 \pm 0.04$ | $13.93 \pm 0.11$ | $89.03 \pm 0.03$ | $83.55 \pm 0.05$ | $75.48 \pm 0.12$ | *n.a.* | $78.48 \pm 0.12$ | $65.22 \pm 0.15$ | *n.a.* |
| | VGCN-Energy | $83.14 \pm 0.03$ | $7.96 \pm 0.16$ | $89.66 \pm 0.05$ | $81.70 \pm 0.07$ | $83.15 \pm 0.07$ | *n.a.* | $75.67 \pm 0.10$ | $78.44 \pm 0.10$ | *n.a.* |
| | VGCN-Ensemble | $83.41 \pm 0.01$ | $8.45 \pm 0.01$ | $89.87 \pm 0.00$ | $81.85 \pm 0.00$ | $74.24 \pm 0.00$ | *n.a.* | $75.80 \pm 0.00$ | $64.02 \pm 0.00$ | *n.a.* |
| | VGCN-BNN | $82.83 \pm 0.06$ | $15.66 \pm 0.13$ | $88.49 \pm 0.04$ | $82.18 \pm 0.23$ | $73.18 \pm 1.10$ | *n.a.* | $76.17 \pm 0.36$ | $63.51 \pm 1.40$ | *n.a.* |
| | GKDE-GCN | $81.91 \pm 0.19$ | $\mathbf{6.53} \pm 0.14$ | $89.33 \pm 0.04$ | $82.23 \pm 0.08$ | $82.09 \pm 0.18$ | *n.a.* | $75.88 \pm 0.12$ | $77.03 \pm 0.39$ | *n.a.* |
| | GPN | $81.16 \pm 0.12$ | $11.68 \pm 0.18$ | $88.51 \pm 0.04$ | $83.25 \pm 0.12$ | $\mathbf{86.28} \pm 0.17$ | $80.95 \pm 0.24$ | $75.79 \pm 0.28$ | $\mathbf{79.97} \pm 0.20$ | $72.81 \pm 0.46$ |
| CiteSeer | LP | $54.05 \pm 0.00$ | $37.38 \pm 0.00$ | $57.34 \pm 0.00$ | $65.99 \pm 0.00$ | $67.54 \pm 0.00$ | *n.a.* | $48.12 \pm 0.00$ | $48.59 \pm 0.00$ | *n.a.* |
| | GKDE | $53.67 \pm 0.00$ | $36.29 \pm 0.00$ | $49.62 \pm 0.00$ | $63.75 \pm 0.00$ | $63.91 \pm 0.00$ | *n.a.* | $56.74 \pm 0.00$ | $56.79 \pm 0.00$ | *n.a.* |
| | Matern-GGP | $53.25 \pm 0.03$ | $12.76 \pm 0.03$ | $53.83 \pm 0.06$ | $59.57 \pm 0.00$ | $59.56 \pm 0.05$ | *n.a.* | $36.05 \pm 0.04$ | $36.24 \pm 0.19$ | *n.a.* |
| | GGP | $68.85 \pm 0.01$ | $20.84 \pm 0.01$ | $69.74 \pm 0.01$ | $\mathbf{74.56} \pm 0.06$ | $74.10 \pm 0.07$ | *n.a.* | $50.39 \pm 0.08$ | $48.74 \pm 0.08$ | *n.a.* |
| | APPNP | $70.24 \pm 0.02$ | $4.88 \pm 0.03$ | $72.83 \pm 0.03$ | $72.91 \pm 0.09$ | *n.a.* | *n.a.* | $56.31 \pm 0.14$ | *n.a.* | *n.a.* |
| | VGCN | $68.98 \pm 0.02$ | $4.33 \pm 0.05$ | $70.79 \pm 0.02$ | $72.16 \pm 0.08$ | *n.a.* | *n.a.* | $53.71 \pm 0.08$ | *n.a.* | *n.a.* |
| | RGCN | $\mathbf{70.57} \pm 0.03$ | $15.09 \pm 0.09$ | $72.15 \pm 0.03$ | $74.56 \pm 0.07$ | *n.a.* | *n.a.* | $\mathbf{58.63} \pm 0.07$ | *n.a.* | *n.a.* |
| | VGCN-Dropout | $68.41 \pm 0.03$ | $7.31 \pm 0.07$ | $70.44 \pm 0.06$ | $71.31 \pm 0.08$ | $60.05 \pm 0.12$ | *n.a.* | $52.05 \pm 0.14$ | $36.95 \pm 0.10$ | *n.a.* |
| | DropEdge | $69.33 \pm 0.03$ | $5.64 \pm 0.06$ | $71.02 \pm 0.05$ | $73.42 \pm 0.05$ | $63.23 \pm 0.17$ | *n.a.* | $55.70 \pm 0.10$ | $39.38 \pm 0.14$ | *n.a.* |
| | VGCN-Energy | $68.98 \pm 0.02$ | $4.33 \pm 0.05$ | $70.79 \pm 0.02$ | $72.16 \pm 0.08$ | $76.08 \pm 0.11$ | *n.a.* | $53.71 \pm 0.08$ | $58.35 \pm 0.17$ | *n.a.* |
| | VGCN-Ensemble | $69.26 \pm 0.00$ | $4.14 \pm 0.02$ | $70.63 \pm 0.00$ | $72.23 \pm 0.00$ | $58.61 \pm 0.01$ | *n.a.* | $54.04 \pm 0.00$ | $38.93 \pm 0.01$ | *n.a.* |
| | VGCN-BNN | $68.06 \pm 0.07$ | $8.42 \pm 0.20$ | $69.84 \pm 0.04$ | $71.64 \pm 0.31$ | $64.16 \pm 1.75$ | *n.a.* | $52.60 \pm 0.47$ | $46.72 \pm 1.76$ | *n.a.* |
| | GKDE-GCN | $69.55 \pm 0.03$ | $\mathbf{3.88} \pm 0.06$ | $70.76 \pm 0.04$ | $73.34 \pm 0.15$ | $\mathbf{76.19} \pm 0.31$ | *n.a.* | $54.25 \pm 0.16$ | $59.07 \pm 0.42$ | *n.a.* |
| | GPN | $66.13 \pm 0.17$ | $7.42 \pm 0.22$ | $69.79 \pm 0.10$ | $72.46 \pm 0.27$ | $70.74 \pm 0.26$ | $66.65 \pm 0.29$ | $55.14 \pm 0.46$ | $50.52 \pm 0.34$ | $44.93 \pm 0.31$ |
| PubMed | LP | $78.40 \pm 0.00$ | $45.07 \pm 0.00$ | $89.18 \pm 0.00$ | $\mathbf{80.32} \pm 0.00$ | $79.64 \pm 0.00$ | *n.a.* | $71.01 \pm 0.00$ | $\mathbf{72.98} \pm 0.00$ | *n.a.* |
| | GKDE | $77.10 \pm 0.01$ | $40.02 \pm 0.01$ | $88.16 \pm 0.00$ | $69.66 \pm 0.00$ | $68.47 \pm 0.00$ | *n.a.* | $55.81 \pm 0.00$ | $54.33 \pm 0.00$ | *n.a.* |
| | Matern-GGP | $78.77 \pm 0.00$ | $12.37 \pm 0.00$ | $90.33 \pm 0.01$ | $46.69 \pm 0.00$ | $45.75 \pm 0.00$ | *n.a.* | $39.85 \pm 0.00$ | $39.63 \pm 0.00$ | *n.a.* |
| | GGP | *n.f.* | *n.f.* | *n.f.* | *n.f.* | *n.f.* | *n.f.* | *n.f.* | *n.f.* | *n.f.* |
| | APPNP | $\mathbf{86.88} \pm 0.01$ | $2.57 \pm 0.05$ | $94.83 \pm 0.01$ | $74.76 \pm 0.06$ | *n.a.* | *n.a.* | $61.84 \pm 0.07$ | *n.a.* | *n.a.* |
| | VGCN | $86.70 \pm 0.01$ | $2.30 \pm 0.05$ | $94.77 \pm 0.01$ | $72.58 \pm 0.04$ | *n.a.* | *n.a.* | $60.54 \pm 0.04$ | *n.a.* | *n.a.* |
| | RGCN | $85.87 \pm 0.01$ | $4.86 \pm 0.05$ | $94.73 \pm 0.01$ | $71.49 \pm 0.14$ | *n.a.* | *n.a.* | $60.54 \pm 0.13$ | *n.a.* | *n.a.* |
| | VGCN-Dropout | $86.49 \pm 0.01$ | $5.53 \pm 0.06$ | $94.72 \pm 0.01$ | $71.10 \pm 0.04$ | $67.27 \pm 0.06$ | *n.a.* | $59.47 \pm 0.04$ | $54.24 \pm 0.08$ | *n.a.* |
| | DropEdge | $86.57 \pm 0.01$ | $5.11 \pm 0.04$ | $94.72 \pm 0.01$ | $72.09 \pm 0.02$ | $68.57 \pm 0.04$ | *n.a.* | $59.84 \pm 0.03$ | $54.95 \pm 0.06$ | *n.a.* |
| | VGCN-Energy | $86.70 \pm 0.01$ | $2.30 \pm 0.05$ | $94.77 \pm 0.01$ | $72.58 \pm 0.06$ | $72.63 \pm 0.06$ | *n.a.* | $60.54 \pm 0.04$ | $60.63 \pm 0.10$ | *n.a.* |
| | VGCN-Ensemble | $86.64 \pm 0.00$ | $2.29 \pm 0.01$ | $\mathbf{94.88} \pm 0.00$ | $72.71 \pm 0.00$ | $70.99 \pm 0.00$ | *n.a.* | $60.47 \pm 0.00$ | $59.31 \pm 0.00$ | *n.a.* |
| | VGCN-BNN | $85.56 \pm 0.05$ | $14.00 \pm 0.11$ | $94.34 \pm 0.03$ | $65.41 \pm 0.58$ | $63.77 \pm 1.50$ | *n.a.* | $53.23 \pm 0.40$ | $54.36 \pm 1.54$ | *n.a.* |
| | GKDE-GCN | $86.14 \pm 0.07$ | $\mathbf{1.36} \pm 0.09$ | $94.66 \pm 0.00$ | $73.53 \pm 0.06$ | $74.47 \pm 0.11$ | *n.a.* | $61.36 \pm 0.04$ | $61.96 \pm 0.27$ | *n.a.* |
| | GPN | $84.10 \pm 0.26$ | $4.31 \pm 0.09$ | $94.08 \pm 0.02$ | $71.84 \pm 0.08$ | $73.91 \pm 0.20$ | $71.20 \pm 0.15$ | $57.92 \pm 0.10$ | $67.19 \pm 0.25$ | $59.72 \pm 0.18$ |

Table 14: Accuracy and ECE scores on the clean graphs. Accuracy and OOD detection scores on Left-Out classes using AUC-ROC and AUC-PR scores. OOD-AUC-ROC and OOD-AUC-APR scores are given as *[Alea w/ Net] / [Epist w/ Net] / [Epist w/o Net]*. $n.a.$ means model or metric not applicable and $n.f.$ means not finished within our constraints. Bold numbers indicate best results for Accuracy, ECE and OOD detection.

| | Model | Clean Graph | | Leave-Out Classes | | | | | | |
| | | ID-ACC | ID-ECE | ID-ACC | OOD-AUC-ROC | | | OOD-AUC-PR | | |
| | | | | | Alea w/ Net | Epist w/ Net | Epist w/o Net | Alea w/ Net | Epist w/ Net | Epist w/o Net |
|---|---|---|---|---|---|---|---|---|---|---|
| Amazon Computers | LP | $79.49 \pm 0.00$ | $69.49 \pm 0.00$ | $83.28 \pm 0.00$ | $86.74 \pm 0.00$ | $83.88 \pm 0.00$ | *n.a.* | $67.10 \pm 0.00$ | $63.08 \pm 0.00$ | *n.a.* |
| | GKDE | $63.49 \pm 0.00$ | $38.93 \pm 0.00$ | $71.41 \pm 0.00$ | $75.14 \pm 0.00$ | $73.58 \pm 0.00$ | *n.a.* | $49.21 \pm 0.00$ | $47.68 \pm 0.00$ | *n.a.* |
| | Matern-GGP | $80.23 \pm 0.00$ | $12.13 \pm 0.01$ | $86.94 \pm 0.01$ | $79.00 \pm 0.00$ | $79.10 \pm 0.00$ | *n.a.* | $49.57 \pm 0.00$ | $49.91 \pm 0.04$ | *n.a.* |
| | GGP | *n.f.* | *n.f.* | *n.f.* | *n.f.* | *n.f.* | *n.f.* | *n.f.* | *n.f.* | *n.f.* |
| | APPNP | $80.12 \pm 0.04$ | $11.83 \pm 0.04$ | $87.72 \pm 0.02$ | $81.30 \pm 0.02$ | *n.a.* | *n.a.* | $53.02 \pm 0.04$ | *n.a.* | *n.a.* |
| | VGCN | $81.66 \pm 0.04$ | $9.90 \pm 0.03$ | $88.95 \pm 0.02$ | $82.76 \pm 0.03$ | *n.a.* | *n.a.* | $57.49 \pm 0.06$ | *n.a.* | *n.a.* |
| | RGCN | $68.43 \pm 0.24$ | $23.62 \pm 0.16$ | $78.52 \pm 0.13$ | $76.40 \pm 0.16$ | *n.a.* | *n.a.* | $53.16 \pm 0.20$ | *n.a.* | *n.a.* |
| | VGCN-Dropout | $81.29 \pm 0.05$ | $11.52 \pm 0.03$ | $88.54 \pm 0.02$ | $81.99 \pm 0.04$ | $72.90 \pm 0.04$ | *n.a.* | $55.66 \pm 0.08$ | $41.38 \pm 0.04$ | *n.a.* |
| | DropEdge | $82.09 \pm 0.03$ | $11.40 \pm 0.03$ | $88.62 \pm 0.02$ | $82.79 \pm 0.02$ | $75.35 \pm 0.06$ | *n.a.* | $56.77 \pm 0.02$ | $43.94 \pm 0.07$ | *n.a.* |
| | VGCN-Energy | $81.66 \pm 0.04$ | $9.90 \pm 0.03$ | $88.95 \pm 0.02$ | $82.76 \pm 0.03$ | $83.43 \pm 0.04$ | *n.a.* | $57.49 \pm 0.06$ | $60.64 \pm 0.10$ | *n.a.* |
| | VGCN-Ensemble | $81.66 \pm 0.04$ | $9.96 \pm 0.02$ | **89.00** $\pm 0.02$ | $82.80 \pm 0.02$ | $83.77 \pm 0.55$ | *n.a.* | $57.46 \pm 0.04$ | $57.62 \pm 0.84$ | *n.a.* |
| | VGCN-BNN | $68.20 \pm 0.15$ | $20.27 \pm 0.45$ | $79.65 \pm 0.06$ | $82.16 \pm 0.24$ | $69.72 \pm 2.37$ | *n.a.* | $58.10 \pm 0.53$ | $52.08 \pm 2.48$ | *n.a.* |
| | GKDE-GCN | $65.90 \pm 0.53$ | **9.04** $\pm 0.28$ | $82.73 \pm 0.37$ | $77.03 \pm 0.34$ | $70.32 \pm 0.66$ | *n.a.* | $49.81 \pm 0.44$ | $45.92 \pm 0.90$ | *n.a.* |
| | GPN | **82.10** $\pm 0.29$ | $9.22 \pm 0.19$ | $88.48 \pm 0.07$ | $82.49 \pm 0.17$ | **87.63** $\pm 0.18$ | $74.55 \pm 0.24$ | $56.78 \pm 0.38$ | **67.94** $\pm 0.28$ | $48.03 \pm 0.34$ |
| Amazon Photos | LP | $85.88 \pm 0.00$ | $73.38 \pm 0.00$ | $89.27 \pm 0.00$ | **94.24** $\pm 0.00$ | $90.26 \pm 0.00$ | *n.a.* | **90.24** $\pm 0.00$ | $85.55 \pm 0.00$ | *n.a.* |
| | GKDE | $75.38 \pm 0.00$ | $52.21 \pm 0.00$ | $85.94 \pm 0.00$ | $76.51 \pm 0.00$ | $60.83 \pm 0.00$ | *n.a.* | $66.72 \pm 0.00$ | $59.09 \pm 0.00$ | *n.a.* |
| | Matern-GGP | $86.10 \pm 0.01$ | **8.54** $\pm 0.02$ | $88.65 \pm 0.00$ | $87.26 \pm 0.00$ | $86.75 \pm 0.00$ | *n.a.* | $75.22 \pm 0.00$ | $74.39 \pm 0.03$ | *n.a.* |
| | GGP | *n.f.* | *n.f.* | *n.f.* | *n.f.* | *n.f.* | *n.f.* | *n.f.* | *n.f.* | *n.f.* |
| | APPNP | **92.12** $\pm 0.01$ | $13.68 \pm 0.02$ | **95.42** $\pm 0.01$ | $77.45 \pm 0.10$ | *n.a.* | *n.a.* | $67.50 \pm 0.12$ | *n.a.* | *n.a.* |
| | VGCN | $90.95 \pm 0.01$ | $10.37 \pm 0.03$ | $94.24 \pm 0.01$ | $82.44 \pm 0.07$ | *n.a.* | *n.a.* | $72.60 \pm 0.11$ | *n.a.* | *n.a.* |
| | RGCN | $81.00 \pm 0.42$ | $40.20 \pm 0.25$ | $87.59 \pm 0.69$ | $75.25 \pm 0.23$ | *n.a.* | *n.a.* | $67.53 \pm 0.29$ | *n.a.* | *n.a.* |
| | VGCN-Dropout | $90.42 \pm 0.01$ | $12.76 \pm 0.05$ | $94.04 \pm 0.01$ | $80.90 \pm 0.08$ | $70.11 \pm 0.14$ | *n.a.* | $70.55 \pm 0.12$ | $53.16 \pm 0.13$ | *n.a.* |
| | DropEdge | $91.03 \pm 0.01$ | $12.25 \pm 0.02$ | $94.22 \pm 0.01$ | $82.48 \pm 0.09$ | $71.67 \pm 0.11$ | *n.a.* | $72.75 \pm 0.15$ | $54.98 \pm 0.10$ | *n.a.* |
| | VGCN-Energy | $90.95 \pm 0.01$ | $10.37 \pm 0.03$ | $94.24 \pm 0.01$ | $82.44 \pm 0.07$ | $79.64 \pm 0.07$ | *n.a.* | $72.60 \pm 0.11$ | $71.71 \pm 0.14$ | *n.a.* |
| | VGCN-Ensemble | $90.94 \pm 0.01$ | $10.38 \pm 0.02$ | $94.28 \pm 0.01$ | $82.72 \pm 0.03$ | $88.53 \pm 0.39$ | *n.a.* | $72.98 \pm 0.08$ | $83.28 \pm 0.46$ | *n.a.* |
| | VGCN-BNN | $84.51 \pm 0.23$ | $29.42 \pm 0.32$ | $91.88 \pm 0.19$ | $72.03 \pm 0.38$ | $64.10 \pm 1.98$ | *n.a.* | $62.85 \pm 0.49$ | $54.09 \pm 1.76$ | *n.a.* |
| | GKDE-GCN | $79.74 \pm 0.99$ | $12.84 \pm 0.31$ | $89.84 \pm 0.73$ | $73.65 \pm 1.13$ | $69.09 \pm 0.81$ | *n.a.* | $62.45 \pm 1.20$ | $59.68 \pm 0.75$ | *n.a.* |
| | GPN | $90.44 \pm 0.07$ | $11.95 \pm 0.12$ | $94.01 \pm 0.07$ | $82.72 \pm 0.24$ | $91.98 \pm 0.22$ | $76.57 \pm 0.49$ | $74.55 \pm 0.39$ | $86.29 \pm 0.35$ | $64.00 \pm 0.68$ |
| Coauthor CS | LP | $83.34 \pm 0.00$ | $76.67 \pm 0.00$ | $82.89 \pm 0.00$ | $86.64 \pm 0.00$ | $86.40 \pm 0.00$ | *n.a.* | $79.05 \pm 0.00$ | $79.56 \pm 0.00$ | *n.a.* |
| | GKDE | $79.27 \pm 0.00$ | $70.74 \pm 0.00$ | $78.84 \pm 0.00$ | $79.32 \pm 0.00$ | $77.59 \pm 0.00$ | *n.a.* | $66.30 \pm 0.00$ | $64.69 \pm 0.00$ | *n.a.* |
| | Matern-GGP | $83.56 \pm 0.01$ | **6.18** $\pm 0.00$ | $83.21 \pm 0.00$ | $73.57 \pm 0.00$ | $73.75 \pm 0.00$ | *n.a.* | $62.05 \pm 0.00$ | $61.74 \pm 0.00$ | *n.a.* |
| | GGP | *n.f.* | *n.f.* | *n.f.* | *n.f.* | *n.f.* | *n.f.* | *n.f.* | *n.f.* | *n.f.* |
| | APPNP | **92.96** $\pm 0.01$ | $8.70 \pm 0.02$ | **93.51** $\pm 0.01$ | $81.88 \pm 0.03$ | *n.a.* | *n.a.* | $75.85 \pm 0.06$ | *n.a.* | *n.a.* |
| | VGCN | $92.61 \pm 0.01$ | $7.00 \pm 0.03$ | $93.07 \pm 0.01$ | $85.35 \pm 0.03$ | *n.a.* | *n.a.* | $80.87 \pm 0.06$ | *n.a.* | *n.a.* |
| | RGCN | $92.02 \pm 0.02$ | $11.52 \pm 0.06$ | $92.49 \pm 0.01$ | $77.00 \pm 0.08$ | *n.a.* | *n.a.* | $71.87 \pm 0.11$ | *n.a.* | *n.a.* |
| | VGCN-Dropout | $92.30 \pm 0.01$ | $12.41 \pm 0.05$ | $92.83 \pm 0.01$ | $84.04 \pm 0.03$ | $73.09 \pm 0.10$ | *n.a.* | $78.99 \pm 0.07$ | $55.79 \pm 0.14$ | *n.a.* |
| | DropEdge | $92.63 \pm 0.01$ | $11.40 \pm 0.03$ | $92.92 \pm 0.01$ | $84.62 \pm 0.06$ | $75.68 \pm 0.08$ | *n.a.* | $79.74 \pm 0.08$ | $58.85 \pm 0.11$ | *n.a.* |
| | VGCN-Energy | $92.61 \pm 0.01$ | $7.00 \pm 0.03$ | $93.07 \pm 0.01$ | $85.35 \pm 0.03$ | $87.33 \pm 0.04$ | *n.a.* | $80.87 \pm 0.06$ | $82.79 \pm 0.11$ | *n.a.* |
| | VGCN-Ensemble | $92.66 \pm 0.01$ | $7.03 \pm 0.02$ | $93.07 \pm 0.00$ | $85.43 \pm 0.00$ | $83.19 \pm 0.07$ | *n.a.* | $80.88 \pm 0.01$ | $72.27 \pm 0.14$ | *n.a.* |
| | VGCN-BNN | $92.21 \pm 0.04$ | $12.15 \pm 0.10$ | $92.54 \pm 0.04$ | $80.48 \pm 0.25$ | $70.75 \pm 0.65$ | *n.a.* | $73.64 \pm 0.35$ | $54.41 \pm 0.76$ | *n.a.* |
| | GKDE-GCN | $92.35 \pm 0.09$ | $8.04 \pm 0.11$ | $93.13 \pm 0.01$ | $85.02 \pm 0.03$ | $84.45 \pm 0.06$ | *n.a.* | $80.15 \pm 0.07$ | $77.90 \pm 0.12$ | *n.a.* |
| | GPN | $86.88 \pm 0.10$ | $18.92 \pm 0.11$ | $88.21 \pm 0.10$ | $69.49 \pm 0.39$ | **92.09** $\pm 0.20$ | $88.84 \pm 0.31$ | $55.41 \pm 0.46$ | **90.28** $\pm 0.18$ | $86.54 \pm 0.42$ |
| Coauthor Physics | LP | $90.75 \pm 0.00$ | $70.75 \pm 0.00$ | $95.39 \pm 0.00$ | $91.78 \pm 0.00$ | $90.03 \pm 0.00$ | *n.a.* | $70.58 \pm 0.00$ | $69.63 \pm 0.00$ | *n.a.* |
| | GKDE | $87.75 \pm 0.00$ | $56.04 \pm 0.00$ | $93.30 \pm 0.00$ | $87.02 \pm 0.00$ | $84.64 \pm 0.00$ | *n.a.* | $57.00 \pm 0.00$ | $52.49 \pm 0.00$ | *n.a.* |
| | Matern-GGP | *n.f.* | *n.f.* | *n.f.* | *n.f.* | *n.f.* | *n.f.* | *n.f.* | *n.f.* | *n.f.* |
| | GGP | *n.f.* | *n.f.* | *n.f.* | *n.f.* | *n.f.* | *n.f.* | *n.f.* | *n.f.* | *n.f.* |
| | APPNP | $95.56 \pm 0.00$ | $1.57 \pm 0.01$ | **97.96** $\pm 0.00$ | $90.37 \pm 0.02$ | *n.a.* | *n.a.* | $61.46 \pm 0.05$ | *n.a.* | *n.a.* |
| | VGCN | $95.59 \pm 0.00$ | **1.28** $\pm 0.02$ | **97.96** $\pm 0.00$ | $90.29 \pm 0.02$ | *n.a.* | *n.a.* | $63.63 \pm 0.09$ | *n.a.* | *n.a.* |
| | RGCN | $95.33 \pm 0.00$ | $5.11 \pm 0.07$ | $97.92 \pm 0.00$ | $75.62 \pm 0.13$ | *n.a.* | *n.a.* | $56.27 \pm 0.13$ | *n.a.* | *n.a.* |
| | VGCN-Dropout | $95.51 \pm 0.00$ | $3.07 \pm 0.02$ | $97.92 \pm 0.00$ | $89.63 \pm 0.03$ | $87.10 \pm 0.04$ | *n.a.* | $62.53 \pm 0.10$ | $51.19 \pm 0.12$ | *n.a.* |
| | DropEdge | $95.59 \pm 0.00$ | $2.81 \pm 0.01$ | $97.92 \pm 0.00$ | $90.56 \pm 0.02$ | $88.76 \pm 0.04$ | *n.a.* | $64.59 \pm 0.08$ | $55.32 \pm 0.14$ | *n.a.* |
| | VGCN-Energy | $95.59 \pm 0.00$ | $1.28 \pm 0.02$ | $97.96 \pm 0.00$ | $90.29 \pm 0.02$ | $91.08 \pm 0.05$ | *n.a.* | $63.63 \pm 0.09$ | $69.41 \pm 0.11$ | *n.a.* |
| | VGCN-Ensemble | $95.58 \pm 0.00$ | $1.29 \pm 0.01$ | $97.96 \pm 0.00$ | $90.35 \pm 0.00$ | $92.39 \pm 0.00$ | *n.a.* | $63.67 \pm 0.00$ | $71.30 \pm 0.02$ | *n.a.* |
| | VGCN-BNN | $95.46 \pm 0.02$ | $5.64 \pm 0.09$ | $97.94 \pm 0.01$ | $90.73 \pm 0.21$ | $90.09 \pm 0.50$ | *n.a.* | $66.98 \pm 0.32$ | $61.27 \pm 1.47$ | *n.a.* |
| | GKDE-GCN | **95.61** $\pm 0.00$ | $1.51 \pm 0.02$ | $97.95 \pm 0.00$ | $87.38 \pm 0.09$ | $84.62 \pm 0.19$ | *n.a.* | $57.97 \pm 0.22$ | $56.30 \pm 0.51$ | *n.a.* |
| | GPN(16) | $94.32 \pm 0.02$ | $10.61 \pm 0.03$ | $97.40 \pm 0.01$ | $85.20 \pm 0.17$ | **94.51** $\pm 0.15$ | $89.63 \pm 0.24$ | $61.89 \pm 0.20$ | **83.73** $\pm 0.31$ | $66.44 \pm 0.65$ |
| OGBN Arxiv | LP | $64.27$ | $61.77$ | $66.84$ | **80.04** | $75.22$ | *n.a.* | $65.21$ | **67.69** | *n.a.* |
| | GKDE | $48.87$ | $22.59$ | $51.51$ | $68.12$ | $65.80$ | *n.a.* | $47.22$ | $45.23$ | *n.a.* |
| | Matern-GGP | *n.f.* | *n.f.* | *n.f.* | *n.f.* | *n.f.* | *n.f.* | *n.f.* | *n.f.* | *n.f.* |
| | GGP | *n.f.* | *n.f.* | *n.f.* | *n.f.* | *n.f.* | *n.f.* | *n.f.* | *n.f.* | *n.f.* |
| | APPNP | $71.46 \pm 0.09$ | $3.96 \pm 0.06$ | $75.47 \pm 0.15$ | $65.21 \pm 0.14$ | *n.a.* | *n.a.* | $43.23 \pm 0.06$ | *n.a.* | *n.a.* |
| | VGCN | $71.89 \pm 0.08$ | $2.18 \pm 0.10$ | $75.61 \pm 0.11$ | $64.91 \pm 0.28$ | *n.a.* | *n.a.* | $42.72 \pm 0.32$ | *n.a.* | *n.a.* |
| | VGCN-Dropout | $71.76 \pm 0.07$ | $2.29 \pm 0.08$ | $75.47 \pm 0.12$ | $65.35 \pm 0.27$ | $64.24 \pm 0.26$ | *n.a.* | $43.09 \pm 0.30$ | $41.58 \pm 0.23$ | *n.a.* |
| | VGCN-Energy | $71.89 \pm 0.08$ | $2.18 \pm 0.10$ | $75.61 \pm 0.11$ | $64.91 \pm 0.28$ | $64.50 \pm 0.38$ | *n.a.* | $42.72 \pm 0.32$ | $42.41 \pm 0.39$ | *n.a.* |
| | VGCN-Ensemble | **72.59** | **2.10** | **76.12** | $65.93$ | $70.77$ | *n.a.* | $43.84$ | $50.63$ | *n.a.* |
| | VGCN-BNN | *n.a.* | *n.a.* | *n.a.* | *n.a.* | *n.a.* | *n.a.* | *n.a.* | *n.a.* | *n.a.* |
| | GKDE-GCN | $68.99 \pm 0.40$ | $8.43 \pm 0.43$ | $73.89 \pm 0.33$ | $68.84 \pm 0.18$ | $72.44 \pm 0.50$ | *n.a.* | $49.71 \pm 0.59$ | $52.23 \pm 0.66$ | *n.a.* |
| | GPN | $69.08 \pm 0.21$ | $6.96 \pm 0.31$ | $73.84 \pm 0.21$ | $66.33 \pm 0.23$ | $74.82 \pm 0.27$ | $62.17 \pm 0.23$ | $46.35 \pm 0.26$ | $58.71 \pm 0.34$ | $43.01 \pm 0.36$ |

Table 15: Accuracy and ECE scores on the clean graphs. Accuracy and OOD detection scores on Left-Out classes using AUC-ROC and AUC-PR scores. OOD-AUC-ROC and OOD-AUC-APR scores are given as *[Alea w/ Net] / [Epist w/ Net] / [Epist w/o Net]*. *n.a.* means either model or metric not applicable and *n.f.* means not finished within our constraints. Bold numbers indicate best results for Accuracy, ECE and OOD detection.

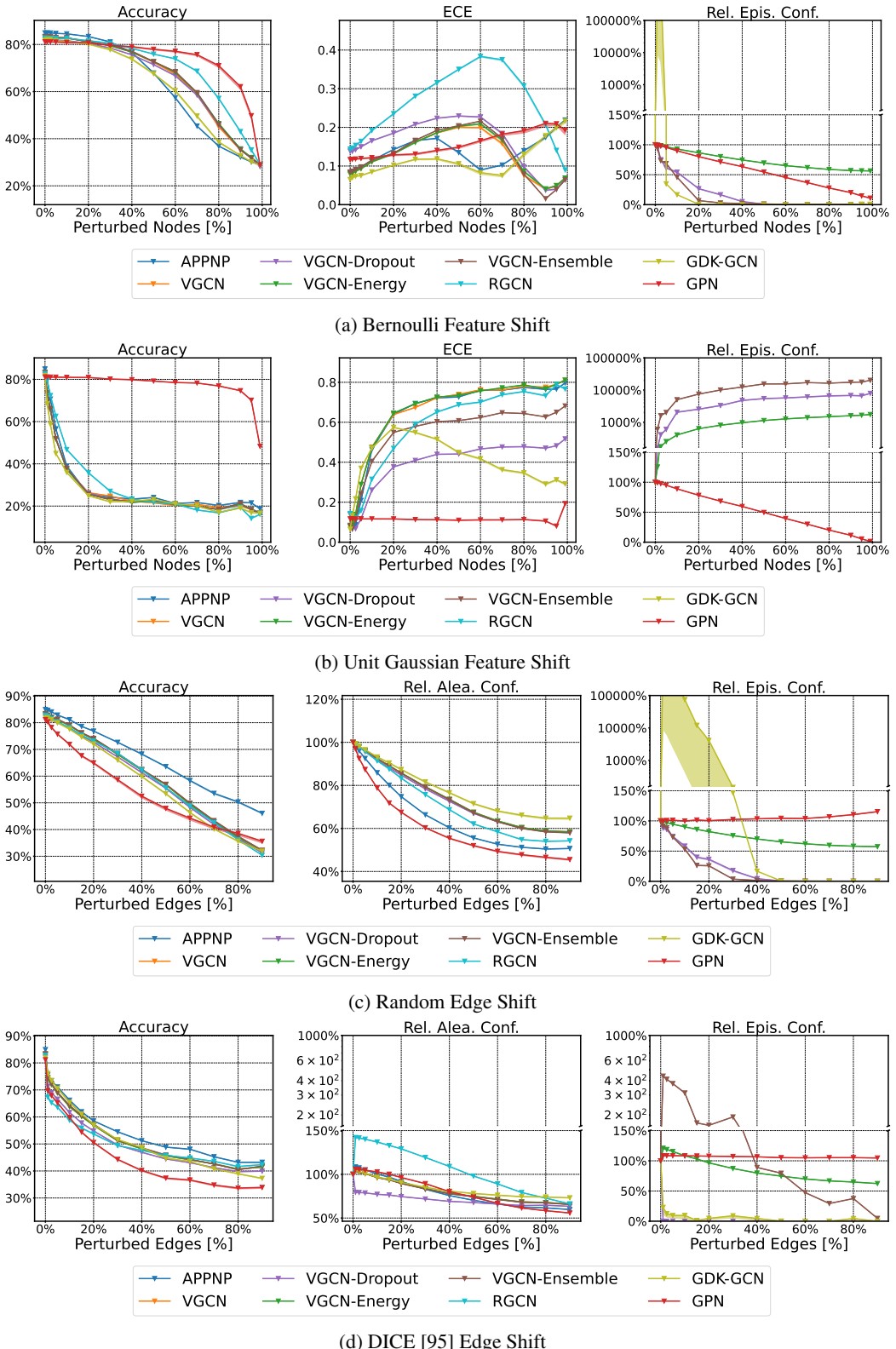

Figure 5: Relative performance over different degrees of corruption of *CoraML*. For feature shifts, we perturb different fractions of nodes (whose features are replaced with either random vectors from a Bernoulli noise or a Unit Gaussian noise) and show accuracy, ECE, and relative average epistemic confidence. For edge shifts, we perturb different fractions of edges (by replacing them at random or using the global and untargeted DICE [95] attack) and show accuracy, relative average aleatoric confidence, and relative average epistemic confidence.

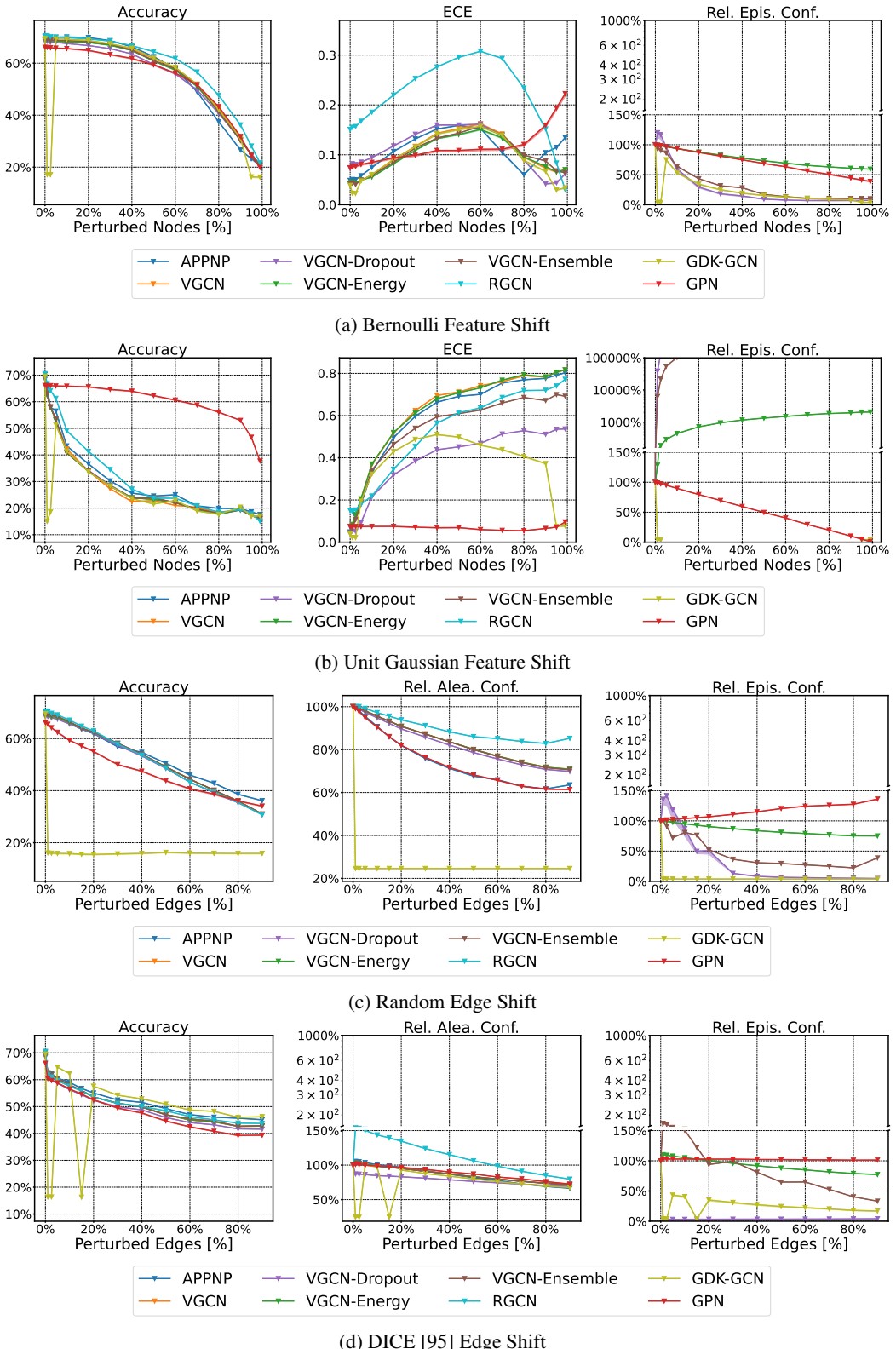

Figure 6: Relative performance over different degrees of corruption of *CiteSeer*. For feature shifts, we perturb different fractions of nodes (whose features are replaced with either random vectors from a Bernoulli noise or a Unit Gaussian noise) and show accuracy, ECE, and relative average epistemic confidence. For edge shifts, we perturb different fractions of edges (by replacing them at random or using the global and untargeted DICE [95] attack) and show accuracy, relative average aleatoric confidence, and relative average epistemic confidence.

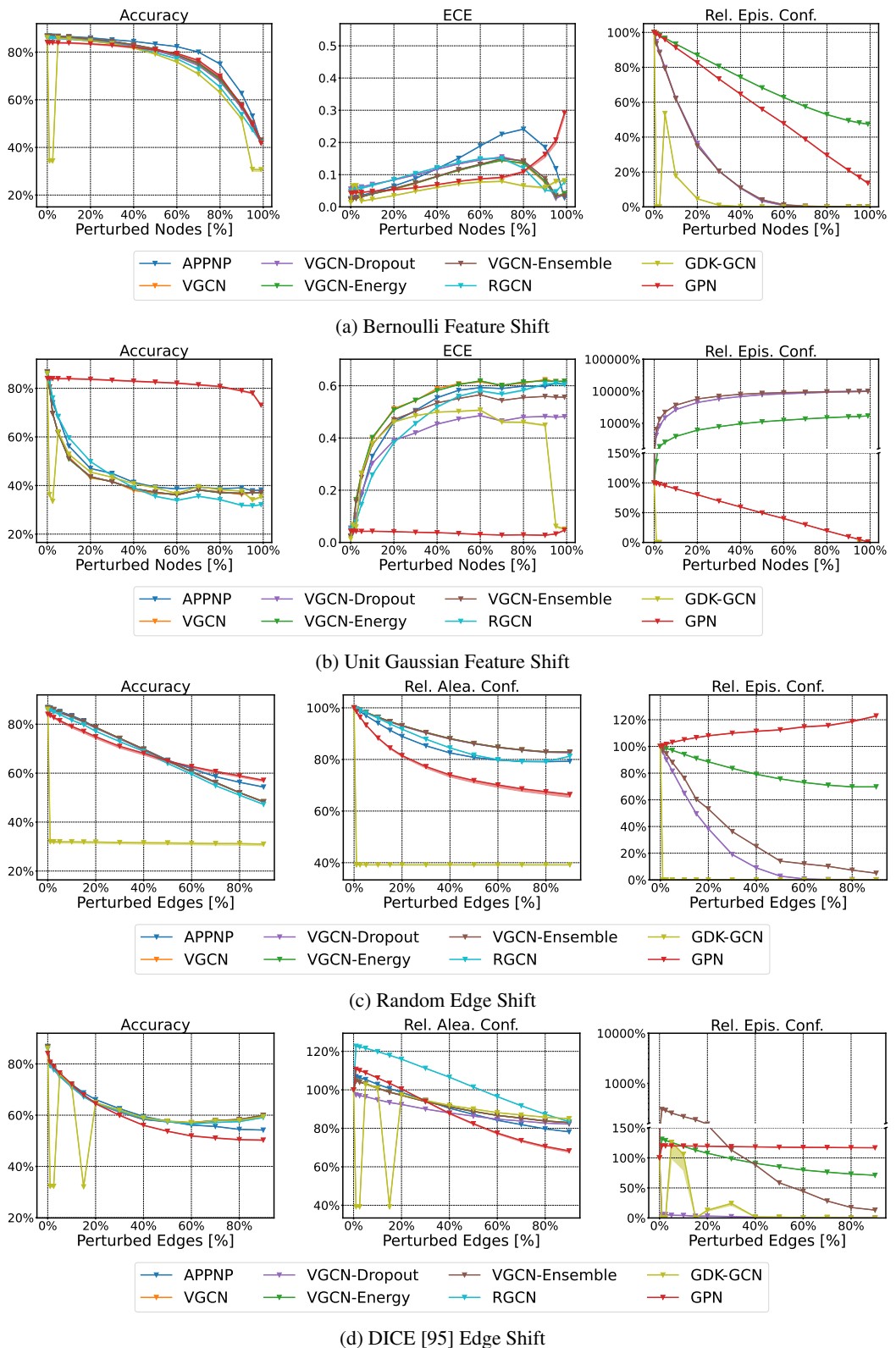

Figure 7: Relative performance over different degrees of corruption of *PubMed*. For feature shifts, we perturb different fractions of nodes (whose features are replaced with either random vectors from a Bernoulli noise or a Unit Gaussian noise) and show accuracy, ECE, and relative average epistemic confidence. For edge shifts, we perturb different fractions of edges (by replacing them at random or using the global and untargeted DICE [95] attack) and show accuracy, relative average aleatoric confidence, and relative average epistemic confidence.

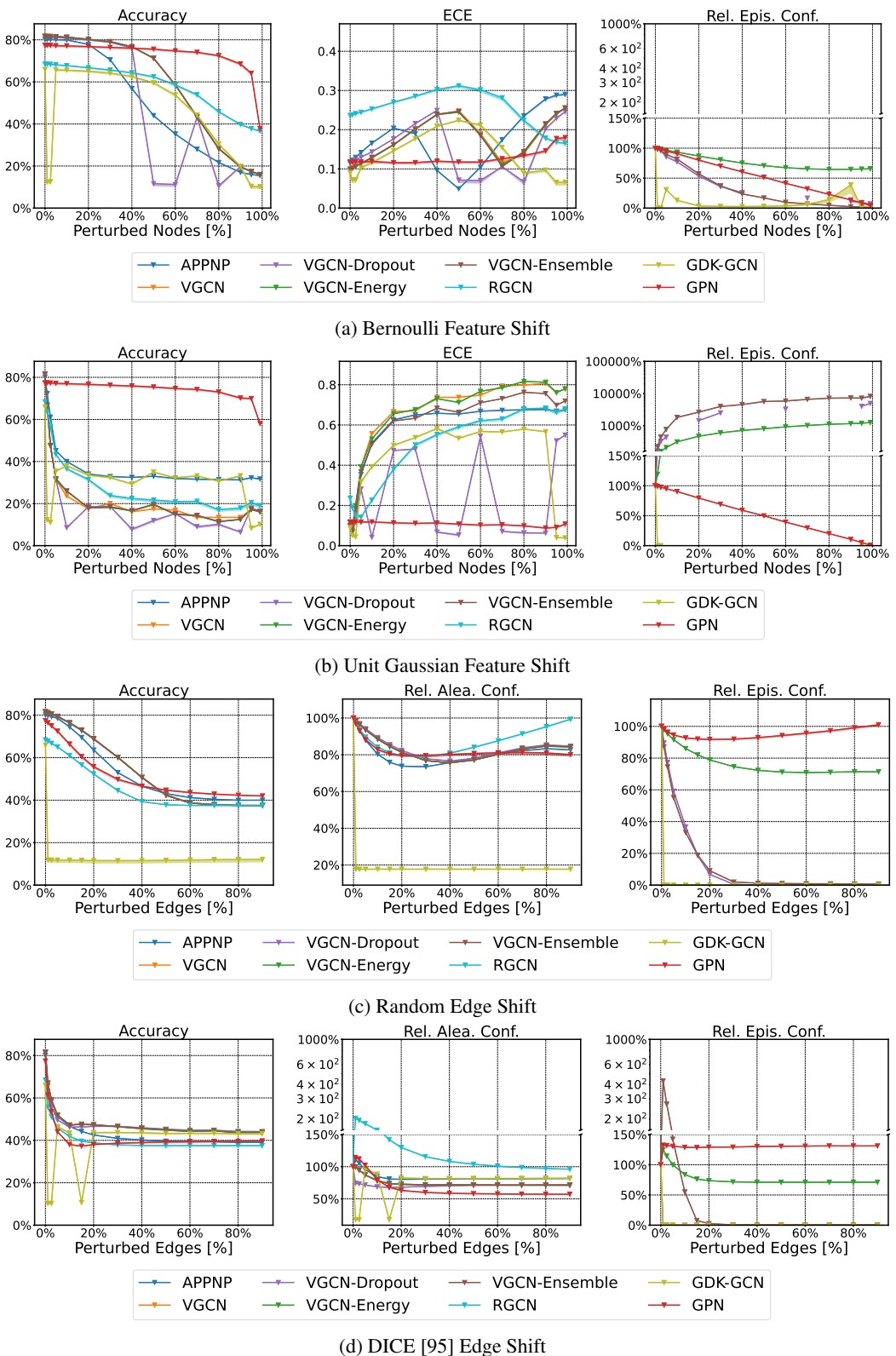

Figure 8: Relative performance over different degrees of corruption of *Amazon Computers*. For feature shifts, we perturb different fractions of nodes (whose features are replaced with either random vectors from a Bernoulli noise or a Unit Gaussian noise) and show accuracy, ECE, and relative average epistemic confidence. For edge shifts, we perturb different fractions of edges (by replacing them at random or using the global and untargeted DICE [95] attack) and show accuracy, relative average aleatoric confidence, and relative average epistemic confidence.

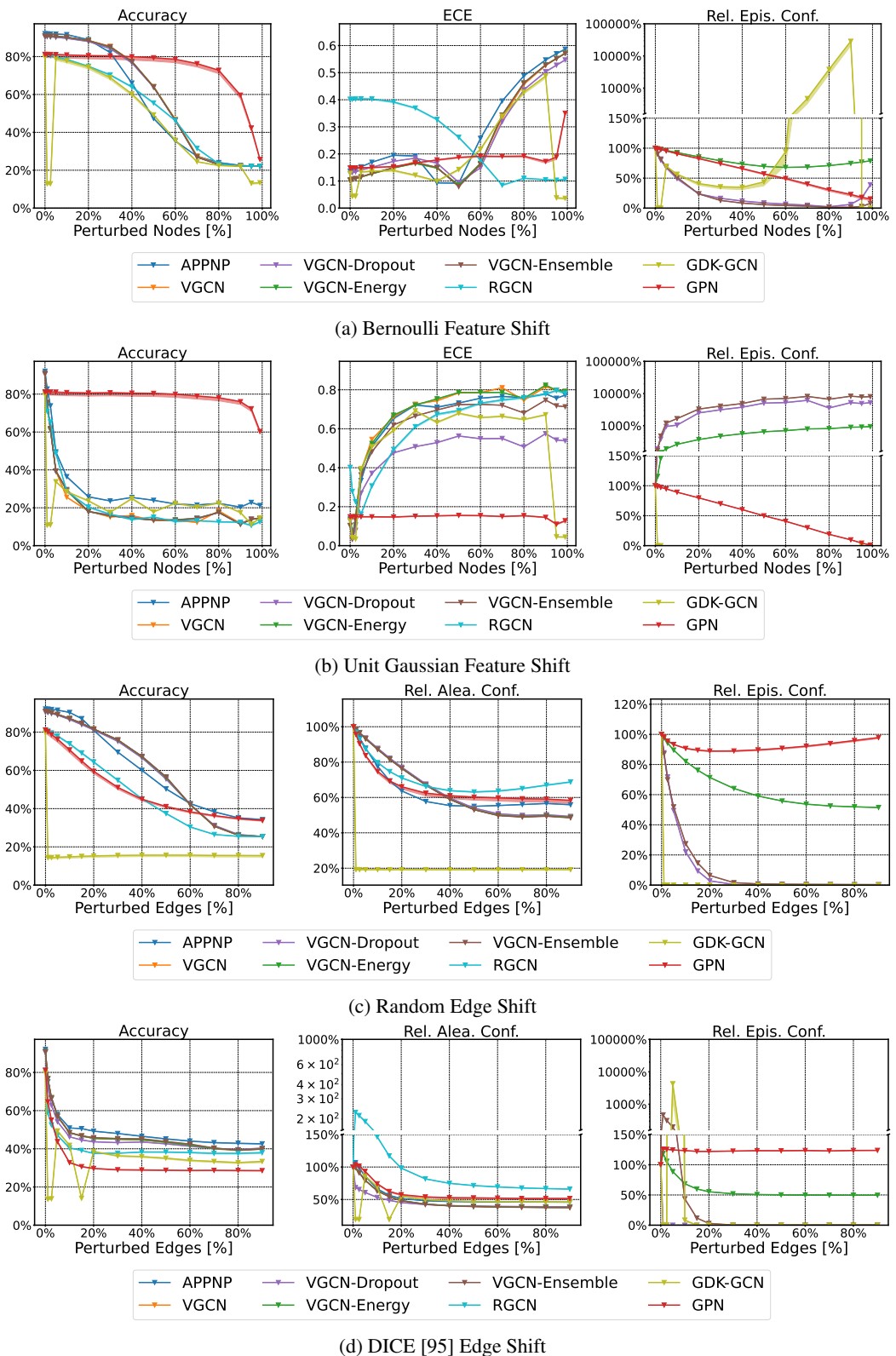

Figure 9: Relative performance over different degrees of corruption of *Amazon Photos*. For feature shifts, we perturb different fractions of nodes (whose features are replaced with either random vectors from a Bernoulli noise or a Unit Gaussian noise) and show accuracy, ECE, and relative average epistemic confidence. For edge shifts, we perturb different fractions of edges (by replacing them at random or using the global and untargeted DICE [95] attack) and show accuracy, relative average aleatoric confidence, and relative average epistemic confidence.

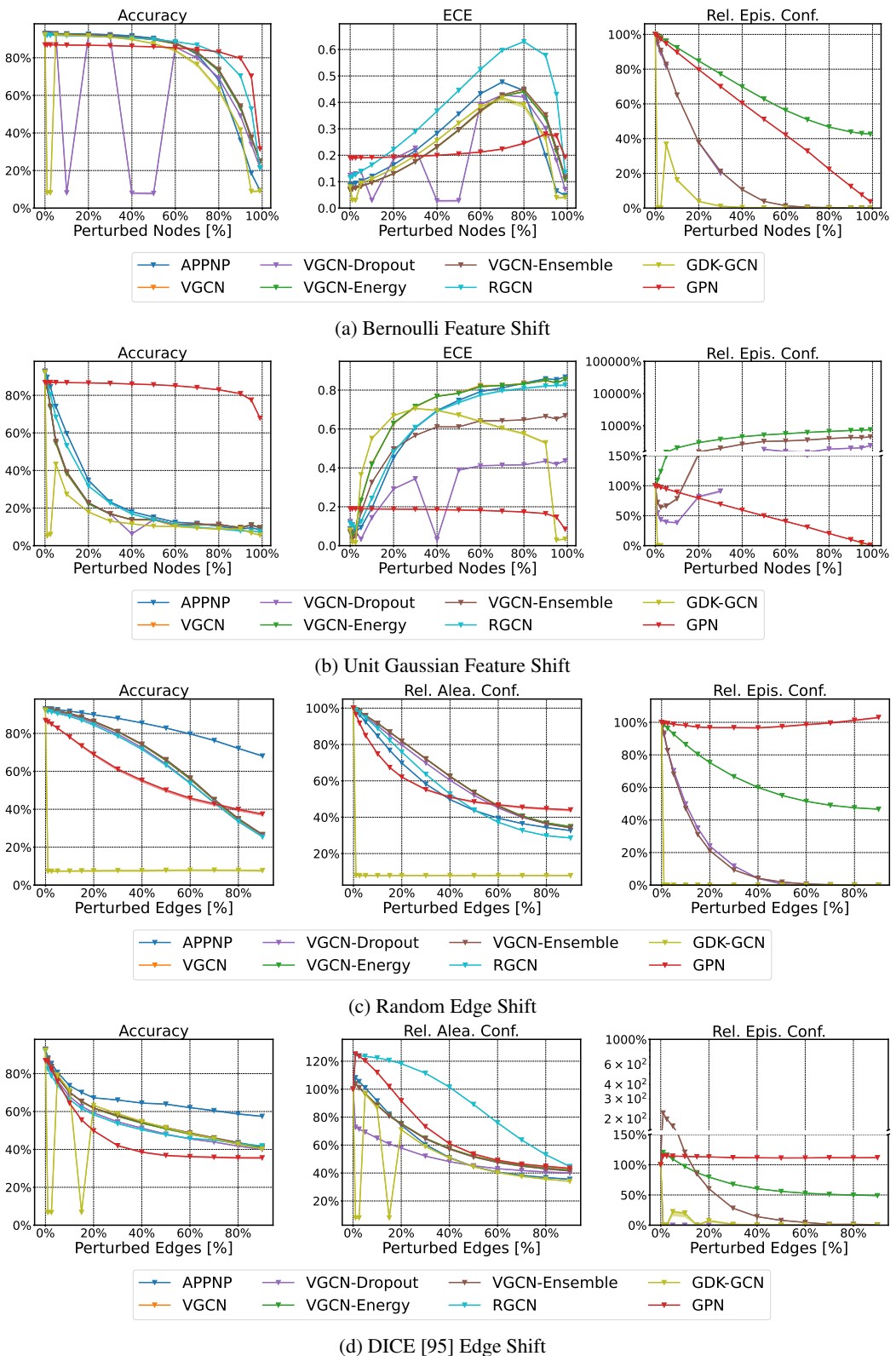

Figure 10: Relative performance over different degrees of corruption of *Coauthor CS*. For feature shifts, we perturb different fractions of nodes (whose features are replaced with either random vectors from a Bernoulli noise or a Unit Gaussian noise) and show accuracy, ECE, and relative average epistemic confidence. For edge shifts, we perturb different fractions of edges (by replacing them at random or using the global and untargeted DICE [95] attack) and show accuracy, relative average aleatoric confidence, and relative average epistemic confidence.

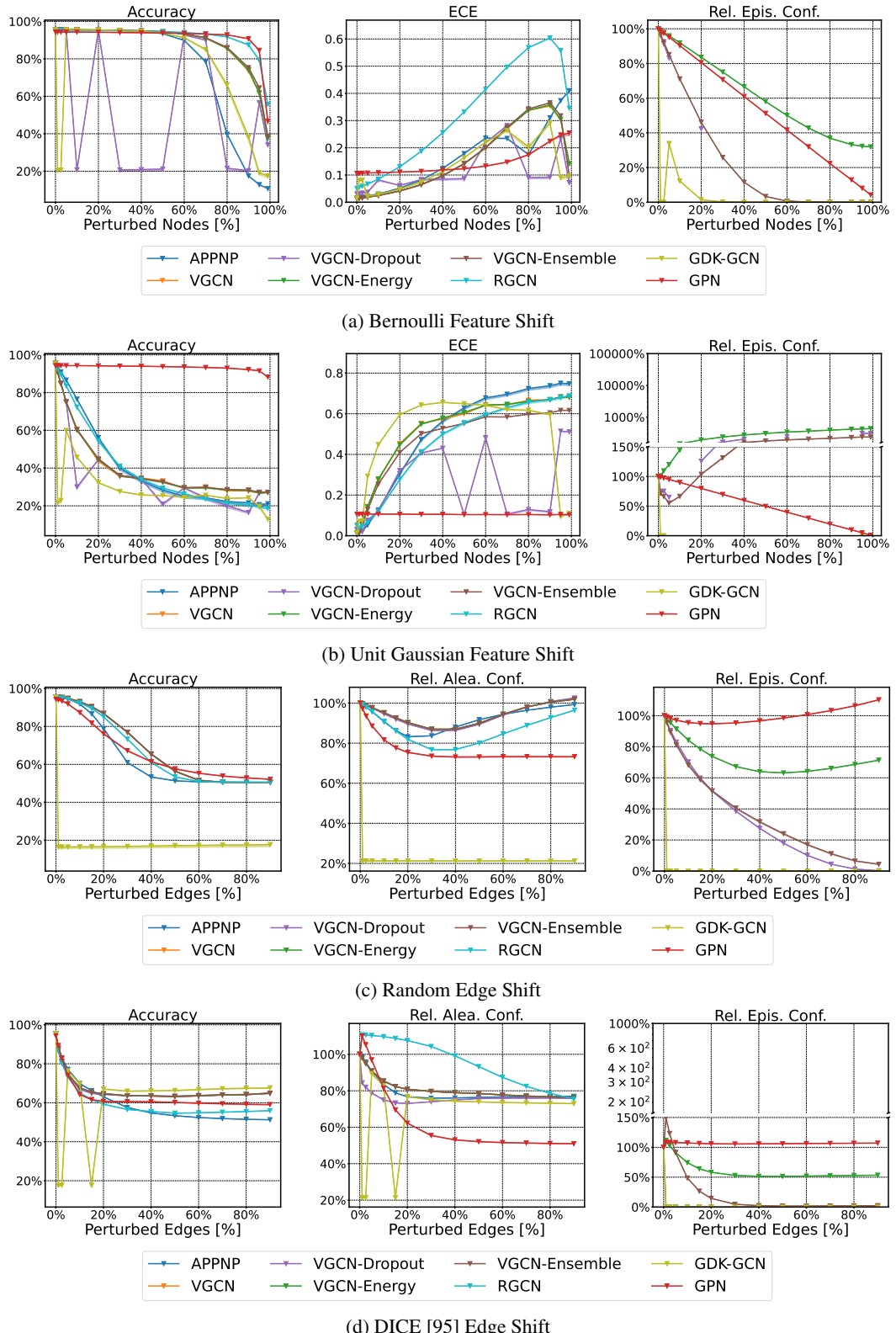

(a) Bernoulli Feature Shift

(b) Unit Gaussian Feature Shift

(c) Random Edge Shift

(d) DICE [95] Edge Shift

Figure 11: Relative performance over different degrees of corruption of *Coauthor Physics*. For feature shifts, we perturb different fractions of nodes (whose features are replaced with either random vectors from a Bernoulli noise or a Unit Gaussian noise) and show accuracy, ECE, and relative average epistemic confidence. For edge shifts, we perturb different fractions of edges (by replacing them at random or using the global and untargeted DICE [95] attack) and show accuracy, relative average aleatoric confidence, and relative average epistemic confidence.

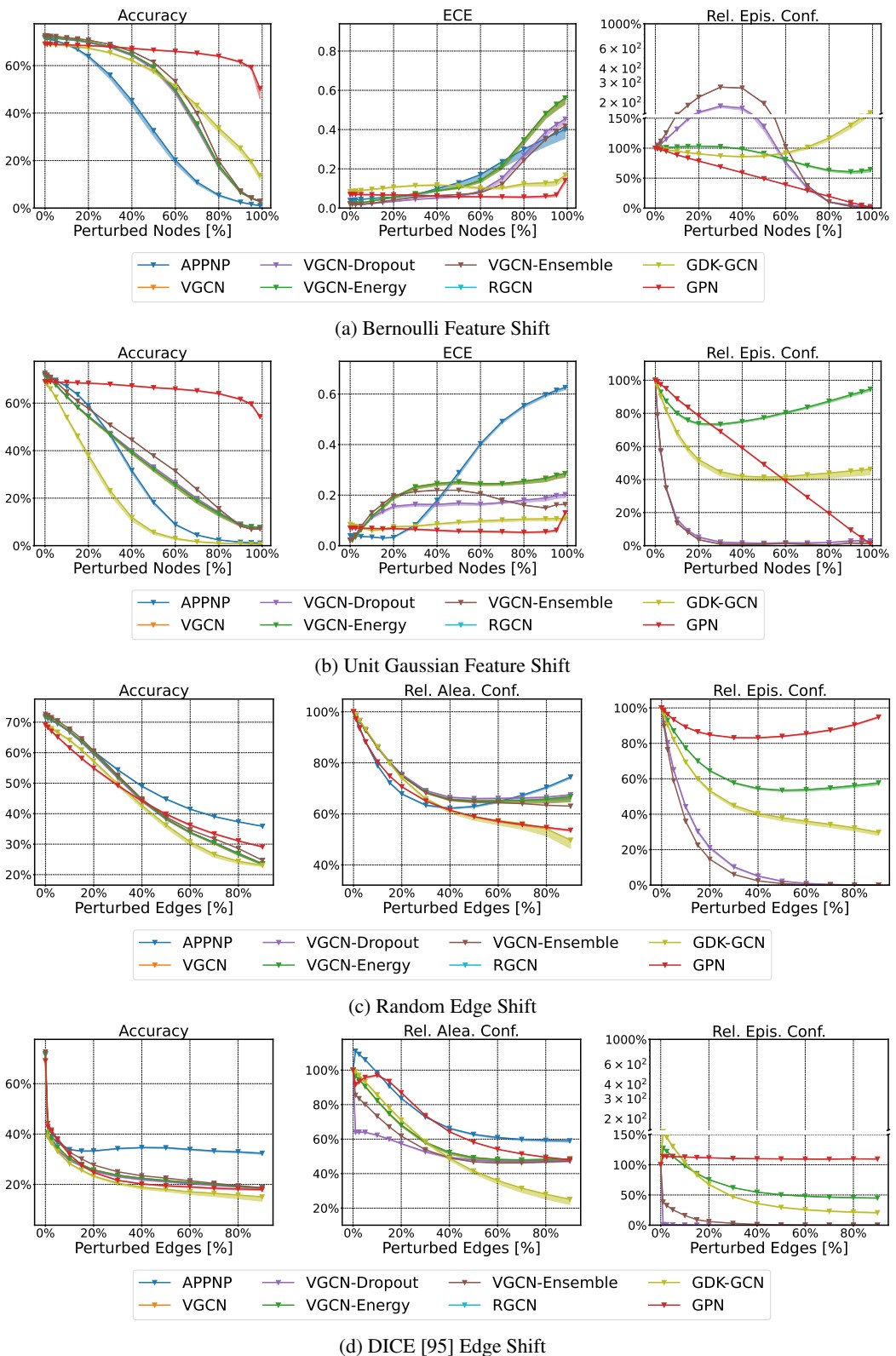

Figure 12: Relative performance over different degrees of corruption of *OGBN Arxiv*. For feature shifts, we perturb different fractions of nodes (whose features are replaced with either random vectors from a Bernoulli noise or a Unit Gaussian noise) and show accuracy, ECE, and relative average epistemic confidence. For edge shifts, we perturb different fractions of edges (by replacing them at random or using the global and untargeted DICE [95] attack) and show accuracy, relative average aleatoric confidence, and relative average epistemic confidence.

| Node | Node Representation |
|---|---|
| 220 | **Abstract:** `IlliGAL Report No. 95006 July 1995`
**Bag-of-Word:** `['1995', 'report', 'july']` |
| 197 | **Abstract:** `Report of the 1996 Workshop on Reinforcement`
**Bag-of-Word:** `['workshop', 'reinforcement', 'report', '1996']` |
| 193 | **Abstract:** `Report of the 1996 Workshop on Reinforcement`
**Bag-of-Word:** `['workshop', 'reinforcement', 'report', '1996']` |
| 258 | **Abstract:** `TIK-Report Nr. 11, December 1995 Version 2 (2. Edition)`
**Bag-of-Word:** `['version', '1995', 'report', '11']` |
| 2319 | **Abstract:** `    We tend to think of what we really know as what we can talk about, and disparage knowledge that we can't verbalize. [Dowling 1989, p. 252]`
**Bag-of-Word:** `['knowledge', 'know', '1989', 'tend']` |
| 1945 | **Abstract:** `Reihe FABEL-Report Status: extern Dokumentbezeichner: Org/Reports/nr-35 Erstellt am: 21.06.94 Korrigiert am: 28.05.95 ISSN 0942-413X`
**Bag-of-Word:** `['reports', '94', 'report', '95']` |
| 293 | **Abstract:** `Keith Mathias and Darrell Whitley Technical Report CS-94-101 January 7, 1994`
**Bag-of-Word:** `['technical', '1994', '94', 'cs', 'report', 'january']` |
| 99 | **Abstract:** `   Multigrid Q-Learning Charles W. Anderson and Stewart G. Crawford-Hines Technical Report CS-94-121 October 11, 1994`
**Bag-of-Word:** `['learning', 'technical', '1994', '94', 'cs', 'report', '11']` |
| 766 | **Abstract:** `Internal Report 97-01`
**Bag-of-Word:** `['internal', 'report', '97']` |
| 992 | **Abstract:** `A Learning Result for Abstract`
**Bag-of-Word:** `['learning', 'result', 'abstract']` |
| 2030 | **Abstract:** `    DRAFT March 16, 1998 available via anonymous ftp: site ftp.uwasa.fi directory cs/report94-1 file gaGPbib.ps.Z`
**Bag-of-Word:** `['available', '1998', 'cs', '16', 'anonymous', 'ftp', 'march', 'fi', 'site']` |
| 991 | **Abstract:** `    In IEEE Transactions on Neural Networks, 7(1):97-106, 1996 Also available as GMD report 794`
**Bag-of-Word:** `['available', 'networks', 'neural', 'report', 'ieee', '1996', '97']` |
| 74 | **Abstract:** `    Empirical Comparison of Gradient Descent and Exponentiated Gradient Descent in Supervised and Reinforcement Learning Technical Report 96-70`
**Bag-of-Words:** `['learning', 'technical', 'empirical', 'gradient', 'comparison', 'reinforcement', 'supervised', 'report', 'descent', '96']` |
| 865 | **Abstract:** `  fl Partially supported by the Advanced Research Projects Agency (AFOSR 90-0083). y Partially supported by the Air Force Office of Scientific Research (AFOSR F49620-92-J-0499), the Advanced Research Projects Agency (ONR N00014-92-J-4015), and the Office of Naval Research (ONR N00014-91-J-4100). z Partially funded by the Air Force Office of Scientific Research (AFOSR F49620-92-J-0334) and the Office of Naval Research (ONR N00014-91-J-4100 and ONR N00014-94-1-0597).`
**Bag-of-Words:** `['fl', 'research', '90', 'partially', '94', 'advanced', 'projects', 'agency', 'supported', 'n00014', 'office', 'naval', 'force', 'air', 'scientific', 'afosr', '91', '92', 'onr', 'funded']` |
| 858 | **Abstract:** `Technical Report UMIACS-TR-97-77 and CS-TR-3843 Abstract`
**Bag-of-Words:** `['technical', 'abstract', 'cs', 'report', '97', 'tr']` |
| 1944 | **Abstract:** `Reihe FABEL-Report Status: extern Dokumentbezeichner: Org/Reports/nr-13 Erstellt am: 22.12.93 Korrigiert am: 02.02.94 ISSN 0942-413X`
**Bag-of-Words:** `['reports', '13', '94', 'report', '22', '93', '12']` |
| 1710 | **Abstract:** `Knowledge Systems Laboratory March 1995 Report No. KSL 95-32`
**Bag-of-Words:** `['systems', 'knowledge', 'laboratory', '1995', 'report', 'march', '95']` |
| 2491 | **Abstract:** `Edward S. Orosz and Charles W. Anderson Technical Report CS-94-111 April 27, 1994`
**Bag-of-Words:** `['technical', '1994', '94', 'cs', 'report', 'april']` |
| 221 | **Abstract:** `V. Scott Gordon and Darrell Whitley Technical Report CS-93-114 September 16, 1993`
**Bag-of-Words:** `['technical', '1993', 'cs', 'report', '16', '93']` |
| 1874 | **Abstract:** `COINS Technical Report 94-61 August 1994`
**Bag-of-Words:** `['technical', '1994', '94', 'report', 'august']` |

Table 16: 20 abstracts and their representation from the CoraML dataset obtained after selecting the abstracts GPN has assigned the lowest feature evidence. Abstracts sorted in ascending order of feature evidence.

| Node | Node Representation |
|------|---------------------|
| 1637 | **Abstract:** A pervasive, yet much ignored, factor in the analysis of processing-failures is the problem of misorganized knowledge. If a systems knowledge is not indexed or organized correctly, it may make an error, not because it does not have either the general capability or specific knowledge to solve a problem, but rather because it does not have the knowledge sufficiently organized so that the appropriate knowledge structures are brought to bear on the problem at the appropriate time. In such cases, the system can be said to have forgotten the knowledge, if only in this context. This is the problem of forgetting or retrieval failure. This research presents an analysis along with a declarative representation of a number of types of forgetting errors. Such representations can extend the capability of introspective failure-driven learning systems, allowing them to reduce the likelihood of repeating such errors. Examples are presented from the Meta-AQUA program, which learns to improve its performance on a story understanding task through an introspective meta-analysis of its knowledge, its organization of its knowledge, and its reasoning processes. 
 **Bag-of-Word** ['retrieval', 'number', 'problem', 'learning', 'systems', 'representations', 'knowledge', 'representation', 'understanding', 'program', 'learns', 'time', 'make', 'examples', 'appropriate', 'general', 'error', 'analysis', 'presented', 'cases', 'performance', 'correctly', 'task', 'presents', 'improve', 'reasoning', 'likelihood', 'research', 'driven', 'processes', 'does', 'reduce', 'processing', 'organization', 'extend', 'solve', 'specific', 'meta', 'factor', 'declarative', 'failure', 'capability', 'context', 'allowing', 'structures', 'types', 'ignored', 'failures', 'story', 'errors', 'sufficiently', 'brought', 'organized', 'introspective', 'bear'] |
| 1375 | **Abstract:** This paper describes an interactive planning system that was developed inside an Intelligent Decision Support System aimed at supporting an operator when planning the initial attack to forest fires. The planning architecture rests on the integration of case-based reasoning techniques with constraint reasoning techniques exploited, mainly, for performing temporal reasoning on temporal metric information. Temporal reasoning plays a central role in supporting interactive functions that are provided to the user when performing two basic steps of the planning process: plan adaptation and resource scheduling. A first prototype was integrated with a situation assessment and a resource allocation manager subsystem and is currently being tested. 
 **Bag-of-Word** ['intelligent', 'information', 'paper', 'planning', 'based', 'case', 'integrated', 'functions', 'describes', 'decision', 'allocation', 'architecture', 'support', 'reasoning', 'mainly', 'techniques', 'process', 'supporting', 'role', 'integration', 'exploited', 'currently', 'initial', 'user', 'tested', 'interactive', 'basic', 'developed', 'temporal', 'central', 'assessment', 'situation', 'steps', 'adaptation', 'operator', 'scheduling', 'constraint', 'performing', 'plays', 'provided', 'resource', 'prototype', 'metric', 'aimed', 'plan', 'inside'] |
| 2672 | **Abstract:** Inductive learning systems are designed to induce hypotheses, or general descriptions of concepts, from instances of these concepts. Among the wide variety of techniques used in inductive learning systems, algorithms derived from nearest neighbour (NN) pattern classification have been receiving attention lately, mainly due to their incremental nature. Nested Generalized Exemplar (NGE) theory is an inductive learning theory which can be viewed as descent from nearest neighbour classification. In NGE theory, the induced concepts take the form of hyperrectangles in a n-dimensional Euclidean space. The axes of the space are defined by the attributes used for describing the examples. This paper proposes a fuzzified version of the original NGE algorithm, which accepts input examples given as feature/fuzzy value pairs, and generalizes them as fuzzy hyperrectangles. It presents and discusses a metric for evaluating the fuzzy distance between examples, and between example and fuzzy hyperrectangles; criteria for establishing the reliability of fuzzy examples, by strengthening the exemplar which makes the right prediction and weakening the exemplar which makes a wrong one and criteria for producing fuzzy generalizations, based on the union of fuzzy sets. Keywords : exemplar-based learning, nested generalized exemplar, nearest neighbour, fuzzy NGE. 
 **Bag-of-Word** Bag-of-Words ['paper', 'used', 'learning', 'systems', 'feature', 'theory', 'based', 'input', 'algorithm', 'generalizations', 'algorithms', 'given', 'hypotheses', 'form', 'attributes', 'examples', 'general', 'concepts', 'instances', 'evaluating', 'prediction', 'classification', 'space', 'keywords', 'presents', 'pattern', 'version', 'receiving', 'mainly', 'techniques', 'nature', 'sets', 'wide', 'right', 'makes', 'variety', 'attention', 'example', 'original', 'describing', 'distance', 'generalized', 'inductive', 'nearest', 'dimensional', 'descriptions', 'value', 'incremental', 'designed', 'viewed', 'discusses', 'derived', 'fuzzy', 'pairs', 'generalizes', 'producing', 'defined', 'descent', 'induced', 'nn', 'wrong', 'criteria', 'reliability', 'proposes', 'metric', 'induce', 'euclidean'] |

Table 17: 3 abstracts and their representation from the CoraML dataset obtained after selecting the abstracts GPN has assigned the highest feature evidence. Abstracts sorted in descending order of feature evidence.

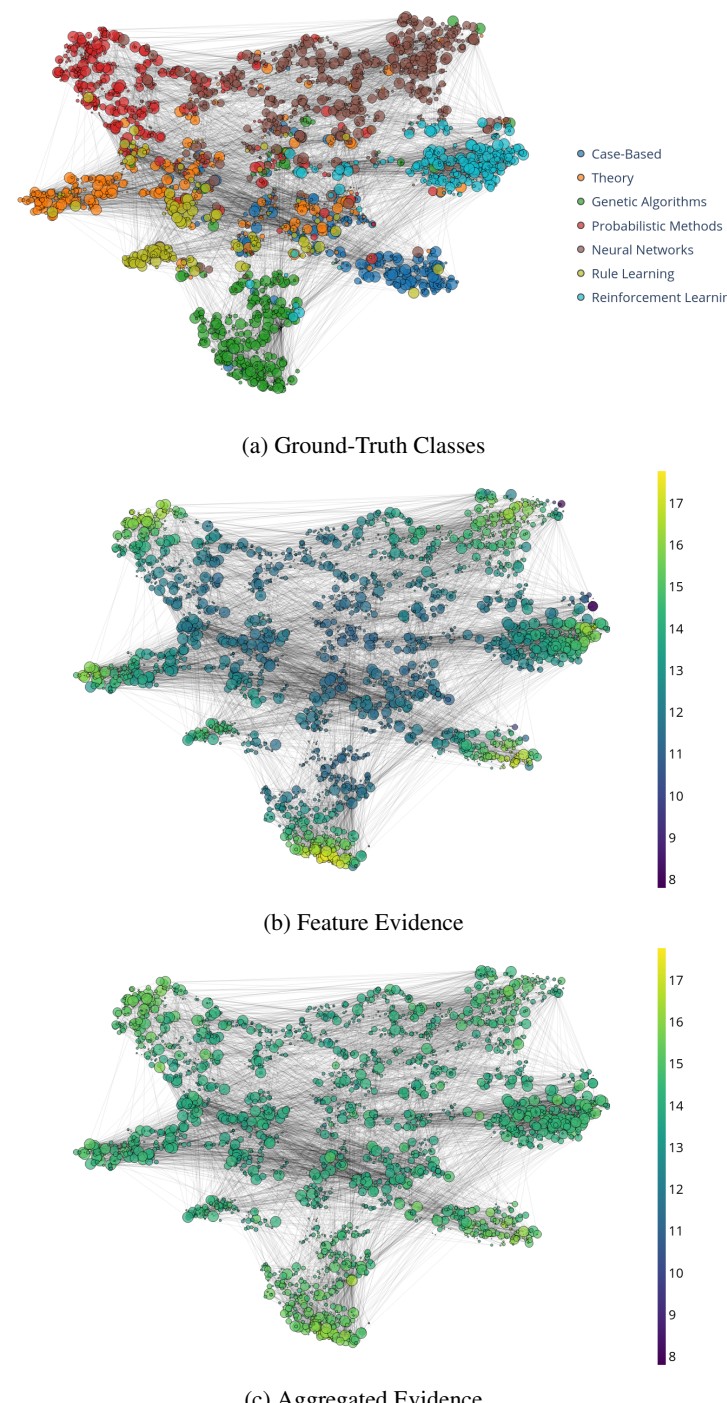

(a) Ground-Truth Classes

(b) Feature Evidence

(c) Aggregated Evidence

Figure 13: Latent space visualizations on the clean CoraML graph. The ground-truth classes are shown in different colors. The feature and aggregated evidence are represented in log-scale.

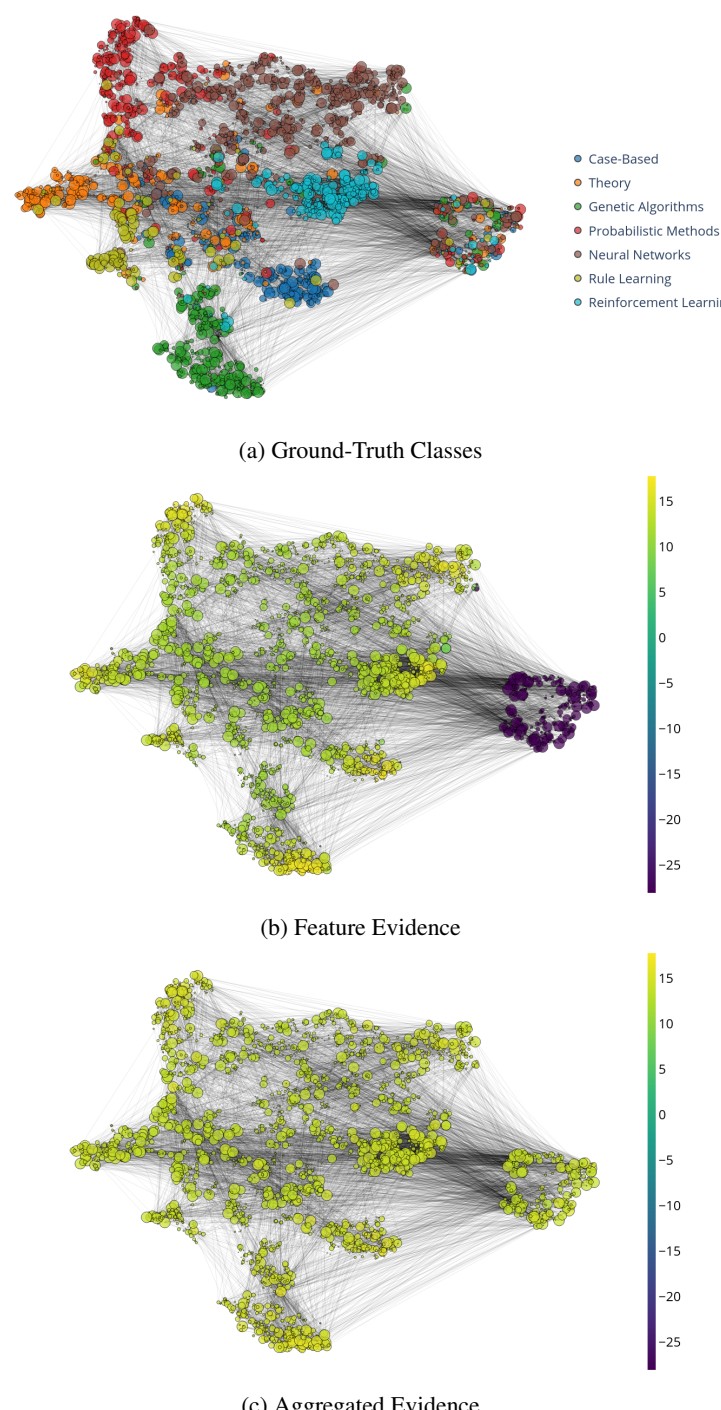

(a) Ground-Truth Classes

(b) Feature Evidence

(c) Aggregated Evidence

Figure 14: Latent space visualizations on the CoraML where $10\%$ of the nodes are perturbed with unit Gaussians. The ground-truth classes are shown in different colors. The feature and aggregated evidence are represented in log-scale.

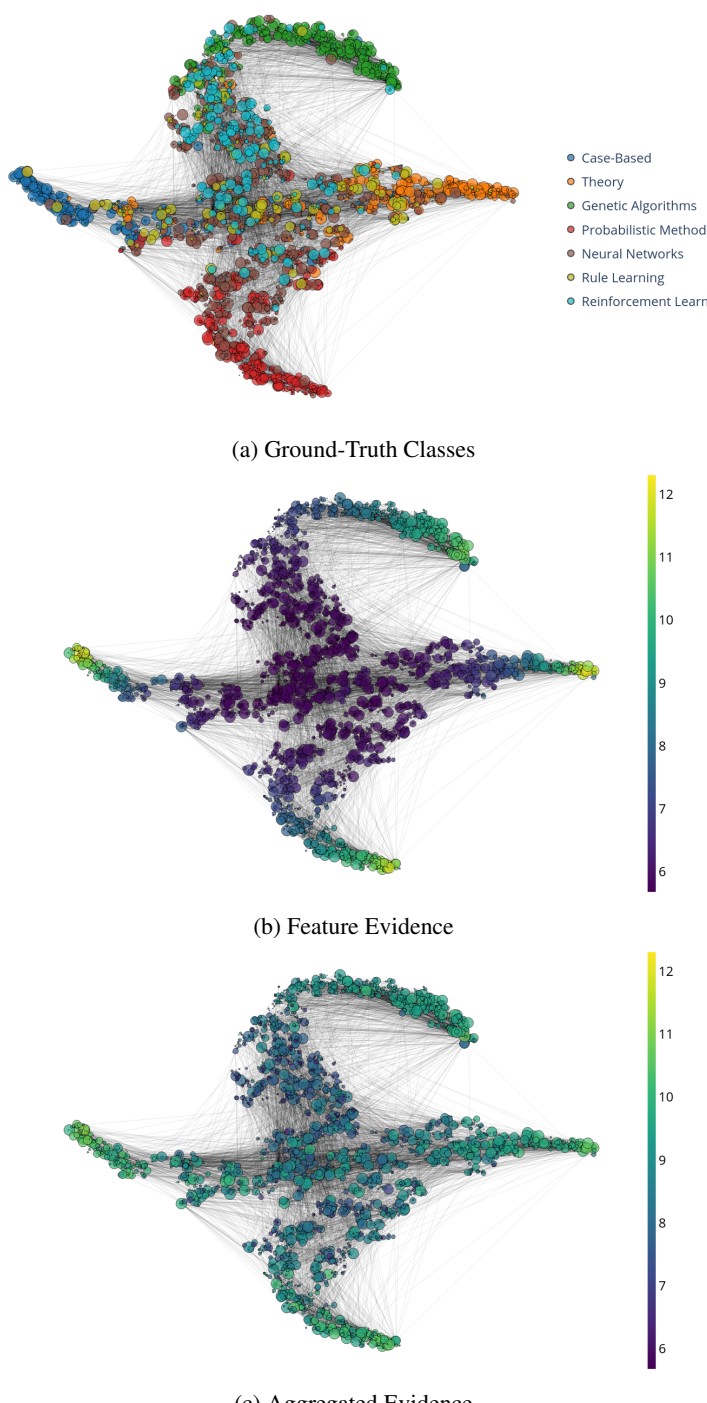

(a) Ground-Truth Classes

(b) Feature Evidence

(c) Aggregated Evidence

Figure 15: Latent space visualizations on the CoraML with Left-Out classes experiments. The ground-truth classes are shown in different colors. Nodes of the classes *Neural Networks*, *Rule Learning*, and *Reinforcement Learning* have been left out for training but are kept in the graph. The feature and aggregated evidence are represented in log-scale.

| Model | CoraML | CiteSeer | PubMed | Amazon C. | Amazon Ph. | Coauthor CS | Coauthor Ph. | OGBN Arxiv |
|---|---|---|---|---|---|---|---|---|
| APPNP | $2.87 \pm 0.01$ | $2.97 \pm 0.01$ | $3.07 \pm 0.02$ | $6.53 \pm 0.05$ | $3.50 \pm 0.03$ | $6.67 \pm 0.01$ | $10.33 \pm 0.04$ | $\mathbf{49.50 \pm 0.33}$ |
| VGCN | $\mathbf{2.55 \pm 0.04}$ | $2.73 \pm 0.02$ | $2.70 \pm 0.01$ | $6.41 \pm 0.09$ | $3.96 \pm 0.03$ | $6.42 \pm 0.02$ | $12.27 \pm 0.04$ | $60.90 \pm 0.04$ |
| VGCN-Dropout | $24.40 \pm 0.17$ | $26.96 \pm 0.25$ | $30.43 \pm 0.20$ | $58.24 \pm 0.20$ | $36.56 \pm 0.13$ | $62.21 \pm 0.26$ | $120.68 \pm 0.21$ | $637.63 \pm 0.43$ |
| VGCN-Energy | $2.78 \pm 0.03$ | $2.78 \pm 0.02$ | $3.24 \pm 0.03$ | $6.03 \pm 0.03$ | $3.79 \pm 0.01$ | $6.59 \pm 0.02$ | $12.46 \pm 0.03$ | $62.45 \pm 0.32$ |
| VGCN-Ensemble | $23.18 \pm 0.63$ | $23.10 \pm 0.02$ | $27.22 \pm 0.42$ | $52.78 \pm 0.65$ | $32.81 \pm 0.14$ | $54.17 \pm 0.46$ | $108.58 \pm 0.51$ | $548.27$ |
| GKDE-GCN | $2.61 \pm 0.03$ | $\mathbf{2.18 \pm 0.01}$ | $\mathbf{2.57 \pm 0.02}$ | $\mathbf{3.54 \pm 0.04}$ | $\mathbf{3.00 \pm 0.10}$ | $\mathbf{4.92 \pm 0.02}$ | $\mathbf{8.79 \pm 0.09}$ | $62.21 \pm 0.34$ |
| GPN | $9.35 \pm 0.06$ | $9.02 \pm 0.02$ | $14.13 \pm 0.02$ | $27.48 \pm 0.09$ | $14.52 \pm 0.05$ | $48.07 \pm 0.15$ | $35.94 \pm 0.13$ | $275.69 \pm 0.91$ |

Table 18: Inference time (in ms) across different datasets evaluated on a single NVIDIA GTX 1080 Ti. Bold numbers indicate the fastest algorithm during inference for each dataset while red numbers indicate the slowest one.

| Model | CoraML | CiteSeer | PubMed | Amazon C. | Amazon Ph. | Coauthor CS | Coauthor Ph. | OGBN Arxiv |
|---|---|---|---|---|---|---|---|---|
| APPNP | 45.44 | 27.74 | **54.92** | 257.55 | 230.75 | 166.84 | 217.73 | 425.75 |
| VGCN | 47.28 | 32.98 | 55.77 | 211.79 | 174.00 | 167.77 | 194.53 | **329.43** |
| VGCN-Dropout | 47.28 | 32.98 | 55.77 | 211.79 | 174.00 | 167.77 | 194.53 | **329.43** |
| VGCN-Energy | 47.28 | 32.98 | 55.77 | 211.79 | 174.00 | 167.77 | 194.53 | **329.43** |
| VGCN-Ensemble | 472.82 | 329.84 | 557.65 | 2117.86 | 1740.00 | 1677.72 | 1945.34 | 3294.34 |
| GKDE-GCN | 46.48 | **17.77** | 523.41 | 354.66 | 473.71 | 247.01 | 475.40 | 60185.40 |
| GKDE | 5.12 | 5.04 | 54.72 | 61.27 | 21.24 | 103.34 | 320.98 | 59775.90 |
| GPN | **10.20** | 39.40 | 59.15 | **81.59** | **64.72** | **32.80** | **93.38** | 2393.03 |

Table 19: Average training times (in s) for a single model and initializations across different datasets evaluated mostly on a single NVIDIA GTX 1080 Ti. Bold numbers indicate the fastest algorithm during training for each dataset while red numbers indicate the slowest one. The gray line shows the GKDE component of the GKDE-GCN approach as reference.

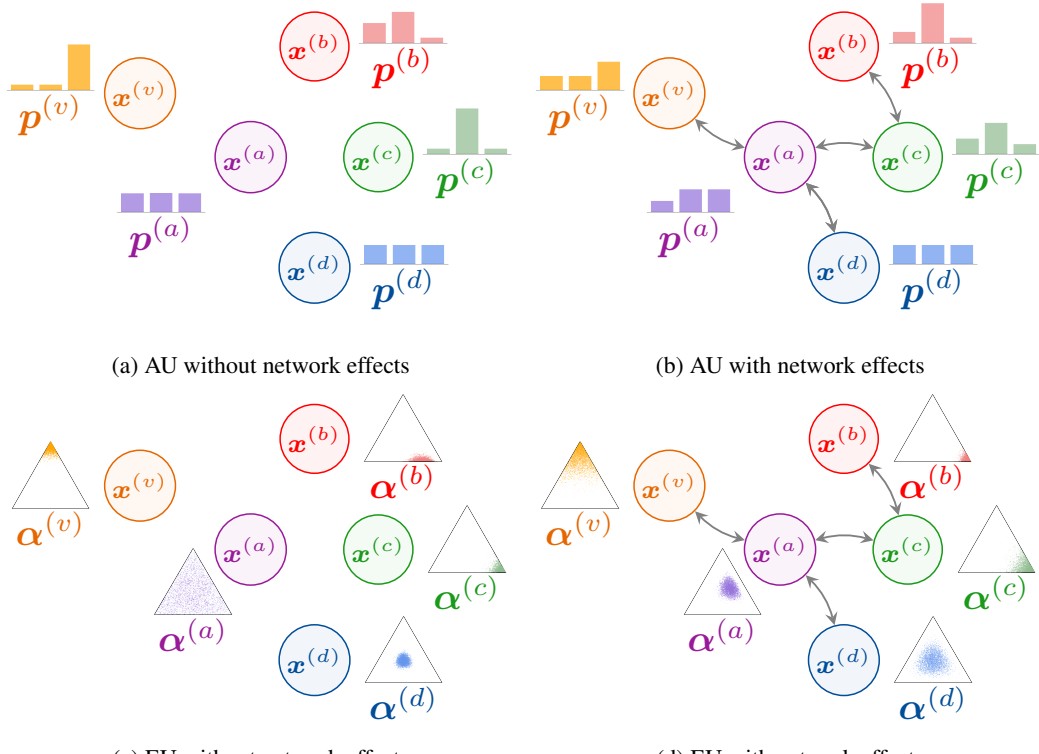

(a) AU without network effects

(b) AU with network effects

(c) EU without network effects

(d) EU with network effects

Figure 16: Illustration of aleatoric uncertainty (AU) and epistemic uncertainty (EU) without and with network effects (i.e. i.i.d. inputs vs interdependent inputs). Each node of one color has the same features in all four cases. Network effects are visualized through edges between nodes which change the predicted distributions. The aleatoric uncertainty is high if the categorical distribution $\hat{y}^{(v)} \sim \mathrm{Cat}(p^{(v)})$ is flat. The epistemic uncertainty is high if the Dirichlet distribution $p^{(v)} \sim \mathrm{Dir}(\alpha^{(v)})$ is spread out. Note that node with high epistemic certainty in the absence of network effects (e.g. orange) get less certain with neighbors being epistemically uncertain (purple). Epistemically uncertain nodes (purple) get more certain with certain neighbors on the other hand. Similar effects are shown for aleatoric uncertainty. For more details behind that reasoning, see our axiomatic approach in Sec. 3.1.