# OpenReview forum: "Graph Posterior Network: Bayesian Predictive Uncertainty for Node Classification"
_NeurIPS.cc/2021/Conference — NeurIPS 2021 Poster_

### Official Review · Reviewer_H3WY · 2021-06-27

**Rating:** 7
**Confidence:** 4

**Summary:**

In this paper, the authors proposed a graph posterior network based on the uncertainty theory to enhance the node classification in graph learning. The uncertainty estimation based on the interdependent node inputs instead of assuming i.i.d inputs is interesting. Through theoretical analysis and experiments on 8 data sets, they claim that the proposed framework has advantages in node classification and OOD detection.

**Limitations And Societal Impact:**

The authors discussed the limitations of the proposed method.

**Main Review:**

In general, the node classification based on uncertainty is not a new problem. However, the proposed method study the influence of the interdependent inputs on GNN is new. Furthermore, the proposed Graph Posterior Network is a combination of two existing methods. One is APPNP [47], the other is PostNet [14]. It is good to see that the authors discussed the key difference between this paper and the above two papers. The authors provide relatively comprehensive related works from two perspectives (Uncertainty for i.i.d. inputs and Uncertainty for graphs).

This submission is technically sound. Most claims are well supported by theoretical analysis and experiments. The proposed method theoretically proves the appropriateness of the fusion of the two known methods. Furthermore, the authors also discussed the limitations of their approach, which is good for understanding the applicability of their method in real-world tasks. This work is complete.

The paper is well-written in several sections. However, the Axioms and Theorems in Section 3 seem too dense. I suggest the authors can draw a figure to show the relationships between each Axiom and each theorem. The authors had provided many experiment details in the Appendix to reproduce the results in the main paper.

The results shown in this paper seem important. The researchers or practitioners may like to use the ideas from this paper in their research works. According to the experiment results, this paper advances the state of the art in a demonstrable way. The proposed method is tested on several existing data. There are no unique data, unique theoretical, and unique experimental approaches.

My major comments are as follows,

1. In table 1, why baseline methods do not have values in [Epist w/o Net]? What is the meaning of '_'?

2. It is hard to see the efficiency of the proposed methods compared to the baseline methods.

My minor comments are as follows,

1. In line 359, Tab.3 should be Figure 3.

2. In line 375, Tab.10 should be Table 2.

**Time Spent Reviewing:**

4

---

> ### Author Response · Authors · 2021-08-09
> **Author Response to Reviewer H3WY**
>
> We would like to thank you for your valuable comments and suggestions. In particular, we appreciate the reviewer's positive feedback and we are eager to provide answers to the reviewer's questions. Further, we are happy to provide additional clarifications in case you have follow-up questions.
>
>
> **[Epist w/o Net] scores:**
> GPN is the only model under consideration which can disentangle uncertainty with and without network effects. Hence, the [Epist w/o Net] scores cannot be computed for baselines as they either all use the graph information for their uncertainty prediction or do not estimate epistemic uncertainty. We will replace the notation of “-” with “N/A” to highlight that those measures are not applicable for some models.
>
>
> **Efficiency:**
> We provide a comparison of inference times for most of the datasets and models under consideration in Tab B and a comparison of training times in Tab C. GPN needs a single pass for uncertainty estimation but requires the additional evaluation of one normalizing flow per class compared to APPNP. Hence, GPN brings a small computational overhead for uncertainty estimation at inference time. Furthermore, GPN is usually converging relatively fast during training and does not require precomputing kernel values. In contrast, GKDE-GCN requires the computation of the underlying Graph Kernel with a complexity of $O(N^2)$ where $N$ is the number of nodes in the graph (see app. Sec. D). Finally, GPN is significantly more efficient than dropout or ensemble approaches as it does not require training or evaluating multiple models.
>
> Tab B: Inference Time (in ms), NVIDIA GTX 1080 Ti
>
> | Model         | CoraML         | CiteSeer       | PubMed         | Amazon Computers   | Amazon Photos   | Coauthor CS     | Coauthor Physics   | OGBN-Arxiv      |
> |:--------------|:---------------|:---------------|:---------------|:------------------|:---------------|:---------------|:------------------|:----------------|
> | APPNP         | 2.87 +/- 0.01  | 2.97 +/- 0.01  | 3.07 +/- 0.02  | 6.53 +/- 0.05     | 3.50 +/- 0.03  | 6.67 +/- 0.01  | 10.33 +/- 0.04    | 49.50 +/- 0.33  |
> | VGCN          | 2.55 +/- 0.04  | 2.73 +/- 0.02  | 2.70 +/- 0.01  | 6.41 +/- 0.09     | 3.96 +/- 0.03  | 6.42 +/- 0.02  | 12.27 +/- 0.04    | 60.90 +/- 0.04  |
> | VGCN-Dropout  | 24.40 +/- 0.17 | 26.96 +/- 0.25 | 30.43 +/- 0.20 | 58.24 +/- 0.20    | 36.56 +/- 0.13 | 62.21 +/- 0.26 | 120.68 +/- 0.21   | 637.63 +/- 0.43 |
> | VGCN-Energy   | 2.78 +/- 0.03  | 2.78 +/- 0.02  | 3.24 +/- 0.03  | 6.03 +/- 0.03     | 3.79 +/- 0.01  | 6.59 +/- 0.02  | 12.46 +/- 0.03    | 62.45 +/- 0.32  |
> | VGCN-Ensemble | 23.18 +/- 0.63 | 23.10 +/- 0.02 | 27.22 +/- 0.42 | 52.78 +/- 0.65    | 32.81 +/- 0.14 | 54.17 +/- 0.46 | 108.58 +/- 0.51   | 548.27          |
> | GKDE-GCN      | 2.61 +/- 0.03  | 2.18 +/- 0.01  | 2.57 +/- 0.02  | 3.54 +/- 0.04     | 3.00 +/- 0.10  | 4.92 +/- 0.02  | 8.79 +/- 0.09     | 62.21 +/- 0.34  |
> | GPN           | 9.35 +/- 0.06  | 9.02 +/- 0.02  | 14.13 +/- 0.02 | 27.48 +/- 0.09    | 14.52 +/- 0.05 | 48.07 +/- 0.15 | 35.94 +/- 0.13    | 275.69 +/- 0.91 |
>
>
> Tab C: Average Training Times (in s), NVIDIA GTX 1080 Ti
>
> | Model         |   CoraML |   CiteSeer |   PubMed |   Amazon Computers |   Amazon Photos |   Coauthor CS |   Coauthor Physics | OGBN-Arxiv   |
> |:--------------|---------:|-----------:|---------:|------------------:|---------------:|-------------:|------------------:|:-------------|
> | APPNP         |    45.44 |      27.74 |    54.92 |            257.55 |         230.75 |       166.84 |            217.73 | 425.75       |
> | VGCN          |    47.28 |      32.98 |    55.77 |            211.79 |         174    |       167.77 |            194.53 | 329.43       |
> | VGCN-Dropout  |    47.28 |      32.98 |    55.77 |            211.79 |         174    |       167.77 |            194.53 | 329.43       |
> | VGCN-Energy   |    47.28 |      32.98 |    55.77 |            211.79 |         174    |       167.77 |            194.53 | 329.43       |
> | VGCN-Ensemble |   472.82 |     329.84 |   557.65 |           2117.86 |        1740    |      1677.72 |           1945.34 | 3294.34      |
> | GKDE-GCN      |    46.48 |      17.77 |   523.41 |            354.66 |         473.71 |       247.01 |            475.4  | 60185.40     |
> | GKDE          |     5.12 |       5.04 |    54.72 |             61.27 |          21.24 |       103.34 |            320.98 | 59775.90     |
> | GPN           |    10.2  |      39.4  |    59.15 |             81.59 |          64.72 |        32.8  |             93.38 | 2393.03      |

---

### Official Review · Reviewer_jUVU · 2021-07-17

**Rating:** 7
**Confidence:** 5

**Summary:**

This paper introduced a well-grounded framework for uncertainty estimation on interdependent nodes. This paper first proposed explicit and motivated axioms describing desired properties for aleatoric and epistemic uncertainty in the absence or in the presence of network effects. Then it proposed GPN, a GNN for uncertainty estimation which provably follows our axioms. Furthermore, extensive experiments was conducted to evaluate the uncertainty performances of a broad range of baselines for OOD detection and robustness against node features or edge shifts. GPN outperforms all baselines in these experiments.

**Ethical Concerns:**

No Ethical Concerns

**Limitations And Societal Impact:**

Yes,  the authors adequately addressed the limitations and potential negative societal impact of this work.

**Main Review:**

This paper is well written and pertinent to the NeurIPS. The related work is extensive and impressive, I like the Axioms style. But I have some concerns about the experiment part.

1. Since GPN was designed based on APPNP and PostNet, it is necessary to do an ablation study to analyze the contribution of each component.

2. It might be necessary to add one important baseline, PostNet (ignore graph) or PostNet+GCN (simply combine PostNet and GCN), to show the advantage of GPN over PostNet in graph data.

3. For baseline GKD-GCN, it should be evaluated based on vacuity uncertainty, which is naturally designed to detect OOD. Note that GKD-GCN was proposed to improve the estimation of vacuity.

4. It is better to evaluate the proposed method in the misclassification detection task, which is an important task in uncertainty estimation [1] [2]. And this paper also discussed aleatoric uncertainty, which is not appropriate for OOD detection but is suitable for misclassification detection.

5. Why was GPN robust to feature perturbations, especially for accuracy? Which component supports this observation?

Some minor issues:

1. The indexes (1), (2), (3) are missing in Figure 2.
2. Typo in Table 1, “GDK-GCN” → “GKD-GCN”.
3. Typo in Line-358, “Tab. 3” → “Figure 3”

[1] Hendrycks, Dan, and Kevin Gimpel. "A baseline for detecting misclassified and out-of-distribution examples in neural networks." ICLR 2017.

[2] Malinin, Andrey, and Mark Gales. "Predictive uncertainty estimation via prior networks." NeurIPS 2018.

**Time Spent Reviewing:**

4

---

> ### Author Response · Authors · 2021-08-09
> **Author Response to Reviewer jUVU**
>
> We would like to thank you for your valuable comments and suggestions. Furthermore, we are happy to address the review's concerns about the experimental part. We are happy to provide additional explanations in case you have follow-up questions.
>
>
> **Ablated models \& Baselines (1 \& 2):**
> We provided ablated models and further baselines in appendix E.1 and referred to it in the main text (see l.291-296). This includes APPNP, PostNet, PostNet with graph diffusion at test time. While APPNP does not produce epistemic uncertainty scores, PostNet is not able to generalize well in the graph domain by just incorporating information from node features. We considered PostNet with graph diffusion at test time as another baseline which accounts for the graph information and can disentangle between uncertainty without and with network effect contrary to a GCN-like architecture. The results show indeed the advantage of GPN for uncertainty estimation and validate the core components of GPN.
>
>
> **Vacuity for GKDE (3):**
> We compared all models with Dirichlet parametrization with the same uncertainty scores (i.e. the pseudo-count $\alpha_{0}$ ) for fairness. Note that there exists a monotonic mapping from pseudo-counts to vacuity scores i.e. $vacuity = \frac{K}{\alpha_{0}}$ . Hence, the AUC-ROC and AUC-PR scores which evaluate the ranking of the examples lead to the *exact* same final scores. We are happy to add this equivalence in Sections C and D in the appendix when presenting details of metrics and models used in our experiments for clarity.
>
>
> **Misclassification task (4):**
> In addition to the calibration and OOD scores, we now also report results for misclassification detection with aleatoric and epistemic uncertainty on several datasets and models in Tab A1 to Tab A8 for the sake of completeness and comparability. GPN performs competitively with the baselines. Moreover, we observe indeed that epistemic uncertainty is better for OOD detection and aleatoric uncertainty is better for misclassification detection as for example also observed in [100].
>
>
> **Robustness to feature perturbation (5):**
> The aggregation step can recover from perturbed features (see l. 339-343). Equation (2) shows the relation between feature and aggregated pseudo-counts. For a node $v$ with perturbed features, the feature pseudo-counts will be small i.e. $\beta_c^{(v)} \approx 0$. Hence the aggregated pseudo-counts will mostly rely on the pseudo-counts $\beta_c^{(v)}$ from nodes $u$ with high PPR scores and will discard $\beta_c^{(v)}$ stemming from perturbed features. We also illustrate this behavior with the visualization of feature and aggregated evidence scores on CoraML in Fig. 14 in the appendix. Moreover, this is in line with related work like [23, A] that postulates that the final prediction should be obtained after discarding uncertain nodes in the message passing process.
>
>
> [A] M. Orbach, \& K. Crammer. Graph-based transduction with confidence. ECML PKDD 2012.
>
>
> Tab A1: Misclassification Experiments on CoraML
>
> | Model         | Alea w/ Net AUC-ROC   | Epis w/ Net AUC-ROC   | Epis w/o Net AUC-ROC   |
> |:--------------|:----------------------|:----------------------|:-----------------------|
> | APPNP         | **83.64 +/- 0.08**    | N/A                   | N/A                    |
> | VGCN          | 81.02 +/- 0.07        | N/A                   | N/A                    |
> | VGCN-Dropout  | 81.42 +/- 0.06        | 65.52 +/- 0.28        | N/A                    |
> | VGCN-Ensemble | 81.12 +/- 0.17        | 72.62 +/- 0.19        | N/A                    |
> | VGCN-BNN      | 80.64 +/- 0.10        | 65.40 +/- 0.48        | N/A                    |
> | GDK-GCN       | 80.80 +/- 0.14        | 76.83 +/- 0.17        | N/A                    |
> | GPN           | 81.19 +/- 0.13        | 78.10 +/- 0.26        | 77.46 +/- 0.24         |
>
>
> Tab A2: Misclassification Experiments on CiteSeer
>
> | Model         | Alea w/ Net AUC-ROC   | Epis w/ Net AUC-ROC   | Epis w/o Net AUC-ROC   |
> |:--------------|:----------------------|:----------------------|:-----------------------|
> | APPNP         | 73.55 +/- 0.08        | N/A                   | N/A                    |
> | VGCN          | 74.64 +/- 0.09        | N/A                   | N/A                    |
> | VGCN-Dropout  | 74.81 +/- 0.11        | 64.09 +/- 0.16        | N/A                    |
> | VGCN-Ensemble | 74.42 +/- 0.26        | 68.15 +/- 0.23        | N/A                    |
> | VGCN-BNN      | 73.28 +/- 0.11        | 61.68 +/- 0.29        | N/A                    |
> | GDK-GCN       | 75.45 +/- 0.11        | 73.83 +/- 0.12        | N/A                    |
> | GPN           | **75.89 +/- 0.15**    | 74.16 +/- 0.17        | 72.50 +/- 0.12         |
>
>
> Tab A3: Misclassification Experiments on PubMed
>
> | Model         | Alea w/ Net AUC-ROC   | Epis w/ Net AUC-ROC   | Epis w/o Net AUC-ROC   |
> |:--------------|:----------------------|:----------------------|:-----------------------|
> | APPNP         | 80.98 +/- 0.02        | N/A                   | N/A                    |
> | VGCN          | 81.16 +/- 0.02        | N/A                   | N/A                    |
> | VGCN-Dropout  | 80.46 +/- 0.04        | 72.69 +/- 0.09        | N/A                    |
> | VGCN-Ensemble | **81.31 +/- 0.06**    | 79.30 +/- 0.08        | N/A                    |
> | VGCN-BNN      | 79.96 +/- 0.07        | 72.63 +/- 0.45        | N/A                    |
> | GDK-GCN       | 80.95 +/- 0.09        | 73.99 +/- 0.27        | N/A                    |
> | GPN           | 80.46 +/- 0.13        | 75.38 +/- 0.25        | 80.48 +/- 0.13         |
>
>
> Tab A4: Misclassification Experiments on Amazon Computers
>
> | Model         | Alea w/ Net AUC-ROC   | Epis w/ Net AUC-ROC   | Epis w/o Net AUC-ROC   |
> |:--------------|:----------------------|:----------------------|:-----------------------|
> | APPNP         | 79.75 +/- 0.03        | N/A                   | N/A                    |
> | VGCN          | 82.08 +/- 0.03        | N/A                   | N/A                    |
> | VGCN-Dropout  | **82.70 +/- 0.04**    | 72.02 +/- 0.11        | N/A                    |
> | VGCN-Ensemble | 82.05 +/- 0.06        | 67.62 +/- 0.44        | N/A                    |
> | VGCN-BNN      | 82.15 +/- 0.17        | 48.65 +/- 0.82        | N/A                    |
> | GDK-GCN       | 79.66 +/- 0.19        | 73.66 +/- 0.15        | N/A                    |
> | GPN           | 82.20 +/- 0.10        | 77.58 +/- 0.16        | 70.06 +/- 0.19         |
>
>
> Tab A5: Misclassification Experiments on Amazon Photos
>
> | Model         | Alea w/ Net AUC-ROC   | Epis w/ Net AUC-ROC   | Epis w/o Net AUC-ROC   |
> |:--------------|:----------------------|:----------------------|:-----------------------|
> | APPNP         | 85.74 +/- 0.06        | N/A                   | N/A                    |
> | VGCN          | 87.94 +/- 0.05        | N/A                   | N/A                    |
> | VGCN-Dropout  | **89.52 +/- 0.05**    | 78.46 +/- 0.10        | N/A                    |
> | VGCN-Ensemble | 88.08 +/- 0.16        | 76.05 +/- 0.54        | N/A                    |
> | VGCN-BNN      | 84.17 +/- 0.19        | 51.84 +/- 0.66        | N/A                    |
> | GDK-GCN       | 84.11 +/- 0.29        | 75.07 +/- 0.50        | N/A                    |
> | GPN           | 87.21 +/- 0.10        | 83.38 +/- 0.30        | 79.93 +/- 0.23         |
>
> Tab A6: Misclassification Experiments on Coauthor CS
>
> | Model         | Alea w/ Net AUC-ROC   | Epis w/ Net AUC-ROC   | Epis w/o Net AUC-ROC   |
> |:--------------|:----------------------|:----------------------|:-----------------------|
> | APPNP         | 89.92 +/- 0.03        | N/A                   | N/A                    |
> | VGCN          | 89.46 +/- 0.03        | N/A                   | N/A                    |
> | VGCN-Dropout  | 88.46 +/- 0.04        | 79.03 +/- 0.08        | N/A                    |
> | VGCN-Ensemble | 89.51 +/- 0.08        | 86.61 +/- 0.09        | N/A                    |
> | VGCN-BNN      | 89.01 +/- 0.06        | 78.40 +/- 0.23        | N/A                    |
> | GDK-GCN       | 89.24 +/- 0.05        | 80.98 +/- 0.13        | N/A                    |
> | GPN           | 85.72 +/- 0.15        | 81.56 +/- 0.29        | **94.41 +/- 0.11**     |
>
>
> Tab A7: Misclassification Experiments on Coauthor Physics
>
> | Model         | Alea w/ Net AUC-ROC   | Epis w/ Net AUC-ROC   | Epis w/o Net AUC-ROC   |
> |:--------------|:----------------------|:----------------------|:-----------------------|
> | APPNP         | 93.27 +/- 0.02        | N/A                   | N/A                    |
> | VGCN          | 92.86 +/- 0.02        | N/A                   | N/A                    |
> | VGCN-Dropout  | 92.28 +/- 0.03        | 89.85 +/- 0.04        | N/A                    |
> | VGCN-Ensemble | 92.95 +/- 0.07        | 91.92 +/- 0.07        | N/A                    |
> | VGCN-BNN      | 92.44 +/- 0.09        | 89.03 +/- 0.27        | N/A                    |
> | GDK-GCN       | 92.77 +/- 0.02        | 86.12 +/- 0.06        | N/A                    |
> | GPN           | 91.14 +/- 0.04        | 89.63 +/- 0.07        | **93.89 +/- 0.05**     |
>
> Tab A8: Misclassification Experiments on OGBN-Arxiv
>
> | Model         | Alea w/ Net AUC-ROC   | Epis w/ Net AUC-ROC   | Epis w/o Net AUC-ROC   |
> |:--------------|:----------------------|:----------------------|:-----------------------|
> | APPNP         | 77.55 +/- 0.05        | N/A                   | N/A                    |
> | VGCN          | 77.89 +/- 0.05        | N/A                   | N/A                    |
> | VGCN-Dropout  | 78.11 +/- 0.05        | 71.74 +/- 0.14        | N/A                    |
> | VGCN-Ensemble | **78.14**             | 71.48                 | N/A                    |
> | GDK-GCN       | 77.47 +/- 0.33        | 77.55 +/- 0.33        | N/A                    |
> | GPN           | 75.44 +/- 0.19        | 72.71 +/- 0.28        | 61.45 +/- 0.49         |

---

> > ### Comment · Reviewer_jUVU · 2021-08-14
> > **Thanks for your response**
> >
> > Thanks for your response. Most of my concerns are addressed. I will increase my score.

---

### Official Review · Reviewer_fvEX · 2021-07-17

**Rating:** 6
**Confidence:** 4

**Summary:**

The paper makes three contributions: (i) three axioms are specified to characterize the requirements of predictive uncertainty behaviour in homophilic attributed graphs; (ii) a new inference model is proposed and theorems are provided that demonstrate how the model behaves with respect to its characterization of uncertainty; and (iii) via numerical experiments, it is demonstrated that the proposed model achieves better uncertainty estimation performance.

**Limitations And Societal Impact:**

Yes

**Main Review:**

The paper contains some very interesting ideas and the proposed model is well-constructed. The experiments are thorough, with comparison to multiple baselines across 8 datasets. These experiments demonstrate that the proposed model significantly outperforms existing methods in terms of its uncertainty quantification for the task of node classification.

On the negative side, the paper is very difficult to read, and there needs to be considerably more care with definitions of terms and explanations of concepts.

(1) The axioms are imprecisely stated and contain important elements that are not defined.
The figures used to explain the axioms introduce concepts and variables that are not defined or explained until later in Section 3.2. For example, it is not clear what Dirichlet distribution is being referred to.

In Axiom 3.1 it is written “A node with features more different from training features should be assigned higher uncertainty”. This sentence raises several questions. How is “different” defined or measured? What does it mean to “assign a node higher uncertainty”? When the axiom is imprecise in this way, it is very difficult to claim that a particular model obeys it. The authors claim that “The first theorem shows that GPN follows Ax. 3.1” but it actually shows something that is much more specific. It only characterizes the behaviour of the prediction as the distance (measured in any way) approaches infinity. It does not establish that for one node with “more different” features than another the uncertainty in the prediction will be higher (for some definition of “different” and some definition of “uncertainty”).

Axiom 3.2 states that “a node’s prediction… should have higher aleatoric uncertainty”. At this stage of the paper, there is no quantitative definition of aleatoric or epistemic uncertainty. The approaches to quantitatively assess these are only made clear in Section 3.2.

The text after Axiom 3.2 is confusing: “a node $v$ with confident feature predictions $x^{(v)}$. Above $x^{(v)}$ are specified as node features. Now they are node feature “predictions”?

In a similar vein, “more conflicting” in Axiom 3.3. is not defined.

Are the axioms supposed to relate to only a predictive model under a specific framework for uncertainty characterization? Or do we need to provide definitions and metrics for of all of these concepts – “uncertainty”, “aleatoric uncertainty”, “epistemic uncertainty”, “different”, “conflicting” – as well as the predictive model?

(2) It is extremely difficult to understand the proposed method and model from Section 3.2 alone. There is too much reliance on material from [14, 15]. While it is fine to refer a reader to other papers for further information and detail, the main method should be clear for a reader without it being essential to refer to other work. For example, phrases like “the epistemic uncertainty can be measured by the pseudo-count $\alpha_0$ of fictitious observations” are very hard to understand when “fictitious observations” have not been defined or introduced.

Overall, I would like to recommend the paper for acceptance based on its technical content, but I think the axioms and the presentation need major improvement.

***
After author response: I acknowledge that the authors have provided a thorough commentary regarding my criticisms and they can all be addressed by modifying the text (although I view these changes as being important, since I think some of the claims about the theoretical results are not strictly correct as they stand and the axioms are vague). I have raised my overall score.




**Time Spent Reviewing:**

4 hours

---

> ### Author Response · Authors · 2021-08-09
> **Author Response to Reviewer fvEX**
>
> We would like to thank you for your valuable comments and suggestions. Following your concerns, we would like to provide comments on and suggestions for our axiomatic sections and reliance to existing works. We are happy to provide additional clarifications in case you have follow-up questions.
>
> **Axioms definition (1):**
> We designed the axioms to be informal and generic for three main reasons. First, it allows them to intuitively indicate the expected behavior of the uncertainty estimation without complex mathematical notations similarly to [23, A]. E.g. using an arbitrary distance measure in the axioms might be unnecessarily math-heavy. Second, it makes the axioms application independent. Using a specific notion of e.g. "different" would already require to consider a specific application. Third, it makes the axioms model-agnostic. Indeed, it enables other models to instantiate the axioms with different existing formal definitions of the concepts of ”epistemic uncertainty”, “aleatoric uncertainty” or “different” which can be found in related work. For example, the epistemic uncertainty might refer to the variance of your predictions [29], the differentiable entropy of prior distributions [59], distance to all class centroids in feature space [B], the likelihood of latent representations [C], or the Dirichlet pseudo-count [14]. Similarly, the aleatoric uncertainty might refer to the max probability [59], the entropy of the predicted categorical distribution [59], or the simultaneous closeness of a sample to two centroids [B]. In contrast, we designed the theorems to be formal and instantiate one specific definition of these concepts i.e. “aleatoric uncertainty” refers to the entropy of the categorical distribution (see l. 254), the ”epistemic uncertainty” refers to the pseudo-count $\alpha_0$, and the concept of “difference” between clusters refers to their distance (see l.231). To clarify that the axioms are general descriptions of expected behavior for uncertainty estimations, we propose to change the term “Axiom” to “Postulate” or “Assumption”. Alternatively, we propose to make the axioms more formal by using the notations $u_{\text{alea}}^{(v)}$, $u_{\text{epist}}^{(v)}$, $x^{(v)}$ and $y^{(v)}$ introduced in l. 105-109. E.g. the first axiom would become "A node’s prediction $\hat{y}^{(v)}$ in the absence of network effects should only be influenced by its own features $x^{(v)}$. Given a measure of difference, a node with features far from training features should be assigned high uncertainty $u_{\text{alea}}^{(v)}$ and $u_{\text{epist}}^{(v)}$.". Note that the concepts of "far" has already been informally used to introduce concepts of "overconfidence far from training data" [38, 50, 65] or "garbage data" [F] in previous works.
>
>
>
> **Figure clarity (1)**:
> Figure 1 aims at giving an intuitive illustration of the concepts of aleatoric/epistemic uncertainty and without/with network effects early in the paper. While representing aleatoric uncertainty with histograms, epistemic uncertainty with Dirichlet triangles and network effects with edges is common in the literature (e.g. [14, 59, 60, 62, 84] use Dirichlet triangles to represent epistemic uncertainty), we explicitly recall in the caption that a flat categorical distribution indicates high aleatoric uncertainty, and that a spread out Dirichlet distribution indicates high epistemic uncertainty and edges indicate network effects. Hence, necessary concepts and variables are already introduced in the caption of the figure itself with more context following later while linking back to the figure. We are happy to add a forward pointer in the caption of Figure 1 to Sec. 3.2 with more detailed and formal explanations.
>
> **Reliance on existing works (2):**
> We considered the standard Bayesian update framework (incl. the term pseudo-observations) to be known as defined e.g. in [D, E, 6]. We recall the necessary material from [14, 15]. Indeed, we describe the input-dependent Bayesian update from [14] by indexing the standard Bayesian update with sample index $(v)$ in l.152-164 and we explicitly detail how PostNet computes the pseudo-counts for the input-dependent Bayesian update based on the encoder and normalizing flow steps in l.165-172. In addition to that, we are happy to recall the standard Bayesian framework for clarity as suggested by the reviewer.
>
> [A] M. Orbach, \& K. Crammer. Graph-based transduction with confidence. ECML PKDD 2012.
>
> [B] J. van Amersfoort, L. Smith, Y. W. Teh, and Y. Gal. Simple and Scalable Epistemic Uncertainty Estimation Using a Single Deep Deterministic Neural Network. ArXiv 2020.
>
> [C] J. Postels, H. Blum, Y. Strümpler, C. Cadena, R. Siegwart, L. Van Gool, F. Tombari. The Hidden Uncertainty in a Neural Network’s Activations. ArXiv 2021.
>
> [D] C. Bishop. Pattern Recognition and Machine Learning. Springer, 2006.
>
> [E] M. I Jordan. The exponential family: Conjugate priors. 2009.
>
> [F] Y. Gal, L. Smith. Sufficient Conditions for Idealised Models to Have No Adversarial Examples: a Theoretical and Empirical Study with Bayesian Neural Networks. ArXiv 2018.

---

> > ### Comment · Reviewer_fvEX · 2021-08-25
> > **Acknowledgment of response**
> >
> > Thank you for the response. The responses to all three reviews are impressively thorough.
> >
> > Overall, I thought that the technical content of the paper was of high quality. This includes the proposed model, the related theoretical development, and the experiments. I think the paper could be improved by being careful to introduce notation and define terms prior to their use in text and figures. I've re-read Section 3.2 and still think it lacks clarity, but neither of the other reviewers had a significant problem with the presentation of the material, so perhaps this is not such a problem.
> >
> > My main criticism was the axioms. I don't have a problem with axioms being informally stated if all of the components of the axiom are well-defined. However, even if informal terms are used, it should be made clear exactly what the axiom means, and what needs to be introduced/defined/instantiated for the axiom to hold. The axioms in [23] are also informal, but there is a critical difference that all of the terms in the axioms in [23] are introduced (and clearly defined) before the axioms are presented.
> >
> > I don't think you need to change the axioms to "assumptions", nor it is essential that they be formally stated. But I think the text preceding the axioms needs to clearly explain the expressions/concepts that are being used in the axioms. It should be made clear that the axioms pertain to an entire framework for uncertainty quantification, not just a predictive model. In this sense, in order for the axioms to be meaningful, the framework needs to specify definitions for epistemic uncertainty, aleatoric uncertainty, "different", "conflicting", and provide a predictive model.
> >
> > I also think there needs to be more care with the claims around the theorems. For example, I don't agree with the claim that the paper "provide[s] theoretical guarantees showing that GPN fulfills the three axioms". I like the theoretical contribution, but the established results are more limited than the broad requirements specified in the axioms.
> >
> > My major criticisms can be addressed relatively easily by modifying the text, but I think the necessary changes are important. Given the technical quality (and the other reviews), I would like to see the paper accepted, but I don't think it should be published in its current form. If it were a journal paper, my recommendation would be "accept with major revisions", and I would like to review a second version of the paper to make sure suitable changes have been implemented.
> >
> > In light of the response, I have raised my score.

---

### Decision · Program_Chairs · 2021-09-27

**Decision:**

Accept (Poster)

**Comment:**

This paper first proposes axioms describing desired properties of uncertainty in the absence or in the presence of network effects. It then  introduces a novel framework for uncertainty estimation in the graph setting (graph posterior networks). The problem of uncertainty estimation in the graph setting is interesting and relevant and has received much less attention than more standard supervised learning settings. The approach is technically correct. AN extensive set of the experiments includes the assessment of uncertainty estimation quality for OOD detection as well as the assessment of robustness against shifts in attributed graph properties. The proposed approach shows promising performance in these experiments. Following the author response and discussion, the reviewers were in agreement that the paper should be accepted. One concern raised in several of the reviews was that the paper could benefit from reducing the density of the exposition and the reliance on reader familiarity with results from prior works. The authors have committed to addressing these issues in the revised manuscript.